# Extracting Rare Dependence Patterns via Adaptive Sample Reweighting

**Yiqing Li** [* 1]  **Yewei Xia** [* 1 2]  **Xiaofei Wang** [1 3]  **Zhengming Chen** [1 4]  **Liuhua Peng** [5]
**Mingming Gong** [1 5]  **Kun Zhang** [1 6]

## Abstract

Discovering dependence patterns between variables from observational data is a fundamental issue in data analysis. However, existing testing methods often fail to detect subtle yet critical patterns that occur within small regions of the data distribution–patterns we term *rare dependence*. These rare dependencies obscure the true underlying dependence structure in variables, particularly in causal discovery tasks. To address this issue, we propose a novel testing method that combines kernel-based (conditional) independence testing with adaptive sample importance reweighting. By learning and assigning higher importance weights to data points exhibiting significant dependence, our method amplifies the dependence patterns and detects them successfully. Theoretically, we analyze the asymptotic distributions of the statistics in this method and show the uniform bound of the learning scheme. Furthermore, we integrate our tests into the PC algorithm, a constraint-based approach for causal discovery, equipping it to uncover causal relationships even in the presence of rare dependence. Empirical evaluation of synthetic and real-world datasets comprehensively demonstrates the efficacy of our method.

## 1. Introduction

Statistical independence testing, which aims to asses whether two variables are independent according to samples, is a fundamental problem in data analysis and scientific inference. Such tests have applications in a variety of fields. For example, in brain analysis using fMRI data, identifying which brain areas are involved in specific activities and determining statistical associations between these activities is crucial (Fan et al., 2017). Other applications include self-supervised learning (Li et al., 2021), feature selection (Candes et al., 2018), causal inference (Imbens & Rubin, 2015) and more.

**Example 1.1.** $X \sim U(-20, 20)$, $Y = s \cdot e^{-x^2} + \epsilon, \epsilon \sim \mathcal{N}(0, 0.25), s \in \{-1, 1\}$ *with equal probability.*

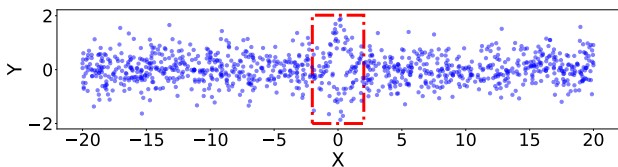

*Figure 1.* 1000 samples generated following Example 1.1. $p$-value of HSIC with default settings on the whole sample is 0.1359 while $p$-value on the samples within the red rectangle is $6.8 \times 10^{-11}$.

Recently, methods have been proposed to test for non-linear dependence (Gretton et al., 2005a; Póczos et al., 2012; Lopez-Paz & Oquab, 2016; Sen et al., 2017; Gao et al., 2018; Li et al., 2024b). Although existing independence tests have achieved significant success, they still fail to detect dependence patterns that are significant only within a small range of the entire distribution's support, which we term **rare dependence**. For instance, in Example 1.1, the dependence pattern here "exists everywhere", but it is only significant in a small range of $X$ near zero and the dependence is further contaminated by the noise. Many conventional independence tests, such as HSIC (Gretton et al., 2005a), struggle to reject the null hypothesis for the samples shown in Fig. 1.

Such rare dependence patterns are also commonly observed in real-world scenarios. In economics, income and consumption may appear weakly related in low-income groups but become strongly coupled at higher income levels. In psychology (Keles et al., 2020), the impact of so-

---

*Equal contribution [1]Department of Machine Learning, Mohamed Bin Zayed University of Artificial Intelligence, Abu Dhabi, UAE [2]Shanghai Key Lab of Intelligent Information Processing, and School of Computer Science, Fudan University, Shanghai, China [3]KLAS and School of Mathematics and Statistics, Northeast Normal University, Changchun, China [4]College of Mathematics and Computer, Shantou University, Shantou, China [5]School of Mathematics and Statistics, The University of Melbourne, Melbourne, VIC, Australia [6]Department of Philosophy, Carnegie Mellon University, Pittsburgh, USA. Correspondence to: Mingming Gong <mingming.gong@unimelb.edu.au>, Kun Zhang <kunz1@cmu.edu>.

*Proceedings of the $42^{nd}$ International Conference on Machine Learning*, Vancouver, Canada. PMLR 267, 2025. Copyright 2025 by the author(s).

cial media on adolescent mental health becomes significant only with excessive usage. Similar rare dependencies occur in medicine (Angst & Clark, 2006), biomedical inference (Tikka et al., 2019), physics (Aryasetiawan et al., 2004; Hwang et al., 2023), autonomous driving (Hwang et al., 2024), online advertising (Oentaryo et al., 2014), and sociology (Huang, 2017), highlighting the practical relevance of detecting rare dependencies. Misclassifying these dependencies as independent can sometimes lead to severe consequences. For example, in causal discovery, such false negatives can propagate errors throughout the graph during the execution of the PC algorithm (Spirtes et al., 2000).

In Example 1.1, dependence is weak and difficult to detect on the whole sample. However, by focusing more "attention" on the red rectangle in Fig. 1, the dependence pattern becomes more obvious and easier to detect. Motivated by this, we design a framework that automatically identifies and amplifies the significantly dependent subsamples and then tests the independence accordingly through reweighted samples. Specifically, the framework assigns an importance weight to each data point, which is produced by a learnable reweighting function. This function is trained by maximizing the reweighted kernel-based dependence measure, which quantifies the dependence between variables calculated on the reweighted samples. The test results based on the data reweighted by the optimized function are then used to decide the null hypothesis. In summary, the proposed framework adaptively assigns more "attention" to the dependent sub-samples, successfully detecting rare dependence.

Besides, we extend the idea to detect conditional rare dependence. Moreover, we design a constraint-based method to discover causal relations in the presence of rare dependence, which incorporates the proposed reweighting tests. We summarize our contributions as follows.

- We propose a novel testing method that combines kernel-based independence tests with adaptive sample importance reweighting to detect dependence even when rare dependence exists. The data point exhibits a more significant dependence pattern will be automatically assigned with larger weights.

- We also extend the idea to detect conditional rare independence. In addition, we integrate our tests into the PC algorithm for causal discovery in the presence of rare dependence.

- Theoretically, we obtain the uniform bound of our learning scheme and derive the asymptotic properties of the importance reweighting statistics of the tests.

- Empirically, we conduct extensive experiments on synthetic and real-world data that demonstrate the efficacy of our method.

## 2. Background

**Notations**. We use uppercase letters $X, Y, Z$ to denote random variables[1]. Their domains are denoted by $\mathcal{X}, \mathcal{Y}, \mathcal{Z}$ respectively. Consider a continuous feature mapping $\psi : \mathcal{X} \mapsto \mathcal{F}_X$ with the corresponding measureable positive definite kernel $k_X := \langle \psi, \psi \rangle$, where $\mathcal{F}_X$ is the corresponding Reproducing Kernel Hilbert Space (RKHS). We set $\psi_X = \psi(X)$ for simplicity. We denote the probability distribution of $X$ as $\mathbb{P}_X = \mathbb{P}(X)$, and the corresponding square-integrable spaces as $\mathcal{L}_X^2$. We assume that $\mathcal{F}_X \subset \mathcal{L}_X^2$. The notations for $Y, Z$ are defined by analogy with $\phi$ and $\varphi$ as feature mappings.

Assume the observed samples $\mathcal{D} = \{(x_i, y_i)\}_{i=1}^n$ are independent and identically distributed (i.i.d.) samples drawn from $\mathbb{P}_{XY}$. Independence tests calculate the test statistics $T : \mathcal{X} \times \mathcal{Y} \to \mathbb{R}$ from observed data $\mathcal{D}$ to summarize the information about the hypothesis

$$\mathcal{H}_0 : X \perp\!\!\!\perp Y \qquad \text{v.s.} \qquad \mathcal{H}_1 : X \not\perp\!\!\!\perp Y.$$

The value of the statistics will be compared to a specific threshold to determine whether to reject the null hypothesis $\mathcal{H}_0$ or not. The performance of an independence test can be evaluated from two aspects: the rate of rejecting $\mathcal{H}_0$ when it is true (Type I error rate); and the rate of not rejecting $\mathcal{H}_0$ when it is false (Type II error rate). Good tests are expected to control the probability of type I errors by a significance level $\alpha$, and a large power, i.e., $1-$ Type II error rate.

Our statistical independence test is based on a kernel-based independence test, Hilbert-Schmidt Independence Criterion.

**Definition 2.1.** (Gretton et al., 2007) The Hilbert-Schmidt Independence Criterion between $X$ and $Y$, denoted as $\mathrm{HSIC}(X, Y)$, is the HS norm of the covariance operator

$$\|\Sigma_{XY}\|_{\mathcal{HS}}^2 = \|\mathbb{E}_{\mathbb{P}_{XY}}[(\psi_X - \mu_X) \otimes (\phi_Y - \mu_Y)]\|_{\mathcal{HS}}^2.$$

where $\mu_X \triangleq \mathbb{E}_{\mathbb{P}_X}[\psi(X)]$, $\mu_Y \triangleq \mathbb{E}_{\mathbb{P}_Y}[\phi(Y)]$, $\otimes$ is the tensor product, and $\|\cdot\|_{\mathcal{HS}}$ is the Hilbert-Schmidt norm.

For characteristic kernels (Gretton, 2015), the independence relationship $X \perp\!\!\!\perp Y$ can be judged by $\mathrm{HSIC}(X, Y) = 0$.

## 3. Method

In this section, we develop a principled framework for detecting dependence. We first describe our motivation with some examples (§ 3.1). We define the importance reweighting functions then derive and analyze the reweighted statistics with carefully defined importance reweighting functions (§ 3.2), and propose a novel framework, that is, an optimization problem to learn the reweighting function and an

---

[1]In some cases, if there is no confusion, it can also be used as sets of random variables.

algorithm to test for independence (§ 3.3). Next, a theoretical bound is given to guarantee the generalizability of our method (§ 3.4). Finally, an extension of our framework for conditional independence (CI) is proposed (§ 3.5).

### 3.1. Motivation

Recall the dependence we mentioned and exemplified in the introduction. We summarize them as "rare dependence" in our paper: the dependence between two variables is significant only on a small portion of the data. Next, we present another example to show the challenges of the independence test when rare dependence occurs. For all tests in this paper, we choose a significance level of $\alpha = 0.05$.

**Example 3.1.** $X \sim U(-4, 1)$. If $X \in [-4, 0) \cup (0.25, 1], Y \sim U(0, 1)$. If $X \in [0, 0.25], Y = 0.9 \cdot X + 0.1 \cdot \epsilon$ where $\epsilon \sim U(0, 1)$. It is clear that $X \not\perp\!\!\!\perp Y$ according to the data generating process, while HSIC test on whole samples (see scatter plot in Fig. 2) fails to catch the dependence.

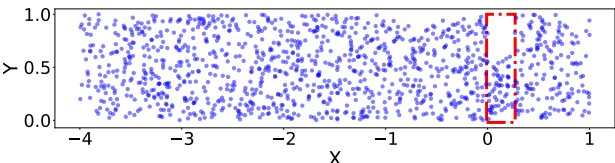

*Figure 2.* 1000 data points generated following Example 3.1. $p$-value of HSIC on whole sample is 0.0613, $p$-value of HSIC on samples within the red rectangle is 0.0027.

Notice that, when rare dependence happens, even collecting more data may not be helpful if we equally consider all data points. Inspired by the definition of "rare events" in (Wang, 2020; Wang et al., 2021), suppose that we have the sample $\{x_i, y_i, c_i\}_i^n \sim$ a kind of distribution that satisfies $\mathbb{P}(C = 1) \to 0, n\mathbb{P}(C = 1) \to \infty$ as $n \to \infty$, where $C \in \{0, 1\}$ with $C = 1$ when $X \not\perp\!\!\!\perp Y$ and $C = 0$ when $X \perp\!\!\!\perp Y$. As shown in Fig. 3, in this case the probability of Type II error will become larger as $n \to \infty$.

Now we turn to the reason why HSIC tests with subsamples can detect the dependence. Notice that the significantly dependent subsamples in both Example 1.1 and 3.1 can be tracked through the values of the variable $X$. And tests on these subsamples are able to reject independence. This observation leads to the proposed method, which reweights the data points using a learned function that takes a reference variable as the input. Note that although this paper mainly focus on HSIC, this behavior also holds in many other popular independence tests, e.g., randomized dependence coefficient (Lopez-Paz et al., 2013) etc.

### 3.2. Dependent Subsample Extraction by Reweighting

Let $X$ and $Y$ be two different random variables (or disjoint sets of variables), and suppose we fail to see a clear de-

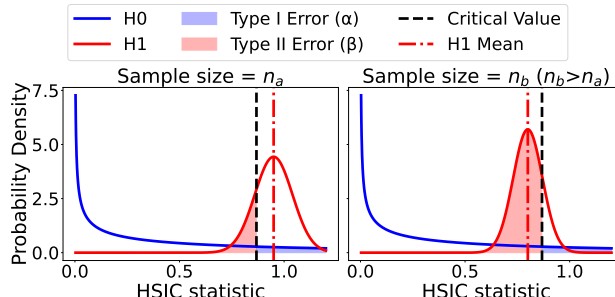

*Figure 3.* Illustration of distributions of HSIC under $\mathcal{H}_0$ and $\mathcal{H}_1$ in the "rare events" setting. Here $n_a$ and $n_b$ denote the sample sizes used in the two settings of the estimated statistical components (asymptotic distribution, mean value of the H1 distribution, and the critical value) with $n_b > n_a$. Note that under $\mathcal{H}_1$, as $n \to \infty$, the ratio of dependent sub-population goes to 0, leading to the mean value of HSIC approaches 0. Therefore the probability of Type II error will increase, making the tests prone to not reject $\mathcal{H}_0$.

pendence pattern for $(X, Y)$ directly. We aim to find the significantly dependent sub-population and change the original distribution by reweighting to enlarge the dependence. We use the weights produced by the reweighting functions $\beta(\cdot)$ as defined below.

**Definition 3.2** (Reweighting function and reweighted distribution). Let $\mathcal{B}$ be the set of reweighting functions,

$$\mathcal{B} \triangleq \left\{ \beta : \mathcal{C} \to \mathbb{R}^{\geq 0} \mid \mathbb{E}_{\mathbb{P}_{XY}}[\beta(C)] = 1 \right\}. \tag{1}$$

$C$ is a reference variable that can be either $X$ or $Y$ (or a subset of $X$ or $Y$) and $\mathcal{C}$ is the domain of $C$ correspondingly. Then $\forall \beta \in \mathcal{B}$, the corresponding reweighted distribution $\tilde{\mathbb{P}}$, which is well defined with the same support of $\mathbb{P}_{XY}$, can be determined by the probability density function (p.d.f.)

$$\tilde{\mathbb{P}}(X, Y) = \beta(C)\mathbb{P}(X, Y). \tag{2}$$

Since the case in which $X$ and $Y$ are sets of random variables are easily generalized, we regard $X$ and $Y$ as random variables in the following discussions for simplicity. If $X \perp\!\!\!\perp Y$ holds, it is imperative to avoid introducing spurious dependence after reweighting, as formalized in the following proposition:

**Proposition 3.3** (Maintain the independence after reweighting). *If $X$ and $Y$ are independent and $C$ is either $X$ or $Y$ but not both, then $X$ and $Y$ are still independent in the reweighted distribution of $(X, Y)$ with weight $\beta(C)$.*

The proof can be found in Appendix C.1. Since it is symmetric with respect to $X$ or $Y$, we set $C = X$ in the following discussion, unless stated otherwise.

As mentioned in Section 3.1, even though a rare dependence exists, it is still possible to detect it by identifying subsamples with significant dependence and increasing their importance ratios in the dependence measure. Therefore,

we aim to find an importance reweighting function $\beta(\cdot)$ to make the dependency pattern between variables as clear as possible. To compare different reweighting functions, we first propose a statistic that quantifies the dependence in the reweighted distribution $\tilde{\mathbb{P}}$ induced by a given reweighting function. This statistic can be directly computed from the original samples, eliminating the need for resampling.

**Reweighted Hilbert-Schmidt Independence Criterion**

With a fixed function $\beta(\cdot)$, at the population level, the HSIC calculated on the reweighted distribution $\tilde{\mathbb{P}}$ is

$$\left\| \mathbb{E}_{\tilde{\mathbb{P}}} \left[ (\psi_X - \mathbb{E}_{\tilde{\mathbb{P}}}[\psi_X]) \otimes (\phi_Y - \mathbb{E}_{\tilde{\mathbb{P}}}[\phi_Y]) \right] \right\|_{\mathcal{HS}}^2, \quad (3)$$

where $\tilde{\mathbb{P}} = \tilde{\mathbb{P}}_{XY} = \beta(X)\mathbb{P}_{XY}$. Rewrite it including only the initial distribution $(X, Y) \sim \mathbb{P}_{XY}$ and $\beta(\cdot)$, we obtain the reweighted HSIC (RHSIC), $\mathrm{HSIC}^\beta(X, Y) \triangleq$

$$\|\mathbb{E}_{\mathbb{P}}[\beta(X)(\psi_X - \mathbb{E}_{\mathbb{P}}[\beta(X)\psi_X]) \otimes (\phi_Y - \mathbb{E}_{\mathbb{P}}[\beta(X)\phi_Y])]\|_{\mathcal{HS}}^2.$$

Consider the test statistic on the dataset $\mathcal{D} = \{(x_i, y_i)\}_{i=1}^n$. Denote $\beta_k \triangleq \beta(x_k)$ as the weight for a data point $(x_k, y_k)$, and $\beta$ is the weight vector for $\mathcal{D}$. A biased estimator of $\mathrm{HSIC}^\beta(X, Y)$ using plug-in estimation method is

$$\mathrm{HSIC}_b^\beta(\mathcal{D}) = \frac{1}{n^2} \mathrm{Tr}\left[ \boldsymbol{K}_X \boldsymbol{H}_\beta \boldsymbol{K}_Y \boldsymbol{H}_\beta \right], \quad (4)$$

where $\boldsymbol{K}_X$ and $\boldsymbol{K}_Y$ are kernel matrices calculated on samples, i.e., $[\boldsymbol{K}_X]_{i,j} = k_X(x_i, x_j), [\boldsymbol{K}_Y]_{i,j} = k_Y(y_i, y_j)$. $\boldsymbol{H}_\beta \triangleq \boldsymbol{D}_\beta(\boldsymbol{I} - \frac{1}{n}\boldsymbol{11}^T\boldsymbol{D}_\beta)$, $\boldsymbol{I}$ is the identity matrix, $\boldsymbol{1}$ is the vector of all 1s, and $\boldsymbol{D}_\beta \triangleq \mathrm{diag}(\beta_1, \ldots, \beta_n)$. When $\beta_k \equiv 1$, it becomes the original HSIC.

We learn the optimal reweighting function to reweight the samples and test independence by maximizing this reweighted statistic. Next we analyze its asymptotic behaviors under $\mathcal{H}_0$ and $\mathcal{H}_1$.

**Asymptotic Distribution for Reweighted HSIC**

We now describe the null distributions of the test statistics given a known reweighting function $\beta(\cdot)$. Suppose $\mathcal{D} = \{w_i\}_{i=1}^n = \{(x_i, y_i)\}_{i=1}^n$. We first define a symmetric function that satisfies $\mathrm{HSIC}_b^\beta(\mathcal{D}) = \frac{1}{n^4} \sum_{i,j,q,r}^n h_{ijqr}^\beta$ as

$$h_{ijqr}^\beta \triangleq \frac{1}{4!} \sum_{(s,t,u,v)}^{(i,j,q,r)} (\beta_s \beta_t k_X^{st} k_Y^{st}$$
$$+ \beta_s \beta_t \beta_u \beta_v k_X^{st} k_Y^{uv} - 2\beta_s \beta_t \beta_u k_X^{st} k_Y^{su}). \quad (5)$$

Here the sum represents all ordered quadruples $(s, t, u, v)$ drawn without replacement from $(i, j, q, r)$ and $\beta_i = \beta(x_i)$. Assume $\mathbb{E}[(h_{ijqr}^\beta)^2] < \infty$, we have We first define a symmetric function that satisfies

**Theorem 3.4** (Null distribution)**.** *Under* $\mathcal{H}_0$, *we have* $\mathbb{E}_i h_{ijqr}^\beta = 0$. *In this case,* $\mathrm{HSIC}_b^\beta(\mathcal{D})$ *converges in distribution to a weighted sum of* $\mathcal{X}^2$ *variables, i.e.,*

$$n\mathrm{HSIC}_b^\beta(\mathcal{D}) \xrightarrow{d} \sum_{l=1}^\infty \lambda_l^\beta \chi_{1l}^2, \quad (6)$$

*where* $\chi_{1l}^2$ *are i.i.d. chi-square variables with freedom one. Denote* $w_i \triangleq (x_i, y_i)$, $\lambda_l^\beta$ *are the solutions to the eigenvalue problem integrating over the distribution of variables* $w_i, w_q$, *and* $w_r$:

$$\lambda_l^\beta \psi_l(w_j) = \int \beta_{iqr} \cdot h_{ijqr}^\beta \psi_l(w_i) dF_{i,q,r}. \quad (7)$$

Next, we give a theorem about the asymptotic distribution when $\mathrm{HSIC}^\beta(X, Y) > 0$, i.e., $X \not\perp\!\!\!\perp Y$. This distribution would be useful in analyzing consistency.

**Theorem 3.5.** *When* $\mathrm{HSIC}^\beta(X, Y) > 0$, $\mathrm{HSIC}_b^\beta(\mathcal{D})$ *converges in distribution to a Gaussian according to:*

$$\sqrt{n}\left(\mathrm{HSIC}_b^\beta(\mathcal{D}) - \mathrm{HSIC}^\beta(X, Y)\right) \xrightarrow{d} \mathcal{N}(0, \sigma_\beta^2). \quad (8)$$

*The variance* $\sigma_\beta^2 = 16(\mathbb{E}_i(\mathbb{E}_{j,q,r} h_{ijqr}^\beta)^2 - \mathrm{HSIC}^\beta(X, Y)^2)$, *where* $\mathbb{E}_{j,q,r} \triangleq \mathbb{E}_{w_j, w_q, w_r}$.

### 3.3. Testing Procedure and Whole Algorithm

Recall that in Pro. 3.3 we require infinite samples. In practice, we only have access to finite samples, introducing dependence between the samples and the function $\beta(\cdot)$ thus breaking the factorizable condition. Empirically, it manifests as the estimated $\hat{\beta}(\cdot)$ overfitting the given samples and always reporting dependence even for independent variables. To avoid it, we split the data into two independent parts. Suppose we have i.i.d. samples $\mathcal{D} = \{(x_i, y_i)\}_{i=1}^n$. We randomly split it into disjoint training ($\mathcal{D}_{tr}$) and testing ($\mathcal{D}_{te}$) data. We optimize the function $\hat{\beta}(\cdot)$ on $\mathcal{D}_{tr}$, and then use it to perform a test on $\mathcal{D}_{te}$. The split ratio is set to 0.5.

**Proposition 3.6.** *Suppose the sample data are i.i.d. and randomly split into* $\mathcal{D}_{te} = \{x_{te}, y_{te}\}$ *and* $\mathcal{D}_{tr}$ *correspondingly.* $\hat{\beta}(\cdot)$ *maximizes the dependence measure of* $\mathcal{D}_{tr}$. *Then* $\hat{\beta}(\cdot)$ *is independent of* $\mathcal{D}_{te}$.

Therefore, if $X \perp\!\!\!\perp Y$, we have that $\hat{\beta}(x_{te})$ is also independent of $Y$, and thus no spurious dependence between $X$ and $Y$ would be introduced in the reweighted distribution using $\hat{\beta}(x_{te})$.

**Final Objective Function**

To ensure that our learned $\hat{\beta}(\cdot)$ satisfies Def. 3.2, we constrain $\hat{\beta}_i \geq 0$ and $\frac{1}{n}\sum_{i=1}^n \hat{\beta}_i = 1$. Besides, we add two regularization terms. First, we aim to select more data points to avoid trivial solutions (e.g., a few extreme $\beta_i$ values assigned to uninformative noise samples while all others

are set to zero). This is achieved by minimizing the sample variance of $\beta_i$, expressed as $\frac{1}{n}\sum_{i=1}^{n}(\beta_i - 1)^2$, which encourages the value of $\beta_i$ to approach 1. Second, we ensure the smoothness of the function $\beta(\cdot)$ by employing the smoothness functional, which could bring more stable generalization by avoiding abrupt changes for nearby inputs. Here we assume that $\beta(\cdot)$ belongs to an RKHS, while the approach can also be applied to other function spaces, such as neural networks. In an RKHS, with the reproducing property, we can represent $\beta(X)$ as $\langle \psi_X^T, \omega \rangle_{\mathcal{F}_X}$, where $\omega \triangleq \psi_X^T \alpha = \sum_{i=1}^{n} \alpha_i \psi(x_i)^T$, with $\alpha$ being the parameter vector. This formulation naturally results in the regularization term $\|\omega\|_{\mathcal{F}_X}^2 = \alpha^T \mathbf{K}_X \alpha$, which corresponds to the smoothness functional in an RKHS (Evgeniou et al., 2000).

In summary, we learn the reweighting function $\beta(\cdot)$ by solving the following constrained optimization problem:

$$\arg\min_{\beta} -\log \hat{J}_{\beta}^{UI} + \lambda_1 \|\omega\|_{\mathcal{F}_X}^2 + \frac{\lambda_2}{n}\sum_{i=1}^{n}(\beta_i - 1)^2,$$

$$s.t. \quad \beta_i \geq 0, \ \sum_{i=1}^{n}\beta_i = n, \tag{9}$$

where $\hat{J}_{\beta}^{UI} \triangleq \text{HSIC}_b^{\beta}(\mathcal{D}_{tr})$, $\lambda_1$ and $\lambda_2$ are hyperparameters that control the regularization terms. In practice, using a normalized version of the statistics during optimization may help avoid overfitting or amplifying noise. Therefore, in the experiment, we use $\hat{J}_{1\beta}^{UI} = \frac{\hat{J}_{\beta}^{UI}}{\hat{J}_{2\beta}^{UI}\hat{J}_{3\beta}^{UI}}$ instead of $\hat{J}_{\beta}^{UI}$, where $\hat{J}_{2\beta}^{UI} = \frac{1}{n}\text{Tr}\left[\mathbf{K}_x\mathbf{H}_{\beta}\right]$ and $\hat{J}_{3\beta}^{UI} = \frac{1}{n}\text{Tr}\left[\mathbf{K}_y\mathbf{H}_{\beta}\right]$.

The null distribution in (6) has a complex form (weighted sum of $\chi_1^2$s) and is hard to calculate the threshold. Instead, in implementation we use the permutation test to approximate the $1-\alpha$ quantile. It permutes the ordering of the $Y$ samples for $B$ times while that of $X$ is kept fixed, using the critical value of this statistic distribution to estimate the true one. The whole algorithm is included in Algorithm 1.

---

**Algorithm 1** Reweighted HSIC (RHSIC)

---

1: **Input:** $\mathcal{D}$: samples. $C$: reference variable. $\alpha$: significance level. $B$: the number of permutations.
2: **Output:** $p$-value and test statistics value.
3: Split $\mathcal{D}$ into $\mathcal{D}_{tr} = \{x_{tr}, y_{tr}\}$ and $\mathcal{D}_{te} = \{x_{te}, y_{te}\}$.
4: Optimize the constrained problem (9) on $\mathcal{D}_{tr}$, to obtain the reweighting function $\hat{\beta}(\cdot)$.
5: Use $\hat{\beta} = \hat{\beta}(x_{te})$ to calculate $T_{obs} = \text{HSIC}_b^{\hat{\beta}}(\mathcal{D}_{te})$.
6: **for** all $k \in \{1, \ldots, B\}$ **do**
7:      Permute $y_{te}$ to get $\tilde{y}_{te}^k$ and $\tilde{\mathcal{D}}_{te}^k = x_{te} \cup \tilde{y}_{te}^k$.
8:      Calculate $k$-th statistics $T_k = \text{HSIC}_b^{\hat{\beta}}(\tilde{\mathcal{D}}_{te}^k)$.
9: **end for**
10: Compute $p$-value by $p = \frac{1}{B}\sum_{k=1}^{B}\mathbb{I}[T_k \geq T_{obs}]$ where $\mathbb{I}$ denotes the indicator function.

---

### 3.4. Generalizability of the Learned Functions

By optimizing Problem (9), we can obtain an empirical optimal reweighting function $\hat{\beta}(\cdot)$ on $\mathcal{D}$. The key question remaining is whether $\hat{\beta}(\cdot)$ will generalize well to the population level $\mathbb{P}(X, Y)$. That is, given a sufficiently large dataset, can the difference between $\text{HSIC}^{\hat{\beta}}(X, Y)$ and the optimal reweighted dependence measure $\text{HSIC}^{\beta^*}(X, Y)$ diminish and convergence to 0. Through some analysis, we can see that $\text{HSIC}^{\hat{\beta}}(X, Y) - \text{HSIC}^{\beta^*}(X, Y)$ is bounded by $2\sup_{\beta \in \mathcal{B}}|\text{HSIC}^{\beta}(X, Y) - \text{HSIC}_b^{\beta}(\mathcal{D})|$. Therefore, to establish convergence, it suffices to show that as the sample size $n \to \infty$, $\text{HSIC}_b^{\beta}(\mathcal{D})$ converges to $\text{HSIC}^{\beta}(X, Y)$ for each $\beta \in \mathcal{B}$. If this condition holds, we can conclude the convergence property of $\text{HSIC}^{\hat{\beta}}(X, Y)$ to $\text{HSIC}^{\beta^*}(X, Y)$ as $n \to \infty$. In the following, we present this result.

**Theorem 3.7** (Uniform Bound). *Suppose $\mathcal{X} \subset \mathbb{R}^d$ is a closed and bounded space and the values of the kernels $k_X$ and $k_Y$ are also bounded. Assume that the reweighting functions $\beta \in \mathcal{B}$ are continuous and Lipschitz. Then with probability at least $1 - \delta$, we have*

$$\sup_{\beta \in \mathcal{B}}\left|\text{HSIC}_b^{\beta}(\mathcal{D}) - \text{HSIC}^{\beta}(X, Y)\right|$$

$$\sim \mathcal{O}\left(\sqrt{\frac{1}{n}\log\frac{1}{\delta}} + \frac{\log n}{n^{\frac{2}{3}}} + \frac{1}{n^{\frac{1}{3}}}\right). \tag{10}$$

The details of the proof can be found in Appendix D.5. This bound guarantees that if our optimization process succeeds, the value of $\text{HSIC}_b^{\hat{\beta}}(\mathcal{D})$ with empirically learned function can converge to the optimal value $\text{HSIC}^{\beta^*}(X, Y)$ as $n \to \infty$ with a rate shown in Eq. (10).

### 3.5. Conditional Independence Test Version

In this section, we extend our method to the Conditional Independence (CI) test. We first provide the conditional version of some previous results. The input of the reweighting function $\beta(\cdot)$ now also includes the conditioning set.

**Definition 3.8** (Conditional version of reweighting function and reweighted distribution). Let $C$ be either $X$ or $Y$ with domain $\mathcal{C}$, the reweighting function set $\mathcal{B}$ satisfies

$$\mathcal{B} = \left\{\beta : \mathcal{C} \times \mathcal{Z} \to \mathbb{R}^{\geq 0} \mid \mathbb{E}_{\mathbb{P}_{XY|Z}}[\beta(C, Z)] = 1\right\}. \tag{11}$$

Then $\forall \beta \in \mathcal{B}$, the corresponding reweighted distribution $\tilde{\mathbb{P}}_{XY|Z}$, which is well defined with the same support of $\mathbb{P}_{XY|Z}$, can be determined by the conditional p.d.f.

$$\tilde{\mathbb{P}}(X, Y \mid Z) = \beta(C, Z)\mathbb{P}(X, Y \mid Z). \tag{12}$$

**Proposition 3.9** (Maintain the conditional independence after reweighting). *If $X$ and $Y$ are conditionally independent given $Z$ and that $C$ is either $X$ or $Y$ but not both, then*

*X and Y are still conditionally independent given Z after being reweighted by the weight $\beta(C, Z)$.*

We then introduce the reweighted statistic to calculate the conditional dependence measure in the reweighted distribution without resampling data. Recall that HSIC is based on the idea that independence can be related to uncorrelatedness between functions in RKHS $\mathcal{F}_X$ and $\mathcal{F}_Y$. For the statistic ($J^{CI}$) of the kernel-based CI test (KCIT), uncorrelatedness is assessed between functions in restricted function spaces, where the effect of $Z$ is accounted for using methods such as regression (Zhang et al., 2012).

**Lemma 3.10** (Characterization based on conditional cross–covariance operators (Fukumizu et al., 2007))**.** *Denote $\ddot{X} \triangleq (X, Z), k_{\ddot{X}} \triangleq k_X k_Z$, and $\mathcal{F}_{\ddot{X}}$ the RKHS corresponding to $k_{\ddot{X}}$. Assume $\mathcal{F}_X \subset \mathcal{L}_X^2, \mathcal{F}_Y \subset \mathcal{L}_Y^2$, and $\mathcal{F}_Z \subset \mathcal{L}_Z^2$. Further assume that $k_{\ddot{X}} k_Y$ is a characteristic[2] kernel on $(\mathcal{X} \times \mathcal{Y}) \times \mathcal{Z}$, and that $\mathcal{F}_Z + \mathbb{R}$ (the direct sum of the two RKHSs) is dense in $\mathcal{L}^2(\mathbb{P}_Z)$. Then*

$$\Sigma_{\ddot{X}Y|Z} = 0 \iff X \perp\!\!\!\perp Y|Z. \quad (13)$$

*$\Sigma_{\ddot{X}Y|Z}$ can be replaced with $\Sigma_{\ddot{X}\ddot{Y}|Z}$, where $\ddot{Y} \triangleq (Y, Z)$.*

Therefore the statistic is defined as $J^{CI} \triangleq \left\| \Sigma_{\ddot{X}Y|Z} \right\|_{\mathcal{HS}}^2$. Actually, according to the discussion in Appendix E.4, we can interpret $\Sigma_{\ddot{X}Y|Z}$ as the partial covariance and obtain it using the cross-covariance between residuals of feature maps regressed on $Z$. That is, $\Sigma_{r_{\ddot{X}} r_Y} = \Sigma_{\ddot{X}Y|Z}$, where $r_X$ and $r_Y$ represent the residual of $\ddot{X}$ and $Y$ regressed on $Z$.

**Reweighted Conditional Dependence Measure**

With a fixed function $\beta(\cdot)$, at the population level, define the reweighted KCIT (RKCIT), $J_\beta^{CI} \triangleq \left\| \Sigma_{\ddot{X}Y|Z}^\beta \right\|_{\mathcal{HS}}^2$, as:

$$\left\| \mathbb{E}_{\tilde{\mathbb{P}}} \left[ (\psi_{\ddot{X}|Z}^\beta - \mathbb{E}_{\tilde{\mathbb{P}}}[\psi_{\ddot{X}|Z}^\beta]) \otimes (\phi_{Y|Z}^\beta - \mathbb{E}_{\tilde{\mathbb{P}}}[\phi_{Y|Z}^\beta]) \right] \right\|_{\mathcal{HS}}^2,$$

where $\psi_{\ddot{X}|Z}^\beta \triangleq \psi_{\ddot{X}} - \mathbb{E}_{\tilde{\mathbb{P}}}[\psi_{\ddot{X}}|Z], \phi_{Y|Z}^\beta \triangleq \phi_Y - \mathbb{E}_{\tilde{\mathbb{P}}}[\phi_Y|Z]$ are the residuals of original feature maps regressing on $Z$. Here $\tilde{\mathbb{P}} = \beta(X, Z)\mathbb{P}(X, Y, Z)$. Denote the centralized kernel matrices for $\ddot{X}, Y, Z$ as $\widetilde{K}_{\ddot{X}}$, $\widetilde{K}_Y$ and $\widetilde{K}_Z$, respectively. Given importance weights $\beta$ and the feature map $\psi(\cdot)$, the unbiased estimator of the centralization now becomes $\tilde{\psi}_u^\beta(x) = \psi(x) - \frac{1}{n} \sum_{i=1}^n \beta_i \psi(x_i) = \psi(x)(I_n - \frac{1}{n} D_\beta \mathbf{1}\mathbf{1}^T)$. Therefore we have the centralized kernel matrix $\widetilde{K}_Z^\beta = (I_n - \frac{1}{n}\mathbf{1}\mathbf{1}^T D_\beta) K_Z (I_n - \frac{1}{n} D_\beta \mathbf{1}\mathbf{1}^T)$. $\widetilde{K}_{\ddot{X}}^\beta$ and $\widetilde{K}_Y^\beta$ are defined similarly. We use kernel ridge regression to estimate conditional expectations and obtain the residual matrix $R_Z^\beta = \epsilon \left[ \widetilde{K}_Z^\beta D_\beta + \epsilon I \right]^{-1}$. The estimated

---

[2]Many popular kernels are characteristic, e.g., Gaussian, Laplace, etc. See the formal definition in Appendix E.1.

residuals become $R_Z^\beta \cdot \psi(\ddot{X})$ and $R_Z^\beta \cdot \phi(Y)$. An estimator of the statistic $J_\beta^{CI}$ calculated on $\mathcal{D} = \{(x_i, y_i, z_i)\}_{i=1}^n$ is:

$$\hat{J}_\beta^{CI} = \frac{1}{n^2} \text{Tr} \left[ \widetilde{K}_{\ddot{X}|Z}^\beta \widetilde{K}_{Y|Z}^\beta \right], \quad (14)$$

$\widetilde{K}_{\ddot{X}|Z}^\beta := R_Z^\beta \widetilde{K}_{\ddot{X}}^\beta R_Z^{\beta T} D_\beta$, $\widetilde{K}_{Y|Z}^\beta := R_Z^\beta \widetilde{K}_Y^\beta R_Z^{\beta T} D_\beta$. Therefore, for the CI test, the final objective function is similar to the optimization problem (9) with $\hat{J}_\beta^{UI}$ replaced by $\hat{J}_\beta^{CI}$. As in the unconditional case, in the experiment we use a normalized version of this conditional dependence measure with the denominators $\hat{J}_{2\beta}^{CI} = \text{Tr}[\widetilde{K}_{\ddot{X}|Z}^\beta]$ and $\hat{J}_{3\beta}^{CI} = \text{Tr}[\widetilde{K}_{Y|Z}^\beta]$.

Similar to independent tests, we also use a permutation test to approximate the $1 - \alpha$ quantile of the null distribution to test for CI. The difference is that the permutation should break the dependence between $Y$ and $X$ while maintaining the conditional distribution of $Y$ given $Z$. Therefore, we use a local permutation method on $Y$ that utilizes the nearest-neighbor search for $Z$ (Runge, 2018).

## 4. Application in Causal Discovery

In this section, we discuss how to discover causal relations when some rare dependence patterns exist. We propose a new constraint-based causal discovery method, named RD-PC (**R**are **D**ependence PC), which leverages the PC algorithm (Spirtes et al., 2000) equipped with our proposed RKCIT to test for CI. Here we assume that no latent confounders exist. With the Markov and faithfulness assumption, the $d$-separation in the ground-truth causal graph $G$ should theoretically exhibit an equivalent relation with the CI relations implied by the distribution. However, rare dependence is hard to detect, which would result in the deletion of some true edges and may even lead to erroneous propagation.

Here, we assume a known reference variable $C$. Specifically, for two variables $X$ and $Y$ from the node set $\mathbf{V}$ that exhibit rare dependence, their dependence is significant only within a region of $C$'s distribution. Here, $C$ could be a third variable other than $X$ and $Y$, providing additional information that enables RKCIT to recover dependence more effectively, thereby enhancing its flexibility.

With some abuse of notation, here we use $\text{KCIT}(X, Y|Z)$ to represent the result of testing $X \perp\!\!\!\perp Y|Z$ by KCIT. $\text{RKCIT}^{\beta(C)}(X, Y|Z)$ is defined similarly.

**Assumption 4.1.** $\forall X, Y \in \mathbf{V}, Z \subseteq \mathbf{V} \setminus \{X, Y\}$, if $\text{KCIT}(X, Y|Z)$ rejects the null hypothesis, then $X \not\perp\!\!\!\perp Y|Z$. Besides, if both $\text{KCIT}(X, Y|Z)$ and $\text{RKCIT}^{\beta(C)}(X, Y|Z)$ fails to reject the null hypothesis, then $X \perp\!\!\!\perp Y|Z$.

Assumption 4.1 ensures the reliability of the dependence detected by KCIT. In addition, it also ensures the reliability

of the independence detected by RKCIT. Then we derive the following rule to check the adjacency between two variables.

**Rule 1.** *For any $X, Y \in \mathbf{V}$, if $\exists Z \subseteq \mathbf{V} \backslash \{X, Y\}$ s.t. both $\text{KCIT}(X, Y|Z)$ and $\text{RKCIT}^{\beta(C)}(X, Y|Z)$ fail to reject the null hypothesis, then $X$ and $Y$ are not adjacent in $G$.*

On the other hand, although $\text{RKCIT}^{\beta(C)}(X, Y|Z)$ could discover more dependence, it introduces an extra variable $C$, which sometimes rejects the null hypothesis even when $X \perp\!\!\!\perp Y|Z$. We summarize the scenarios in the following:

**Proposition 4.2.** *For a pair of variables $X, Y \in V$, suppose that $\exists Z \subseteq V \backslash \{X, Y\}$ s.t. $\text{KCIT}(X, Y|Z)$ fails to reject the null hypothesis. Besides, for all these $Z$, we have that $\text{RKCIT}^{\beta(C)}(X, Y|Z)$ rejects the null hypothesis. Then, under Assumption 4.1, i) $X$ and $Y$ are adjacent with a rare dependence, or ii) $X$ and $Y$ are not adjacent in $G$ and $C$ must be the direct common effect of $X$ and $Y$.*

Proposition 4.2 tells us that if $X \rightarrow C \leftarrow Y$ forms a $V$-structure, Rule 1 can not correctly remove the edge between $X$ and $Y$ when executing the PC algorithm to recover the causal skeleton. Consequently, the inferred graph tends to be a superset of the true one. To recover the true causal skeleton and eliminate extraneous edges, we introduce our correction methods.

**Rule 2.** *For two variables $X, Y \in \mathbf{V}$ that satisfy the condition in Proposition 4.2, if there exists $Z \subseteq \mathbf{V} \backslash \{X, Y\}$, such that $\text{RKCIT}^{\beta(C^{perm})}(X, Y|Z)$ fail to reject the null hypothesis, then $X$ and $Y$ are not adjacent in $G$. Here $C^{perm}$ denotes the shuffled $C$ in dataset $\mathcal{D}$.*

Based on **Rule 1** and **Rule 2**, we summarize the procedure of RD-PC in Algorithm 2. Theorem 4.3 shows its soundness.

**Theorem 4.3.** *With Assumption 4.1, the causal Markov assumption and faithfulness assumption, Algorithm 2 correctly recovers the underlying causal graph structure up to its Markov equivalence class.*

We provide the proof and more discussion in Appendix F.

# 5. Experiments

We apply the proposed testing method to both synthetic and real data to evaluate their performance. Due to space limitation, causal discovery experiments see Appendix G.1. Codes are available at https://github.com/leeedwina430/RKCIT.

## 5.1. Simulation Experiments

For testing independence, we conduct experiments with varying numbers of samples and different levels of rare dependence in two generating settings. For testing CI, we further provide a comparison between various dimensions of the conditioning set. More details see Appendix G.

---

**Algorithm 2** Rare Dependence PC (RD-PC)

1: **Input:** $\mathcal{D}$: dataset. $\mathbf{V}$: node set. $C$: reference variable.
2: **Output:** causal graph $G$.
3: **Stage 1: Causal skeleton discovery.**
4: Initialize a complete undirected graph $G$ on $\mathbf{V}$.
5: Remove the edge connected to $C$ in $G$ by **Rule 1**.
6: For $X, Y \in \mathbf{V} \backslash \{C\}$, remove the edge $(X, Y)$ in $G$ by **Rule 1**. If both $X$ and $Y$ are not adjacent to $C$, using KCIT only is enough.
7: **Stage 2: Eliminating extraneous edges**.
   For $X, Y \in \mathbf{V} \backslash \{C\}$, if both $X$ and $Y$ are adjacent to $C$, check whether $(X, Y)$ are the extraneous edge. Shuffle data of $C$ in $\mathcal{D}$ as $C^{perm}$, if **Rule 2** is satisfied, remove the edge $(X, Y)$, and orient $X \rightarrow C$ and $Y \rightarrow C$.
8: **Stage 3: Determining the orientation**.
   Orient edges in $G$ with the same orientation procedure as the PC algorithm (Meek, 1995).

---

### 5.1.1. INDEPENDENCE TESTS

We use the following two generation process, which covers two common scenarios where rare dependency occurs.

**Data Generation I (DG I).** We slightly modify Example 1.1 with different variance $\sigma^2$ for $\epsilon \sim \mathcal{N}(0, \sigma^2)$ to evaluate power. To test Type I error, we set $Y = \epsilon$ with fixed $\sigma = 0.5$.

**Data Generation II (DG II).** We use a variant of the experiment in (Strobl et al., 2019). To evaluate the Type I error, we generate data that follows $X = f_1(\varepsilon_x), Y = f_2(\varepsilon_y)$, where $\varepsilon_x, \varepsilon_y$ are independently drawn from $\mathcal{N}(0, 1)$ and $f_1$ and $f_2$ are smooth functions chosen uniformly from a set (see Appendix G.1). To compare the power, we generate $X = f_1(\varepsilon_x) + \varepsilon_b$, and then we generate $Y = f_2(\varepsilon_y) + \varepsilon_b$ if $X < \tau$ where $\tau$ is a threshold, and $Y = f_2(\varepsilon_y)$ otherwise.

**Baselines.** All the baselines follow their default settings unless stated otherwise. **HSIC** (Gretton et al., 2007): the original HSIC test using gamma approximation. **RDC** (Lopez-Paz et al., 2013): use canonical correlation between a finite set of random Fourier features. **FHSIC** (Zhang et al., 2018): HSIC using finite-dimensional random Fourier feature mappings to approximate kernels. **FisherScan** (Ma & Mao, 2019): Generalized Fisher's exact test on contingency tables and continuous sample spaces. **LFHSIC** (Ren et al., 2024): HSIC test with adaptively learned bandwidth. For all the other methods in our paper, including ours, use Gaussian kernel with median heuristic bandwidth.

**Results.** In Fig. 4, we demonstrate that our method consistently controls Type I errors and that its power outperforms other baselines in both Data Generation I and II. For DG I, our method performs well because we not only try to find the sub-samples but also reweight them with a weight possibly larger than 1, effectively enlarging the size of the

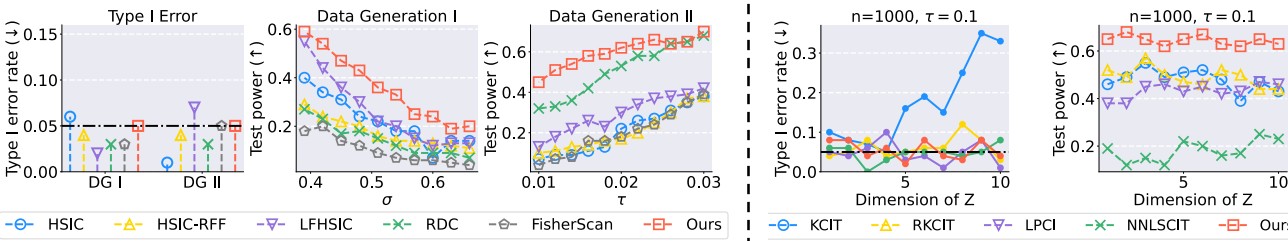

*Figure 4.* **Left**: Type I error rate and power of independence tests on DG I and DG II with 1000 samples. The first graph shows the Type I error rate for 6 methods in both data generations, where the significance level 0.05 is annotated as the black line. The second and third graphs show the test power for DG I and II with different parameters respectively. **Right**: Type I error rate and power of CI tests with different sizes of conditional set $Z$. We fix the sample size $n = 1000$ and the ratio of the rare dependence $\tau = 0.1$.

dependent sub-sample. For DG II, the advantage of our method becomes more obvious when the data become more imbalanced, i.e., $\tau \to 0$. For visualization of the important sub-samples found in example data, see Appendix G.

On the left of Fig. 5 compares the Type I error and power of the methods with different sample sizes $n$, where $n \in \{500, 1000, 1500, 2000, 2500, 3000\}$. We fix $\sigma = 0.5$ and $\tau = 0.01$ for DG I and DG II respectively. Most methods including ours succeed in controlling Type I error rate around 0.05, while LFHSIC and HSIC have a relatively unstable Type I error rate. Our method again consistently performs better in testing power, which confirms the need for sample-level importance reweighting when facing rare dependence. We note that the performance of the baselines also improves as $n$ increases, which is expected since the range of the significant dependent region is fixed. Note that HSIC almost always overlaps with LFHSIC for DG II, manifesting that optimizing the bandwidth of kernels is not enough to detect rare dependence. RDC performs well in DG I while it struggles in DG II, further showing the genericity of our method.

### 5.1.2. CI TESTS

**Data Generation.** We follow the synthetic experiment proposed in (Scetbon et al., 2022) with a slight variation. To compare the Type I error, we generate simulated data by:

$$X = f_1(\bar{Z} + \varepsilon_x), Y = f_2(\bar{Z} + \varepsilon_y)$$

Above, $\bar{Z}$ is the average of $Z = (Z_1, \cdots, Z_{d_z})$, $\varepsilon_x$ and $\varepsilon_y$ are sampled independently from $\mathcal{N}(0, 1)$, and $f_1$ and $f_2$ are smooth functions chosen from the same set as in DG II The following generating function is for evaluating power:

$$\begin{cases} X = f_1(\bar{Z} + \varepsilon_x) + \varepsilon_b, Y = f_2(\bar{Z} + \varepsilon_y) + \varepsilon_b, & \text{if } X < \tau, \\ X = f_1(\bar{Z} + \varepsilon_x) + \varepsilon_b, Y = f_2(\bar{Z} + \varepsilon_y), & \text{if } X \geq \tau. \end{cases}$$

where $Q \sim U(0, 1)$, $\varepsilon_b \sim \mathcal{N}(0, 1)$, $\tau \in [0, 1]$ is a threshold and we set $\tau$ equals to the $\tau$-th percentile of $X$.

**Baselines.** We compare with the following CI methods:

KCIT (Zhang et al., 2012), RCIT (Strobl et al., 2019), CCIT (Sen et al., 2017), GCIT (Bellot & van der Schaar, 2019), FCIT (Chalupka et al., 2018), GCM (Shah & Peters, 2020), and NNLSCIT (Li et al., 2024b). Results of some baselines are shown in Appendix G for beauty.

**Results.** In the right of Fig. 4, we fix $\tau = 0.03$ and compare the performance of different methods with various dimensions of the conditional set $Z$. Most methods except KCIT maintain Type I error around 0.05. This behaviour may be attributed to the error caused by gamma distribution approximating the null distribution. On average, our method significantly outperforms other methods in testing power and achieves a power 50% better than KCIT. All the baselines here are unsensitive to the increasing of the dimension of the conditioning variable.

We also enumerate $\tau \in (0.01, 0.03)$ to generate samples with different ratios of the sub-samples that are significantly dependent, as shown in Fig. 5. We use this setting to model different levels of rare dependence. The dimension of the conditioning variable $Z$ is fixed at 10 and the number of samples $n = 1000$. Our method achieves a better performance than the baselines, especially compared to KCIT.

### 5.2. Real-world Experiments

**Sachs Dataset.** We apply our RHSIC to a flow cytometry dataset (Sachs et al., 2005), which gives $n = 853$ observational measurements of 11 proteins. Here we focus on the dependence relationship between (PKA, PJINK). This pair is dependent according to the ground truth causal graph in Fig. 11. However, as the results discussed in (Mooij & Heskes, 2013), many popular (conditional) independence tests cannot detect this dependence relation. We compared our RHSIC with HSIC since theoretically these two pairs are dependent conditioning on an empty set. The $p$-value produced by HSIC with default settings is 0.601, while our method successfully detected the dependence with $p$-value $= 0.004$. See Appendix G.3 for more details.

**Financial Dataset.** We also apply our method to monthly

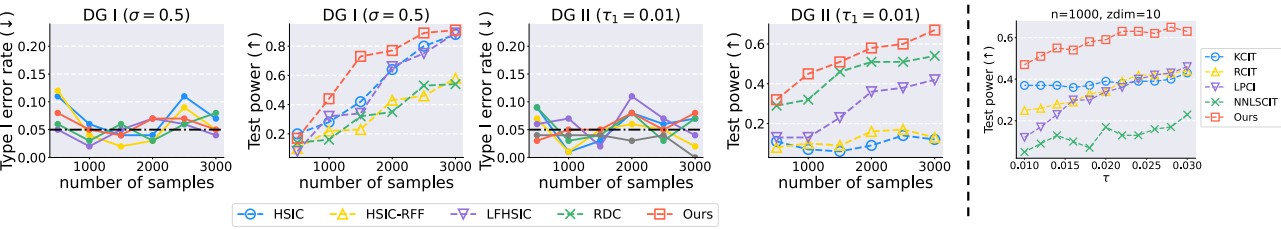

*Figure 5.* **Left**: Type I error rate and Power of DG I (first and second figure) and DG II (third and fourth figure) with different sample sizes. The first row represents Type I error rates and the second row shows test power. **Right**: Testing powers of CI tests with various ratios of significantly dependent sub-samples. We fix the number of samples $n = 1000$ and the size of conditional set $Z$ as 10.

JPY/USD exchange rates (E) and U.S. federal funds rates (F) from 1990 to 2010, sourced from Federal Reserve Economic Data (FRED). Each month records a datapoint, resulting in 251 samples in total. While the original HSIC fails to reject independence with $p$-value equals to $0.2174$, RHSIC detects dependence with $p = 0.0005$ using F as the reference variable. The learned weights assign higher importance to the samples in 2001 and 2008. These correspond to the Dot-com recession and the global financial crisis, respectively — showing that our method not only detects rare dependence but also provides interpretable insights.

## 6. Discussion

Some previous works fall into the subset of our setting. A line of research focusing on "local dependence" (Abberger, 2002; Üçer & Bayramoğlu, 2007; Sricharan et al., 2011; Tjøstheim et al., 2022; Gorsky & Ma, 2022) actually helps quantify and visualize a fine-grained dependence relationship while lacking a formal testing procedure or being restricted to a hard "local region". Context-Specific Independence (CSI) (Pensar et al., 2016; Hwang et al., 2023; Poole & Zhang, 2003; Boutilier et al., 2013; Pensar et al., 2015; Hwang et al., 2024) aims to find a more fine-grained independence relation, which is conceptually related to our objective, though approached from the perspective of independence rather than dependence. The complementarity actually enables our methods to handle such cases, as discussed in the next paragraph. (Wang, 2020; Wang et al., 2021) defined a term "rare events" to describe extremely imbalanced data for parameter estimation. The data-generating process proposed is a specific instantiation of our setting similar to Example 3.1 and thus solvable by our tests.

The importance weights learned by our method naturally highlight subgroups of data that contribute most to capturing the underlying dependence signal. As demonstrated in our real-world dataset analysis, these weights enhance interpretability for downstream applications. Notably, the algorithm in Sec. 4 is capable of uncovering fine-grained causal structures by highlighting context-dependent relationships. This, in turn, facilitates the identification of context-specific

biomarkers for targeted interventions in precision medicine and supports the inference of group-specific causal effects. Additionally, fine-grained causal structures support context-aware fairness by mitigating biases arising from majority groups. As discussed earlier, our method is well-suited to handle context-specific or local independence scenarios, due to the complementary nature of its objectives. For example, Example 3.1 and Data Generation II & III in our experiments involve local dependence, and our method performs well in these cases, as the low (near-zero) weight samples often help to identify locally independent regions.

## 7. Conclusion

We focus on the problem of testing conditional independence in the presence of rare dependence, where the dependence is significant only in small regions of the sample. We propose a reweighted (conditional) dependence measure, which adaptively reweights samples to enhance the ability to detect subtle dependence structures. We provide an asymptotic analysis of the reweighted statistics and a consistency analysis of our approach to ensure its theoretical soundness. The theoretical results are applied to the causal discovery task, demonstrating their applicability in structure learning in the presence of rare dependence. Extensive experiments on synthetic and real-world datasets validated the effectiveness of our method, highlighting its potential for broad applications in data analysis. Future work can explore extending our reweighting methods to other tasks.

## Acknowledgements

The authors would like to thank the anonymous reviewers for their helpful comments. We would also like to acknowledge the support from NSF Award No. 2229881, AI Institute for Societal Decision Making (AI-SDM), the National Institutes of Health (NIH) under Contract R01HL159805, and grants from Quris AI, Florin Court Capital, and MBZUAI-WIS Joint Program. MG was supported by ARC DE210101624, ARC DP240102088, and WIS-MBZUAI 142571. XW acknowledges the support from National Natural Science Foundation of China (Grants No. 12171076).

## Impact Statement

This paper presents work whose goal is to advance the field of Machine Learning and Causality. There are many potential societal consequences of our work, none which we feel must be specifically highlighted here.

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

**Organization of Appendices**

# A. Notations

*Table 1.* Notation Table

| Symbol | Description |
|---|---|
| $X, Y, Z$ | Random variables (or sets of variables) |
| $\mathcal{X}, \mathcal{Y}, \mathcal{Z}$ | Domains for random variables |
| $\mathcal{F}_X, \mathcal{F}_Y, \mathcal{F}_Z$ | Reproducing kernel Hilbert spaces (RKHS) |
| $x, y, z$ | Sample vectors (or matrices) |
| $x_i, y_i, z_i$ | Specific values of sample vectors (or matrices) |
| $k_X(x, x'), k_Y(y, y'), k_Z(z, z')$ | Kernel functions on the input spaces $\mathcal{X}, \mathcal{Y}, \mathcal{Z}$ |
| $\psi(\cdot), \phi(\cdot), \varphi(\cdot)$ | Feature maps for $X, Y, Z$ |
| $\boldsymbol{K}_X, \boldsymbol{K}_Y, \boldsymbol{K}_Z$ | Kernel matrice on samples $x, y, z$ |
| $\Sigma_{XY}$ | Cross-Covariance operator |
| $\|\cdot\|_{\mathcal{F}}$ | Norm in a RKHS |
| $\mathbb{E}[X]$ | Expectation of $X$ |
| $\mathbb{V}\text{ar}[X]$ | Variance of $X$ |
| $\mathbb{C}\text{ov}[X]$ | Covariance of $X$ and $Y$ |
| $\mathbb{R}^{\geq 0}$ | The set of positive real numbers (including 0) |
| $\mathcal{B}(\mathbb{R})$ | Borel $\sigma$-algebra on $\mathbb{R}$ |
| $\mathbb{P}_{XY}$ | Joint distribution of $X$ and $Y$ |
| $\mathbb{P}_{XY|Z}$ | Joint distribution of $X$ and $Y$ conditioned on $Z$ |
| $\text{Tr}[\cdot]$ | The trace of a matrix |
| $\otimes$ | Tensor product |
| $\mathcal{O}$ | Big O notion |
| $n$ | Number of samples |
| $X \perp\!\!\!\perp Y$ | $X$ is independent of $Y$ |
| $(i)_r^n$ | The set of all $r$-tuples drawn without replacement |
| $(n)_k$ | Number of permutations |
| $\mathcal{N}(\Omega, \epsilon, \|\cdot\|_\infty)$ | Covering number with radii $\epsilon$ for $\Omega$ equipped with infinite norm |
| $\mathcal{N}(0, 1)$ | Normal distribution with zero mean and standard deviation 1 |
| $U(0, 1)$ | Uniform distribution in $(0, 1)$ |

# B. Related Works

In this section, we provide a more comprehensive review of literature.

## B.1. Independence Tests

Traditional independent tests for discrete/categorical data include $\mathcal{F}$-test (Tiku, 1967) and Chi-squared test (Greenwood & Nikulin, 1996). Testing independence for continuous variables is more challenging. Pearson correlation coefficient (Benesty et al., 2009) is often used to measure the correlation between variables. However, it only reflects linear dependence. In order to measure broader dependence, a class of kernel-based independence testing (Bach & Jordan, 2002; Gretton et al., 2005b; 2003; 2005a) was proposed. These methods are mainly based on the framework proposed by Rényi (Rényi, 1959) to measure the nonlinear dependence of variables by sufficiently adequate mappings under function classes. Under the reproducing kernel Hilbert space (RKHS) (Berlinet & Thomas-Agnan, 2011) space, the kernel function is defined as a distance metric induced by the inner product of the feature mapping. One widely-used kernel-based independence test is the Hilbert Schmidt Independence Criterion (HSIC) (Gretton et al., 2005a), which measures dependence by the squared *Hilbert-Schmidt norm* induced by the cross-covariance operators in the RKHS space. Besides, the random dependence coefficient (Lopez-Paz et al., 2013) (RDC) was proposed. Compared to the kernel-based method, RDC is computationally efficient and easily implemented. This method ensures the marginal invariance by using copula transformation, and measures the dependence between variables by maximizing the correlation under random projection of copula transformation.

## B.2. CI Tests

Conditional independent tests are generally more difficult than independent tests due to the hardness of estimating the conditional density distribution compared to the marginal distribution. A class of metric-based CI test (Su & White, 2007) employs a number of kernel smoothers to estimate conditional characteristic functions. This type of kernel smoothing estimation has a large computational cost when the condition set is high-dimensional. Another widely used method is kernel-based conditional independence testing such as KCIT (Zhang et al., 2012). KCIT is based on the partial association framework proposed by Daudin (Daudin, 1980) and uses conditional cross-correlation operators to identify conditional independence. Later, the approximate kernel-based method RCIT (Strobl et al., 2019) was proposed, which uses random Fourier features to approximate the Gaussian kernel, resulting in an improvement in the computational efficiency of KCIT. Another class of methods (Doran et al., 2014; Bellot & van der Schaar, 2019; Shi et al., 2021; Runge, 2018; Sen et al., 2017; Mukherjee et al., 2020) obtains the distribution of the statistic under the null hypothesis by estimating the conditional density function or conditional mutual information, followed by a hypothesis test to determine conditional independence. The permutation-based method (Doran et al., 2014) obtains resampled samples of factorized distribution by performing permutations on the samples that satisfy a specific structure. Some other methods (Bellot & van der Schaar, 2019; Shi et al., 2021), use generative models to estimate the conditional density. The method (Runge, 2018; Mukherjee et al., 2020) uses k-nearest-neighbor (KNN) to obtain a factorized distribution or estimate Kullback–Leibler (KL) divergence by classification to estimate conditional mutual information for judging conditional independence.

## B.3. Context-specific Independence or Local (In-)Dependence

The methods for measuring local (in-)dependence (Pensar et al., 2016; Tjøstheim et al., 2022; Hwang et al., 2023) help visualize the fine-grained dependence relationship while lack a formal testing procedure. Context-Specific Independence (CSI) (Poole & Zhang, 2003; Boutilier et al., 2013; Poole, 2013; Pensar et al., 2015; Hwang et al., 2024) aims to find a more fine-grained independence relation, which is conceptually related to our objective, though approached from the perspective of independence rather than dependence. The methods are mainly to find the specific contexts (subspaces of the variable domain) that conditional independence relationships hold. It is a widely used local independence relationships. To encode CSI relationships in graphical models, (Pensar et al., 2015) introduced Labeled Directed Acyclic Graphs (LDAGs). Later, (Pensar et al., 2016) introduced partial conditional independence (PCI), which is a generalization of CSI. Several recent efforts have aimed to identify local causal structures for continuous variables. However, CSI aims to find independence while we focus on dependence. And CSI has a hard threshold for independent and dependent samples, making it a more restricted scenario compared to rare dependence in our paper (Pensar et al., 2015).

## B.4. Causal Discovery

Traditional causal discovery methods include constraint-based and score-based methods. Constraint-based methods such as PC (Spirtes et al., 2000) first use CI tests to obtain the causal skeleton, then employ $V$-structures and consistent propagation to infer the directions. Score-based methods such as GES (Chickering, 2002) search for the best-scored candidate graph with a pre-defined score function (*e.g.*, BIC/MDL score (Chickering & Heckerman, 1997), BGe score (Geiger & Heckerman, 1994) for linear-Gaussian models, the BDeu/BDe (Buntine, 1991; Heckerman et al., 1995) score for discrete data. When the number of variables is large, the score-based methods usually employ a heuristic strategy to speed up the search process. Recently, NOTEARS (Zheng et al., 2018) innovatively proposes a differentiable characterization of acyclicity, which converts score-based causal discovery from a combinatorial optimization problem to a continuous optimization problem. Following that, some variants (Yu et al., 2019; Ng et al., 2020; Wei et al., 2020; Zheng et al., 2020) were proposed to extend NOTEARS into more scenarios. Without further assumptions, we can only recover the underlying causal graph up to is Markov equivalence class. To make it fully identifiable, a widely used class of methods (Shimizu, 2014; Hoyer et al., 2008; Zhang & Hyvarinen, 2012) assumes that the data generation process satisfies a specific functional model. These function causal model-based causal discovery methods can use asymmetries in statistical dependencies to determine the direction between variables.

# C. Reweighting Functions and the Reweighted Distribution

## C.1. Proof of Proposition 3.3

**Proposition 3.3** (Maintain the independence after reweighting). *If $X$ and $Y$ are independent and $C$ is either $X$ or $Y$ but not both, then $X$ and $Y$ are still independent in the reweighted distribution of $(X, Y)$ with weight $\beta(C)$.*

*Proof.* First we consider the case $C = X$. Define $\tilde{\mathbb{P}}_X \triangleq \beta(X) \cdot \mathbb{P}_X$. Since $X \perp\!\!\!\perp Y$, $\mathbb{P}_X \cdot \mathbb{P}_{Y|X} = \mathbb{P}_X \mathbb{P}_Y$.

$$\tilde{\mathbb{P}}_{XY} = \beta(X) \cdot \mathbb{P}_X \cdot \mathbb{P}_{Y|X} = \beta(X)\mathbb{P}_X\mathbb{P}_Y = \tilde{\mathbb{P}}_X\mathbb{P}_Y. \tag{15}$$

And in the other case, $C = Y$. Define this time $\tilde{\mathbb{P}}_Y \triangleq \beta(Y) \cdot \mathbb{P}_Y$. Similarly we have

$$\tilde{\mathbb{P}}_{XY} = \beta(Y) \cdot \mathbb{P}_X \cdot \mathbb{P}_{Y|X} = \beta(Y)\mathbb{P}_X\mathbb{P}_Y = \mathbb{P}_X\tilde{\mathbb{P}}_Y, \tag{16}$$

That is, $\tilde{\mathbb{P}}_{XY}$ is factorable in terms of $X$ and $Y$, meaning that $X$ and $Y$ are still independent according to $\tilde{\mathbb{P}}_{XY}$. □

## C.2. Proof of Proposition 3.9

**Proposition 3.9** (Maintain the conditional independence after reweighting). *If $X$ and $Y$ are conditionally independent given $Z$ and that $C$ is either $X$ or $Y$ but not both, then $X$ and $Y$ are still conditionally independent given $Z$ after reweighted by the weight $\beta(C, Z)$.*

*Proof.* Similar to the unconditional independence, we first set $C = X$ and define $\tilde{\mathbb{P}}_{X|Z} \triangleq \beta(X, Z) \cdot \mathbb{P}_{X|Z}$, we have

$$\tilde{\mathbb{P}}_{XY|Z} = \beta(X, Z) \cdot \mathbb{P}_{X|Z} \cdot \mathbb{P}_{Y|XZ} = \beta(X, Z) \cdot \mathbb{P}_{X|Z} \cdot \mathbb{P}_{Y|Z} = \tilde{\mathbb{P}}_{X|Z}\mathbb{P}_{Y|Z}, \tag{17}$$

since $X \perp\!\!\!\perp Y|Z$ gives us $\mathbb{P}_{X|Z} \cdot \mathbb{P}_{Y|XZ} = \mathbb{P}_{X|Z} \cdot \mathbb{P}_{Y|Z}$. Next, set $C = Y$ and define $\tilde{\mathbb{P}}_{Y|Z} \triangleq \beta(Y, Z) \cdot \mathbb{P}_{Y|Z}$, we have

$$\tilde{\mathbb{P}}_{XY|Z} = \beta(Y, Z) \cdot \mathbb{P}_{X|Z} \cdot \mathbb{P}_{Y|XZ} = \beta(Y, Z) \cdot \mathbb{P}_{X|Z} \cdot \mathbb{P}_{Y|Z} = \tilde{\mathbb{P}}_{X|Z}\tilde{\mathbb{P}}_{Y|Z}. \tag{18}$$

That is, $\tilde{\mathbb{P}}_{XY|Z}$ is factorable in terms of $(X, Z)$ and $(Y, Z)$, meaning that $X$ and $Y$ are still conditionally independent given $Z$ according to $\tilde{\mathbb{P}}_{XY|Z}$. □

## C.3. When $C$ is a Third Variable

We first state a more general case of the definition of the reweighting function and reweighted distribution.

**Definition C.1** (Reweighting function and reweighted distribution). $C$ is either a subset of $X$ or $Y$, or it represents another variable tha is independent of $(X, Y)$. Let $\mathcal{C}$ and $\mathcal{Z}$ be the domain of $C$ and $Z$. The reweighting function set $\mathcal{B}$ satisfies

$$\mathcal{B} = \left\{ \beta : \mathcal{C} \times \mathcal{Z} \to \mathbb{R}^+ \mid \mathbb{E}_{\mathbb{P}_{XY|Z}}[\beta(C, Z)] = 1 \right\}.$$

Then $\forall \beta \in \mathcal{B}$, the corresponding reweighted distribution $\tilde{\mathbb{P}}_{XYC|Z}$, which is well defined with the same support of $\mathbb{P}_{XYC|Z}$, can be determined by the conditional probability density function

$$\tilde{\mathbb{P}}(X, Y, C \mid Z) = \beta(C, Z)\mathbb{P}(X, Y, C \mid Z). \tag{19}$$

The reweighting function and the reweighted distribution in the unconditional case are a special case of this definition where the conditioning set $Z$ is empty. In fact, the results of Proposition 3.3 can be generalized to the case when $C$ is neither $X$ nor $Y$, which is useful in the causal discovery settings in our paper. In general, the following proposition imposes the constraint that $C$ cannot be a common effect of $X$ and $Y$.

**Proposition C.2** (Extention of Proposition 3.3). *If $X$ and $Y$ are independent and that $C$ is a subset of $X$ or $Y$, **or $C$ is a variable that satisfies** $\beta(C) = \beta_X(\tilde{X})\beta_Y(\tilde{Y})$, where $\tilde{X}$ and $\tilde{Y}$ are subsets of $X$ and $Y$, then $X$ and $Y$ are still independent in the importance reweighed distribution of $(X, Y)$ with weight $\beta(C)$.*

The results in Proposition 3.9 can also be generalized to the case where $C$ is a third variable that is neither $X$ nor $Y$.

**Proposition C.3** (Extention of Proposition 3.9). *Consider that $X$ and $Y$ are independent given $\mathbf{Z}$ and $\ddot{C} = \{C, Z\}$ where $C$ is a subset of $X$ or $Y$ or $\ddot{C}$ **is a variable that satisfies** $\beta(\ddot{C}) = \beta_{XZ}(\tilde{X}, Z)\beta_{YZ}(\tilde{Y}, Z)$, where $\tilde{X}$ and $\tilde{Y}$ are subsets of $X$ and $Y$. Then, $X$ and $Y$ are still independent given $Z$ in the importance reweighed distribution of $(X, Y, Z)$ with weight $\beta(\ddot{C})$.*

Note that when $C = Z$ it reduces to the original conditional distribution ($\beta_k \equiv 1$).

### C.4. Proof of Proposition 3.6

**Proposition 3.6** *Suppose the sample data are i.i.d. and randomly split into $\mathcal{D}_{te} = \{x_{te}, y_{te}\}$ and $\mathcal{D}_{tr}$ correspondingly. $\hat{\beta}(\cdot)$ maximizes the dependence measure of $\mathcal{D}_{tr}$. Then $\hat{\beta}(\cdot)$ is independent of $\mathcal{D}_{te}$.*

*Proof.* Since the sample data are i.i.d., $\forall i \neq j, (x_i, y_i) \perp\!\!\!\perp (x_j, y_j)$. By randomly splitting the entire dataset, we have $\mathcal{D}_{te} \perp\!\!\!\perp \mathcal{D}_{tr}$. We can denote $\hat{\beta}(\cdot) = f(\mathcal{D}_{tr})$, where $f$ is a measurable function of $\mathcal{D}_{tr}$. By a standard result in probability (Casella & Berger, 2002, Theorem 4.3.5 and 4.6.12), if $X$ and $Y$ are independent, then any measurable function of $X$ is also independent of $Y$. Applying this fact here gives $\hat{\beta}(\cdot) \perp\!\!\!\perp \mathcal{D}_{te}$. $\square$

## D. Details about the Unconditional Independence Statistic

We first give some preliminaries for later proof and derivation.

**Definition D.1** ($U$-statistics). The statistic $U_n$ defined as follows is called a $U$-statistic with symmetric function $h$ of order $m$:

$$U_n = \binom{n}{m}^{-1} \sum_c h\left(X_{i_1}, \ldots, X_{i_m}\right), \tag{20}$$

where $\sum_c$ denotes the summation over the $\binom{n}{m}$ combinations of $m$ distinct elements $\{i_1, \ldots, i_m\}$ from $\{1, \ldots, n\}$.

For every $U$-statistic $U_n$ as an estimator of $\vartheta = E\left[h\left(X_1, \ldots, X_m\right)\right]$, there is a closely related $V$-statistic defined by

$$V_n = \frac{1}{n^m} \sum_{i_1=1}^n \cdots \sum_{i_m=1}^n h\left(X_{i_1}, \ldots, X_{i_m}\right).$$

**Proposition D.2.** *Let $V_n$ be defined by the above function and we have $n$ i.i.d. samples $\{x_i\}_{i=1}^n$ drawn from $\mathbb{P}_X$.*

*(i) Assume that $\mathbb{E}[|h(X_{i_1}, \ldots, X_{i_m})|] < \infty$ for all $1 \le i_1 \le \cdots \le i_m \le m$. Then the bias of $V_n$ satisfies*

$$b_{V_n}(\mathbb{P}_X) = O\left(n^{-1}\right).$$

*(ii) Assume that $\mathbb{E}\left[h(X_{i_1}, \ldots, X_{i_m})^2\right] < \infty$ for all $1 \le i_1 \le \cdots \le i_m \le m$. Then the variance of $V_n$ satisfies*

$$\mathbb{V}\mathrm{ar}(V_n) = \mathbb{V}\mathrm{ar}(U_n) + O(n^{-2}).$$

We also define some more statistics for later derivation. For $k = 1, \ldots, m$, let

$$\begin{aligned}
h_k(x_1, \ldots, x_k) &= \mathbb{E}\left[h(X_1, \ldots, X_m) \mid X_1 = x_1, \ldots, X_k = x_k\right] \\
&= \mathbb{E}\left[h(x_1, \ldots, x_k, X_{k+1}, \ldots, X_m)\right].
\end{aligned}$$

Note that $h_m = h$. Further define $\zeta_k \triangleq \mathbb{V}\mathrm{ar}(h_k(X_1, \ldots, X_k))$.

**Theorem D.3** ((Shao, 2008), Theorem 3.16). *Let $V_n$ be a V-statistics with $\mathbb{E}\left[h(X_{i_1}, \ldots, X_{i_m})^2\right] < \infty$ for all $1 \le i_1 \le \cdots \le i_m \le m$.*

*(i) If $\zeta_1 \triangleq \mathbb{V}\mathrm{ar}(h_1(X_1)) > 0$, then*

$$\sqrt{n}\left(V_n - \vartheta\right) \xrightarrow{d} N(0, m^2 \zeta_1).$$

*(ii) If $\zeta_1 = 0$ but $\zeta_2 \triangleq \mathbb{V}\mathrm{ar}(h_2(X_1, X_2)) > 0$, then*

$$n\left(V_n - \vartheta\right) \xrightarrow{d} \frac{m(m-1)}{2} \sum_{j=1}^{\infty} \lambda_j \chi^2_{1j},$$

*where $\chi^2_{1j}$'s are i.i.d. random variables having the chi-square distribution $\chi^2_1$ and $\lambda_j$'s are some constants (which may depend on $\mathbb{P}_X$) satisfying $\sum_{j=1}^{\infty} \lambda_j^2 = \zeta_2$.*

## D.1. Characterization of Unconditional Independence

For $(X, Y) \in \mathcal{X} \times \mathcal{Y}$, the cross-covariance operator $\Sigma_{XY} : \mathcal{F}_Y \to \mathcal{F}_X$ is defined by (Fukumizu et al., 2004):

$$\forall f \in \mathcal{F}_X, g \in \mathcal{F}_Y, \ \langle f, \Sigma_{XY} g \rangle_{\mathcal{F}_X} = \mathbb{E}_{XY}[f(X)g(Y)] - \mathbb{E}_X[f(X)]\mathbb{E}_Y[g(Y)]. \tag{21}$$

and the covariance operator itself can be written as

$$\Sigma_{XY} := \mathbb{E}_{XY}\left[(\psi(X) - \mu_X) \otimes (\phi(Y) - \mu_Y)\right], \ \mu_X \triangleq \mathbb{E}_X \psi(X), \mu_Y \triangleq \mathbb{E}_Y \phi(Y), \tag{22}$$

where $\otimes$ is the tensor product. This operator is a generalization of the cross-covariance matrix between random vectors. HSIC is the squared Hilbert-Schmidt norm (the sum of the squared singular values) of this operator, as mentioned in Def. 2.1

$$\begin{aligned}
\mathrm{HSIC}(X, Y) = {} &\mathbb{E}_{XX'YY'}[k_X(X, X')k_Y(Y, Y')] + \mathbb{E}_{XX'}[k_X(X, X')]\mathbb{E}_{YY'}[k_Y(Y, Y')] \\
&- 2\mathbb{E}_{XY}\left[\mathbb{E}_{X'}[k_X(X, X')]\mathbb{E}_{Y'}[k_Y(Y, Y')]\right].
\end{aligned} \tag{23}$$

Assuming the expectations exist, where $X'$ denotes an independent copy of $X$. An unbiased estimator of HSIC in sample $\mathcal{D} = \{(x_i, y_i)\}_{i=1}^n$ drawn from distribution $\mathbb{P}_{XY}$ is the sum of three $U$-statistics: (Gretton et al., 2007)

$$\mathrm{HSIC}_u(\mathcal{D}) = \frac{1}{(n)_2} \sum_{(i,j) \in \mathbf{i}_2^n} k_X^{ij} k_Y^{ij} + \frac{1}{(n)_4} \sum_{(i,j,q,r) \in \mathbf{i}_4^n} k_X^{ij} k_Y^{qr} - 2 \frac{1}{(n)_3} \sum_{(i,j,q) \in \mathbf{i}_3^n} k_X^{ij} k_Y^{iq}, \tag{24}$$

where $k_X^{ij} := k_X(x_i, x_j)$, $k_Y^{ij} := k_Y(y_i, y_j)$, $(n)_m := \frac{n!}{(n-m)!}$, and the index set $\mathbf{i}_r^n$ denotes the set all $r$-tuples drawn without replacement from the set $\{1, \ldots, n\}$. A biased estimator is the one replacing $U$-statistics with $V$-statistics, as in

$$\mathrm{HSIC}_b(\mathcal{D}) = \frac{1}{n^2} \sum_{i,j}^n k_X^{ij} k_Y^{ij} + \frac{1}{n^4} \sum_{i,j,q,r}^n k_X^{ij} k_Y^{qr} - 2 \frac{1}{n^3} \sum_{i,j,q}^n k_X^{ij} k_Y^{iq} = \frac{1}{n^2} \mathrm{Tr}(\boldsymbol{K}_X \boldsymbol{H} \boldsymbol{K}_Y \boldsymbol{H}), \tag{25}$$

where the summation indices now denote all $r$-tuples drawn with replacement from $\{1, \ldots, n\}$ and $\boldsymbol{H} = \boldsymbol{I} - \frac{1}{n}\mathbf{1}\mathbf{1}^\top$.

## D.2. Derivation of the Importance Reweighted Statistics for UI Test

We first consider the statistic for the reweighted distribution on the population level. Given a known reweighting function $\beta(\cdot)$, we use $\beta$ to represent $\beta(x)$, $\mu_X^\beta = \mathbb{E}_{\tilde{\mathbb{P}}_X}[\psi_X] = \mathbb{E}_{\mathbb{P}_X}[\beta \cdot \psi_X]$, and $\mu_Y^\beta = \mathbb{E}_{\tilde{\mathbb{P}}_Y}[\phi_Y] = \mathbb{E}_{\mathbb{P}_{XY}}[\beta \cdot \phi_Y]$. Then

$$
\mathrm{HSIC}^\beta(X,Y) = \left\| \Sigma_{XY}^\beta \right\|_{\mathcal{HS}}^2 = \left\| \mathbb{E}_{\tilde{\mathbb{P}}} \left[ (\psi_X - \mu_X^\beta) \otimes (\phi_Y - \mu_Y^\beta) \right] \right\|_{\mathcal{HS}}^2 = \left\| \mathbb{E}_{\mathbb{P}} \left[ \beta(\psi_X - \mu_X^\beta) \otimes (\phi_Y - \mu_Y^\beta) \right] \right\|_{\mathcal{HS}}^2 .
$$

Suppose now we have data samples $\mathcal{D}$ and obtain its estimator. Let $\psi_i(x)$ be the $i$th dimension of $\psi(x)$ and set $\beta_k \triangleq \beta(x_k)$. Define $\boldsymbol{H}_\beta \triangleq \boldsymbol{D}_\beta \left( \boldsymbol{I} - \frac{1}{n}\mathbf{1}\mathbf{1}^\top \boldsymbol{D}_\beta \right)$ and $\boldsymbol{D}_\beta \triangleq \mathrm{diag}\left( \beta_1, \ldots, \beta_n \right)$, that is, the diagonal matrix with $\beta_i$ on its diagonal. We know the estimator of $\left\| \Sigma_{XY}^\beta \right\|_{\mathcal{HS}}^2$ is $\left\| \widehat{\Sigma}_{XY}^\beta \right\|_{\mathcal{HS}}^2 = \left\| \widehat{\mathbb{E}}_{\tilde{\mathbb{P}}} \left[ (\psi_X - \hat{\mu}_X^\beta) \otimes (\phi_Y - \hat{\mu}_Y^\beta) \right] \right\|_{\mathcal{HS}}^2$. Then on the sample level:

$$
\begin{aligned}
\mathrm{HSIC}_b^\beta(\mathcal{D}) &= \left\| \widehat{\Sigma}_{XY}^\beta \right\|_{\mathcal{HS}}^2 = \left\| \frac{1}{n} \sum_{k=1}^n \left[ \beta_k \left( \psi(x_k) - \frac{1}{n}\sum_{p=1}^n \beta_p \phi(x_p) \right) \right) \otimes \left( \phi(y_k) - \frac{1}{n}\sum_{q=1}^n \beta_q \phi(y_q) \right) \right] \right\|_{\mathcal{HS}}^2 \\
&= \left\| \frac{1}{n} \sum_{k=1}^n \beta_k \left( \psi(x_k) - \frac{1}{n}\sum_{q=1}^n \beta_q \psi(x_q) \right) \left( \phi(x_k) - \frac{1}{n}\sum_{q=1}^n \beta_q \phi(y_q) \right)^T \right\|_{HS}^2 \\
&= \frac{1}{n^2} \sum_{i,j} \left[ \psi_i(x) \left[ \boldsymbol{I} - \frac{1}{n}\boldsymbol{D}_\beta \mathbf{1}\mathbf{1}^T \right] \boldsymbol{D}_\beta \left[ \boldsymbol{I} - \frac{1}{n}\boldsymbol{D}_\beta \mathbf{1}\mathbf{1}^T \right]^T \phi_j(y)^T \right]^2 \\
&= \frac{1}{n^2} \mathrm{Tr} \left[ \psi_X \boldsymbol{H}_\beta \phi_Y^T \cdot \phi_Y \boldsymbol{H}_\beta \psi_X^T \right] = \frac{1}{n^2} \mathrm{Tr} \left[ \boldsymbol{K}_X \boldsymbol{H}_\beta \boldsymbol{K}_Y \boldsymbol{H}_\beta \right] .
\end{aligned}
$$

On the other hand, we can also obtain the estimators through $U$-statistics and $V$-statistics, which is helpful for the later analysis regarding the asymptotic distributions. On the population level,

$$
\begin{aligned}
\mathrm{HSIC}^\beta(X,Y) &= \mathbb{E}_{X,Y,X',Y'} \left[ \langle \beta\,\psi(X) \otimes \phi(Y), \beta\,\psi(X) \otimes \phi(Y) \rangle_{\mathcal{HS}} \right] + \left\langle \mu_X^\beta \otimes \mu_Y^\beta, \mu_X^\beta \otimes \mu_Y^\beta \right\rangle_{\mathrm{HS}} \\
&\quad - 2\mathbb{E}_{X,Y} \left[ \left\langle \mu_X^\beta \otimes \mu_Y^\beta, \beta\,\psi(X) \otimes \phi(Y) \right\rangle_{\mathcal{HS}} \right] \\
&= \mathbb{E}_{X,\dot{X},Y,\dot{Y}} \left[ \beta\dot{\beta}\, k_X(X,\dot{X}) k_Y(Y,\dot{Y}) \right] + \mathbb{E}_{\dot{X},\ddot{X}} \left[ \dot{\beta}\ddot{\beta}\, k_X(\dot{X},\ddot{X}) \right] \mathbb{E}_{Y,\mathring{Y}} \left[ \beta\mathring{\beta}\, k_Y(Y,\mathring{Y}) \right] \\
&\quad - 2\mathbb{E}_{X,Y} \left[ \beta\, \mathbb{E}_{\dot{X}} \left[ \dot{\beta}\, k_X(X,\dot{X}) \right] \mathbb{E}_{\ddot{Y}} \left[ \ddot{\beta}\, k_Y(Y,\ddot{Y}) \right] \right] .
\end{aligned}
$$

Here $(\dot{X}, \dot{Y})$, $(\ddot{X}, \ddot{Y})$, and $(\mathring{X}, \mathring{Y})$ are independent copies of $(X,Y)$, with corresponding weights $\dot{\beta}$, $\ddot{\beta}$ and $\mathring{\beta}$. The HS norm of $\Sigma_{XY}^\beta$ exists when the various expectations over the kernels are bounded, which is true as long as the kernels $k$ and $l$ are bounded. Based on the last equation we can unbiasedly estimate the reweighted HSIC in terms of the sum of three $U$-statistics:

$$
\mathrm{HSIC}_u^\beta(\mathcal{D}) = \frac{1}{(n)_2} \sum_{(i,j) \in \mathbf{i}_2^n} \beta_i \beta_j k_X^{ij} k_Y^{ij} + \frac{1}{(n)_4} \sum_{(i,j,q,r) \in \mathbf{i}_4^n} \beta_i \beta_j \beta_q \beta_r k_X^{ij} k_Y^{qr} - 2\frac{1}{(n)_3} \sum_{(i,j,q) \in \mathbf{i}_3^n} \beta_i \beta_j \beta_q k_X^{ij} k_Y^{iq},
$$

which can also be formulated as one $U$-statistics as follow (from a similar extension by (Song et al., 2007))

$$
\mathrm{HSIC}_u^\beta(\mathcal{D}) = (n)_4^{-1} \sum_{(i,j,q,r) \in \mathbf{i}_4^n} h^\beta(i,j,q,r), \tag{26}
$$

$$
h^\beta(i,j,q,r) \triangleq \frac{1}{4!} \sum_{(s,t,u,v)}^{(i,j,q,r)} \left( \beta_s \beta_t k_X^{st} k_Y^{st} + \beta_s \beta_t \beta_u \beta_v k_X^{st} k_Y^{uv} - 2\beta_s \beta_t \beta_u k_X^{st} k_Y^{su} \right). \tag{27}
$$

The last sum represents all ordered quadruples $(s,t,u,v)$ selected without replacement from $(i,j,q,r)$. Note that $h^\beta(i,j,q,r)$ is $h_{ijqr}^\beta$ in paper. We can substitute the $U$-statistics with $V$-statistics and get the biased estimator of HSIC:

$$
\mathrm{HSIC}_b^\beta(\mathcal{D}) = \frac{1}{n^2} \sum_{i,j} \beta_i \beta_j k_X^{ij} k_Y^{ij} + \frac{1}{n^4} \sum_{i,j,q,r} \beta_i \beta_j \beta_q \beta_r k_X^{ij} k_Y^{qr} - 2\frac{1}{n^3} \sum_{i,j,q} \beta_i \beta_j \beta_q k_X^{ij} k_Y^{iq} = \frac{1}{n^4} \sum_{i,j,q,r} h^\beta(i,j,q,r). \tag{28}
$$

## D.3. Proof of the Asymptotic distribution of Reweighted HSIC

**Theorem 3.4** (Null distribution). *Define $\mathcal{D} = \{(x_i, y_i)\}_{i=1}^n = \{w_i\}_{i=1}^n$. Under $\mathcal{H}_0$, the V-statistic $\mathrm{HSIC}_b^\beta(\mathcal{D})$ in Eq. (28) is degenerate, meaning $\mathbb{E}_i h_{ijqr}^\beta = 0$. In this case, $\mathrm{HSIC}_b^\beta(\mathcal{D})$ converges in distribution according to*

$$m\,\mathrm{HSIC}_b^\beta(\mathcal{D}) \xrightarrow{D} \sum_{l=1}^\infty \lambda_l \chi_{1l}^2, \tag{29}$$

*where $\chi_{1l}^2$ are i.i.d. chi-square variables, and $\lambda_l$ are the solutions to the eigenvalue problem*

$$\lambda_l \psi_l(w_j) = \int \beta_{iqr} \cdot h_{ijqr}^\beta \psi_l(w_i)\,\mathrm{d}F_{i,q,r}, \tag{30}$$

*where the integral is over the distribution of variables $w_i, w_q,$ and $w_r$.*

*Proof.* First we discuss the value of $\zeta_1$ and $\zeta_2$ for $\mathrm{HSIC}_u^\beta(\mathcal{D})$. We calculate $\mathbb{E}_{jqr}\left[h^\beta(i,j,q,r) \mid i\right]$ first for $\zeta_1 = \mathbb{V}\mathrm{ar}_i\left(h_i^\beta(i,j,q,r)\right) = \mathbb{V}\mathrm{ar}_i\left(\mathbb{E}_{jqr}\left[h^\beta(i,j,q,r) \mid i\right]\right)$:

$$\mathbb{E}\left[h^\beta(i,j,q,r)\right] = \frac{1}{4!}\sum_{(s,t,u,v)}^{(i,j,q,r)} \mathbb{E}[\beta_s\beta_t k_X^{st}(k_Y^{st} + \beta_u\beta_v k_Y^{uv} - 2\beta_u k_Y^{su})]$$

$$= \frac{1}{4!}\sum_{(s,t,u,v)}^{(i,j,q,r)} \mathbb{E}[\beta_s\beta_t k_X^{st}][\mathbb{E}[k_Y^{st}] + \mathbb{E}[\beta_u\beta_v]\mathbb{E}[k_Y^{uv}] - \mathbb{E}[2\beta_u k_Y^{su}]].$$

The seconod equation is due to the independence between $X$ and $Y$ under $\mathcal{H}_0$ and in this case we set the input of $\beta$ as $X$. We enumerate all the posibilities of the terms including $i$s:

$$i = s: \beta_i\mathbb{E}\left[\beta_t k_X^{it}\right]\cdot\mathbb{E}\left[k_Y^{it} + k_Y^{uv} - 2k_Y^{iu}\right], \quad i = t: \beta_i\mathbb{E}\left[\beta_s k_X^{st}\right]\cdot\mathbb{E}\left[k_Y^{si} + k_Y^{uv} - 2k_Y^{su}\right],$$

$$i = u: \mathbb{E}\left[\beta_s\beta_t k_X^{st}\right]\cdot\mathbb{E}\left[k_Y^{st} + \beta_i k_Y^{iv} - 2\beta_i k_Y^{si}\right], \quad i = v: \mathbb{E}\left[\beta_s\beta_t k_X^{st}\right]\cdot\mathbb{E}\left[k_Y^{st} + \beta_i l_{u_i} - 2k_Y^{su}\right].$$

When conditioned on $i$, we have $\mathbb{E}[k_Y^{is}] = \mathbb{E}[k_Y^{it}] = \mathbb{E}[k_Y^{iu}] = \mathbb{E}[k_Y^{iv}]$ and $\mathbb{E}[k_Y^{st}] = \mathbb{E}[k_Y^{uv}] = \mathbb{E}[k_Y^{su}]$, etc. Therefore, we have $\mathbb{E}\left[h^\beta(i,j,q,r) \mid i\right] = 0$ for arbitray $i$, which means $\zeta_1 = 0$. Besides, for $\zeta_2 = \mathbb{V}\mathrm{ar}_{ij}\left(h_{ij}^\beta(i,j,q,r)\right) = \mathbb{V}\mathrm{ar}_{ij}\left(\mathbb{E}_{qr}\left[h^\beta(i,j,q,r) \mid i,j\right]\right)$ we also have the following enumerations:

$$i = s, j = t: \quad \beta_i\beta_j k_X^{ij}\cdot\mathbb{E}\left[k_Y^{ij} + k_Y^{uv} - 2k_Y^{iu}\right], \quad j = u: \quad \beta_i\mathbb{E}\left[\beta_t k_X^{it}\right]\cdot\mathbb{E}\left[k_Y^{it} + \beta_j k_Y^{jv} - 2\beta_j k_Y^{ij}\right],$$

$$j = v: \quad \beta_i\mathbb{E}\left[\beta_t k_X^{it}\right]\cdot\mathbb{E}\left[k_Y^{it} + \beta_j k_Y^{uj} - 2k_Y^{iu}\right],$$

$$i = t, j = s: \quad \beta_i\beta_j k_X^{ij}\cdot\mathbb{E}\left[k_Y^{ij} + k_Y^{uv} - 2k_Y^{ju}\right], \quad j = u: \quad \beta_i\mathbb{E}\left[\beta_s k_X^{si}\right]\cdot\mathbb{E}\left[k_Y^{si} + \beta_j k_Y^{jv} - 2\beta_j k_Y^{sj}\right],$$

$$j = v: \quad \beta_i\mathbb{E}\left[\beta_s k_X^{si}\right]\cdot\mathbb{E}\left[k_Y^{si} + \beta_j k_Y^{uj} - 2k_Y^{su}\right],$$

$$i = u, j = s: \quad \beta_j\mathbb{E}\left[\beta_t k_X^{jt}\right]\cdot\mathbb{E}\left[k_Y^{jt} + \beta_i k_Y^{iv} - 2\beta_i k_Y^{ij}\right], \quad j = t: \quad \beta_j\mathbb{E}\left[\beta_s k_X^{sj}\right]\cdot\mathbb{E}\left[k_Y^{sj} + \beta_i k_Y^{iv} - 2\beta_i k_Y^{si}\right],$$

$$j = v: \quad \mathbb{E}\left[\beta_s\beta_t k_X^{st}\right]\cdot\mathbb{E}\left[k_Y^{st} + \beta_j k_Y^{ij} - 2k_Y^{si}\right],$$

$$i = v, j = s: \quad \beta_j\mathbb{E}\left[\beta_t k_X^{jt}\right]\cdot\mathbb{E}\left[k_Y^{jt} + \beta_i k_Y^{ui} - 2k_Y^{ju}\right], \quad j = t: \quad \beta_j\mathbb{E}\left[\beta_s k_X^{sj}\right]\cdot\mathbb{E}\left[k_Y^{sj} + \beta_i k_Y^{ui} - 2k_Y^{su}\right],$$

$$j = u: \quad \mathbb{E}\left[\beta_s\beta_t k_X^{st}\right]\cdot\mathbb{E}\left[k_Y^{st} + \beta_i\beta_j k_Y^{ij} - 2\beta_j k_Y^{sj}\right].$$

Different from the previous discussion, this time it's unable to eliminate the terms and $\mathbb{E}_{qr}\left[h^\beta(i,j,q,r) \mid i,j\right]$ turns out to be a function of $i$ and $j$, i.e., $\zeta_2 > 0$. Then we apply the results in Theorem D.3, and we would have

$$\sqrt{n}\left(\mathrm{HSIC}_b^\beta(\mathcal{D}) - \mathrm{HSIC}^\beta(X,Y)\right) \xrightarrow{d} \sum_{j=1}^\infty \lambda_j \chi_{1j}^2. \tag{31}$$

where $\chi_{1j}^2$ are i.i.d. Chi-square variables, and $\lambda_l$s are the solutions to the eigenvalue problem

$$\lambda_l \psi_l\left(w_j\right) = \int h_{ijqr}^\beta \psi_l\left(w_i\right)\mathrm{d}F_{i,q,r}^\beta = \int \beta_{iqr} \cdot h_{ijqr}^\beta \psi_l\left(w_i\right)\mathrm{d}F_{i,q,r}, \tag{32}$$

where the integral is over the distribution of variables $w_i, w_q$, and $w_r$. $\qquad\square$

**Theorem 3.5.** *Given known $\beta(\cdot)$ function, where $\beta_i = \beta(x_i)$. Given $n$ i.i.d samples $\mathcal{D} = \{(x_i, y_i)\}_{i=1}^n$ with distribution $\mathbb{P}_{XY}$. Using the symmetric kernel function $h^\beta(i,j,q,r)$ we defined above, and assume $\mathbf{E}\left(h^\beta(i,j,q,r)^2\right) < \infty$. Under $\mathcal{H}_1$, an observation of $\mathrm{HSIC}^\beta(X,Y)$, denoted as $\mathrm{HSIC}_b^\beta(\mathcal{D})$, converges in distribution as $n \to \infty$ to a Gaussian according to*

$$\sqrt{n}\left(\mathrm{HSIC}_b^\beta(\mathcal{D}) - \mathrm{HSIC}^\beta(X,Y)\right) \xrightarrow{D} \mathcal{N}(0, \sigma_\beta^2). \tag{33}$$

*The variance is $\sigma_\beta^2 = 16\left(\mathbb{E}_i\left(\mathbb{E}_{j,q,r}h_{ijqr}^\beta\right)^2 - \mathrm{HSIC}^\beta(X,Y)^2\right)$, where $\mathbb{E}_{j,q,r} := \mathbb{E}_{w_j, w_q, w_r}$.*

*Proof.* We know for the biased estimator $\mathrm{HSIC}_b^\beta(\mathcal{D})$ in Eq. (28), its associated U-statistic $\mathrm{HSIC}_u^\beta(\mathcal{D})$ in Eq. (26) has

$$\zeta_1 = \mathbb{V}\mathrm{ar}_i\left(h_i^\beta(i,j,q,r)\right) = \mathbb{V}\mathrm{ar}_i\left(\mathbb{E}_{jqr}\left[h^\beta(i,j,q,r) \mid i\right]\right).$$

Since under $\mathcal{H}_1$, $\mathbb{E}_{jqr}\left[h^\beta(i,j,q,r \mid i)\right]$ would clearly change value for different index $i$ when $n \to \infty$, therefore $\zeta_1 > 0$. Then we have $\mathrm{HSIC}_u^\beta(\mathcal{D})$ converges in distribution as Eq. 8 with variance $\sigma_\beta^2$, according to theorem D.3. Since the difference between $\mathrm{HSIC}_b^\beta(\mathcal{D})$ and $\mathrm{HSIC}_u^\beta(\mathcal{D})$ drops as $1/n$ as shown in Proposition D.2, $\mathrm{HSIC}_b^\beta(\mathcal{D})$ converges asymptotically to the same distribution. $\qquad\square$

### D.4. Derivation of Regularization and Objective Function

For unconditional independence test, we use the negative logarithm of the normalized reweighted HSIC as the dependence loss. The normalized $\mathrm{HSIC}^\beta$ on the population level is shown below, whose estimator is illustrated in paper

$$J_\beta^{UI} = \frac{J_{1\beta}^{UI}}{J_{2\beta}^{UI} J_{3\beta}^{UI}}, \text{ where } J_{1\beta}^{UI} = \sum_{i,j} \mathbb{E}_{\tilde{\mathbb{P}}_{XY}}\left[\left(\psi_i(X) - \mathbb{E}_{\tilde{\mathbb{P}}_X}\psi_i(X)\right)\left(\phi_j(Y) - \mathbb{E}_{\tilde{\mathbb{P}}_Y}\phi_j(Y)\right)\right]^2,$$

$$J_{2\beta}^{UI} = \sum_i \mathbb{E}_{\tilde{\mathbb{P}}_X}\left[\left(\psi_i(X) - \mathbb{E}_{\tilde{\mathbb{P}}_X}\psi_i(X)\right)^2\right], \quad J_{3\beta}^{UI} = \sum_j \mathbb{E}_{\tilde{\mathbb{P}}_Y}\left[\left(\phi_j(Y) - \mathbb{E}_{\tilde{\mathbb{P}}_Y}\phi_j(Y)\right)^2\right].$$

This measure can be considered as a generalization of the squared correlation coefficient. If we replace the operators with normal covariance matrices and HS norms with Frobenious norms (i.e., use an identity map of feature maps $\phi$ and $\varphi$, which will be specified later), then the above measure is the squared correlation coefficient between $X$ and $Y$.

**Proposition D.4.** *The normalized reweighted HSIC, as $J_\beta^{UI}$ defined above, ranges from 0 to 1, i.e., $J_\beta^{UI} \in [0,1]$.*

*Proof.* From Cauchy-Schwarz inequality, we have for all $i, j$

$$\mathbb{C}\mathrm{ov}_{\tilde{\mathbb{P}}_{XY}}[\psi_i(X), \phi_j(Y)] \le \sqrt{\mathbb{V}\mathrm{ar}_{\tilde{\mathbb{P}}_X}[\psi_i(X)]\mathbb{V}\mathrm{ar}_{\tilde{\mathbb{P}}_Y}[\phi_j(Y)]}$$

$$\mathbb{E}_{\tilde{\mathbb{P}}_{XY}}^2\left[\left(\psi_i(X) - \mathbb{E}_{\tilde{\mathbb{P}}_X}\psi_i(X)\right)\left(\phi_j(Y) - \mathbb{E}_{\tilde{\mathbb{P}}_Y}\phi_j(Y)\right)\right] \le \mathbb{E}_{\tilde{\mathbb{P}}_X}\left[\left(\psi_i(X) - \mathbb{E}_{\tilde{\mathbb{P}}_X}\psi_i(X)\right)^2\right]\mathbb{E}_{\tilde{\mathbb{P}}_Y}\left[\left(\phi_j(Y) - \mathbb{E}_{\tilde{\mathbb{P}}_Y}\phi_j(Y)\right)^2\right].$$

Summing over all the $i$ and $j$ gives us $J_{1\beta}^{UI} \le J_{2\beta}^{UI} J_{3\beta}^{UI}$. The equality is achieved when $X = Y$. $\qquad\square$

On the sample level, we use the normalized $\text{HSIC}_b^\beta$, $\hat{J}_\beta^{UI} = \frac{\hat{J}_{1\beta}^{UI}}{\hat{J}_{2\beta}^{UI}\hat{J}_{3\beta}^{UI}}$, where the denominators can be calculated by,

$$
\begin{aligned}
\hat{J}_2^{UI} &= \sum_i \widehat{\mathbb{E}}\left(\left[\psi(x)\right]_i - \widehat{\mathbb{E}}[\psi(x)]_i\right)^2 = \frac{1}{n}\sum_i\left[\sum_{k=1}^n \beta_k\left(\psi_i(x_k) - \frac{1}{n}\sum_{q=1}^n \beta_q \psi_i(x_q)\right)^2\right] \\
&= \frac{1}{n}\sum_i\left[\psi_i(x)\left[\boldsymbol{I} - \frac{1}{n}\boldsymbol{D}_\beta \mathbf{1}\mathbf{1}^T\right]\boldsymbol{D}_\beta\left[\boldsymbol{I} - \frac{1}{n}\boldsymbol{D}_\beta\mathbf{1}\mathbf{1}^T\right]^T\psi_i(x)^T\right] \\
&= \frac{1}{n}\operatorname{Tr}\left[\psi(x)\boldsymbol{H}_\beta\psi(x)^T\right] = \frac{1}{n}\operatorname{Tr}\left[\boldsymbol{K}_X\boldsymbol{H}_\beta\right].
\end{aligned}
$$

And the derivation of the estimator for $\hat{J}_3^{UI}$ is similar and we have $\hat{J}_3^{UI} = \sum_j \widehat{\mathbb{E}}\left(\left[\phi(y)\right]_j - \widehat{\mathbb{E}}[\phi(y)]_j\right)^2 = \frac{1}{n}\operatorname{Tr}\left[\boldsymbol{K}_Y\boldsymbol{H}_\beta\right]$.

For regularization terms, suppose the reweighting function $\beta(\cdot)$ is selected from an RKHS, i.e., $\beta \in \mathcal{B} \subset \mathcal{F}_X$. Then we can denote $\beta$ as $\beta(X) = \psi_X\omega = \boldsymbol{K}_X\alpha$, where $\omega$ is the vector of coefficients that represent $\beta$ in $X$'s RKHS, i.e., $\omega = \psi_X^T\alpha$, and $\alpha$ is a vector of parameters. In order to ensure the smoothness of $\beta$ function to avoid drastic changes for nearby inputs, we want $\|\omega\|^2 = \alpha^\top \boldsymbol{K}_Z\alpha$ to be as small as possible. At the same time, we want to select as many data points as possible, i.e., $\beta_i = \beta(x_i)$ should preferably be close to $1$. Notice that $\beta(X)$ has a mean of $1$, so this condition can be achieved by minimizing the sample variance of $\beta_i$, i.e.,

$$
\frac{1}{n}\sum_{i=1}^n (\beta_i - 1)^2 = \frac{1}{n}\|\boldsymbol{K}_Z\alpha - \mathbf{1}\|^2 = \frac{1}{n}(\alpha^\top\boldsymbol{K}_Z - \mathbf{1}^\top)(\boldsymbol{K}_Z\alpha - \mathbf{1}) = \frac{1}{n}\alpha^\top\boldsymbol{K}_Z^2\alpha - \frac{2}{n}\mathbf{1}^\top\boldsymbol{K}_Z\alpha + 1. \tag{34}
$$

Note that overall these two regularization terms make the variance of the importance reweighted distribution larger. Adding all the regularization terms and simplifying the calculation by taking logarithm, the final expected minimization objective function subject to constraint $\sum_{i=1}^n \beta_i = n$ and $\beta_i \geq 0, \forall i$ is:

$$
\operatorname*{arg\,min}_\alpha \; L \iff \operatorname*{arg\,min}_\alpha \; -\log\hat{J}_\beta^{UI} + \lambda_1\|\omega\|^2 + \frac{\lambda_2}{n}\sum_{i=1}^n(\beta_i - 1)^2,
$$

where $\hat{J}_\beta^{UI} = \frac{\operatorname{Tr}[\boldsymbol{K}_X\boldsymbol{H}_\beta\boldsymbol{K}_Y\boldsymbol{H}_\beta]}{\operatorname{Tr}[\boldsymbol{K}_X\boldsymbol{H}_\beta]\operatorname{Tr}[\boldsymbol{K}_Y\boldsymbol{H}_\beta]}$ and

$$
\begin{aligned}
L = &-\log\operatorname{Tr}\left[\boldsymbol{K}_X\boldsymbol{D}_\beta\left(\boldsymbol{I} - \frac{1}{n}\mathbf{1}\mathbf{1}^\top\boldsymbol{D}_\beta\right)\boldsymbol{K}_Y\boldsymbol{D}_\beta\left(\boldsymbol{I} - \frac{1}{n}\mathbf{1}\mathbf{1}^\top\boldsymbol{D}_\beta\right)\right] \\
&+\log\operatorname{Tr}\left[\boldsymbol{K}_X\boldsymbol{D}_\beta\left(\boldsymbol{I} - \frac{1}{n}\mathbf{1}\mathbf{1}^\top\boldsymbol{D}_\beta\right)\right] + \log\operatorname{Tr}\left[\boldsymbol{K}_Y\boldsymbol{D}_\beta\left(\boldsymbol{I} - \frac{1}{n}\mathbf{1}\mathbf{1}^\top\boldsymbol{D}_\beta\right)\right] \\
&+\frac{1}{2}\alpha^\top\left(\lambda_1\boldsymbol{K}_Z + \frac{\lambda_2}{n}\boldsymbol{K}_Z^2\right)\alpha - \frac{\lambda_2}{n}\mathbf{1}^\top\boldsymbol{K}_Z\vec{\alpha}.
\end{aligned}
\tag{35}
$$

That is, we actually find the optimized $\alpha$ and then calculate $\beta = \boldsymbol{K}_X\alpha$ instead of optimizing $\beta$ directly. We can also translate it into an unconstrained optimization problem, where $\beta = \operatorname{softmax}(\epsilon)$ and $\arg\min$ is taken w.r.t. $\epsilon$. Note that through this transformation, we restrict $\beta$ satisfies the constraints for each optimization step, while in constrained optimization we allow a slight violation (e.g, trust-region Newton-CG (trust-ncg) and Sequential Least Squares Quadratic Programming (SLSQP)). We see the constrianed method significantly outperforms the unconstrained method, which demonstrates that a looser constraint space help find the better solution. For conditional independence test, we substitute $\hat{J}_{1\beta}^{CI}$, $\hat{J}_{2\beta}^{CI}$, and $\hat{J}_{3\beta}^{CI}$ with $\hat{J}_{1\beta}^{CI}$, $\hat{J}_{2\beta}^{CI}$, and $\hat{J}_{3\beta}^{CI}$, and get the final objective function accordingly.

### D.5. Proof of the Uniform Convergence Bound

To measure whether our estimated reweighting function ($\hat{\beta}$) obtained in $\mathcal{D}$ generalizes well on the whole population, we are interested in bounding the difference of $\text{HSIC}^{\hat{\beta}}(X, Y)$ and the optimal reweighted measure $\text{HSIC}^{\beta^*}(X, Y)$, using a decomposition described below.

**Overall Learning Framework**

$$\text{HSIC}^{\beta^*}(X, Y) - \text{HSIC}^{\hat{\beta}}(X, Y)$$

$$= \underbrace{\left[\text{HSIC}^{\beta^*}(X, Y) - \text{HSIC}_b^{\beta^*}(\mathcal{D})\right]}_{A} + \underbrace{\left[\text{HSIC}_b^{\beta^*}(\mathcal{D}) - \text{HSIC}_b^{\hat{\beta}}(\mathcal{D})\right]}_{B} + \underbrace{\left[\text{HSIC}_b^{\hat{\beta}}(\mathcal{D}) - \text{HSIC}^{\hat{\beta}}(X, Y)\right]}_{C}$$

$$\leq \underbrace{\sup_{\beta \in \mathcal{B}} \left[\text{HSIC}^{\beta}(X, Y) - \text{HSIC}_b^{\beta}(\mathcal{D})\right]}_{A'} + 0 + \underbrace{\left[\text{HSIC}_b^{\hat{\beta}}(\mathcal{D}) - \text{HSIC}^{\hat{\beta}}(X, Y)\right]}_{C}$$

$$\leq \underbrace{2 \sup_{\beta \in \mathcal{B}} \left|\text{HSIC}^{\beta}(X, Y) - \text{HSIC}_b^{\beta}(\mathcal{D})\right|}_{A''}.$$

Therefore, if we can prove $A'' \to 0$ as the sample size $n \to \infty$, which requires $\text{HSIC}^{\beta}(X, Y)$ convergences to $\text{HSIC}_b^{\beta}(\mathcal{D})$ for each $\beta \in \mathcal{B}$, we can actually obtain the convergence property of $\text{HSIC}^{\hat{\beta}}(X, Y)$ to $\text{HSIC}^{\beta^*}(X, Y)$ as $n \to \infty$. In the following, we will represent this uniform convergence result. First, we list some required assumptions in the proof.

**Assumptions**

(i) $\mathcal{X} \subset \mathbb{R}^d$ is a closed and bounded space.

(ii) The functions $\beta(\cdot) \in \mathcal{B}$ are continuous and Lipschitz.

(iii) The kernels $k_X$ and $k_Y$ are uniformly bounded by the lower bound 0 and the upper bound:

$$\sup_{x \in \mathcal{X}} k_X(x, x) \leq \nu_1, \quad \sup_{y \in \mathcal{Y}} k_Y(y, y) \leq \nu_2.$$

For the kernels we use for experiments in paper (e.g., Gaussian kernels), $\nu_1 = \nu_2 = \nu = 1$.

*Remark* D.5. Based on assumption (i), let $x \in [-M_1, M_1]$, $\forall x \in \mathcal{X}$. In practice this condition is easily satisfied since we usually normalize the input data into $[0, 1]$ or with mean 0 and standard deviation 1. For assumption (ii), we set the Lipschitz parameter as $M_2$, i.e., for all $x, x' \in \mathcal{X}$ and $x, x' \in [-M_1, M_1]$, $|\beta(x) - \beta(x')| \leq M_2 |x - x'|$. Furthermore, with these assumptions, the value of $\beta(x)$ is also bounded. We denote the bound as $R$, i.e., $\beta(x) \in [0, R], \forall x \in [-M_1, M_1]$. Therefore $\forall \beta \in \mathcal{B}, \|\beta(x)\|_\infty \leq R, \forall x \in \mathcal{X}$. Note that the assumptions (i) and (ii) do not restrict the specific form of the kernels, and common kernels (Gaussian, Laplacian, etc.) and the kernels used in our paper satisfy these properties.

**Lemma D.6** (McDiarmid's Inequality). *Consider $n$ independent random variables $X_1, \ldots, X_n$, and a real-valued function $f(X_1, \ldots, X_n)$ that satisfies the following inequality*

$$\sup_{x_1, \ldots, x_n, x_i'} |f(x_1, \ldots, x_n) - f(x_1, \ldots, x_{i-1}, x_i', x_{i+1}, \ldots, x_n)| \leq c_i$$

*for all $1 \leq i \leq n$. Then for all $\epsilon > 0$:*

$$\mathbb{P}\left[f(X_1, \ldots, X_n) \geq \mathbb{E}f(X_1, \ldots, X_n) + \epsilon\right] \leq \exp\left(\frac{-2\epsilon^2}{\sum_{i=1}^n c_i^2}\right).$$

*Similarly:*

$$\mathbb{P}\left[f(X_1, \ldots, X_n) \leq \mathbb{E}f(X_1, \ldots, X_n) - \epsilon\right] \leq \exp\left(\frac{-2\epsilon^2}{\sum_{i=1}^n c_i^2}\right).$$

**Lemma D.7** (Covering Number of Lipschitz Function Class). *Let $\mathcal{F}$ be the set of functions defined on the interval $[-M_1, M_1]$ with the following properties:*

$$\mathcal{F} = \{f : [-M_1, M_1] \to [0, R], |f(x) - f(y)| \leq M_2 |x - y|, \forall x, y \in [-M_1, M_1]\},$$

*where $M_1, R, M_2 > 0$. Then, the $\epsilon$-covering number $N(\mathcal{F}, \epsilon, \|\cdot\|_\infty)$ of $\mathcal{F}$ under the $\|\cdot\|_\infty$ norm satisfies:*

$$\mathcal{N}(\mathcal{F}, \epsilon, \|\cdot\|_\infty) \leq (4R/\epsilon)^{8M_1 M_2/\epsilon}$$

*for any $\epsilon \in (0, \min\{R, 2M_1 M_2\})$.*

*Proof.* **Partition in the domain:** Divide the domain $[-M_1, M_1]$ into a uniform grid of points:

$$\mathcal{X} = \{x_1, x_2, \ldots, x_N\}, \quad x_i = -M_1 + (i-1)\delta, \quad \delta = \frac{\epsilon}{M_2}.$$

The number of grid points is:

$$N = \lceil 2M_1/\delta \rceil = \lceil 2M_1 M_2/\epsilon \rceil \leq 4M_1 M_2/\epsilon,$$

since $2M_1 M_2/\epsilon \geq 1$.

**Discretization of Function Values:** At each sampling point $x_i \in \mathcal{X}$, the function value $f(x_i)$ is restricted to $[0, R]$. To ensure a precision of $\epsilon$, discretize $[0, R]$ into steps of size $\epsilon$:

$$\mathcal{Y} = \{0, \epsilon, 2\epsilon, \ldots, R\}.$$

The number of discrete values is:

$$\lceil R/\epsilon \rceil \leq 2R/\epsilon,$$

since $R/\epsilon \geq 1$.

**Function Set Construction:** Using the discretized values at the grid points, the set of possible functions is:

$$\mathcal{F}_{\text{cover}} = \{f : \mathcal{X} \to \mathcal{Y}\}.$$

The total number of such functions is:

$$|\mathcal{F}_{\text{cover}}| = (2R/\epsilon)^{4M_1 M_2/\epsilon}.$$

**Covering Property:** By Lipschitz continuity, any function $f \in \mathcal{F}$ can be approximated by a function in $\mathcal{F}_{\text{cover}}$ with a maximum error of $\epsilon$ in the $\|\cdot\|_\infty$ norm. Thus all the close balls whose centers are in $\mathcal{F}_{\text{cover}}$ can cover $\mathcal{F}$. Notice that those centers may lie outside $\mathcal{F}$. Consider close balls $\{\bar{B}(g, \epsilon)\}$ where $g \in \mathcal{F}_{\text{cover}}$ but $g \notin \mathcal{F}$. Choose a $h \in \bar{B}(g, \epsilon) \bigcap \mathcal{F}$ if any. Thus close balls $\{\bar{B}(h, 2\epsilon)\}$ can cover $\mathcal{F}$. Furthermore, the $\|\cdot\|_\infty$-covering number for $\mathcal{F}$ satisfies:

$$\mathcal{N}(\mathcal{F}, \epsilon, \|\cdot\|_\infty) \leq (4R/\epsilon)^{8M_1 M_2/\epsilon}.$$

Without loss of generality, for the ease of notation, we define $M = M_1 M_2$ in our proof. $\square$

**Theorem D.8.** *Let $\eta_\beta$ denote $\text{HSIC}^\beta(X, Y)$ on the importance reweighted distribution $\tilde{\mathbb{P}}_{XY}$, $\hat{\eta}_\beta^{(u)}$ denote the corresponding (unbiased) estimator of $\eta_\beta$, $\Delta_\eta^{(u)}(\beta) := \hat{\eta}_\beta^{(u)} - \eta_\beta$ represents a random error function. $\hat{\eta}_\beta^{(b)}$ and $\Delta_\eta^{(b)}(\beta)$ are their biased counterparts. Under assumptions (i) to (iii), then we have that with probability at least $1 - \delta$,*

$$\sup_\beta |\Delta_\eta^{(u)}(\beta)| \leq (1 + 3R + 2R^2)^2 R^2 \nu^2 \sqrt{\frac{2}{n} \log \frac{2}{\delta} + \frac{16M}{3R} \frac{\log n}{n^{\frac{2}{3}}}} + \frac{4}{n^{\frac{1}{3}}} (1 + 3R + 2R^2) R^2 \nu^2. \tag{36}$$

*Proof.* We use McDiarmid's inequality in Lemma D.6 to obtain the bound.

First, for fixed $\beta = \beta(x)$, we show that $\Delta_\eta(\beta)$ fits the bounded differences property. Since we fix the weights in this part, for simplicity we omit the subscript $\beta$ from the statistics, e.g., shorten $\hat{\eta}_\beta^{(u)}$ to $\hat{\eta}$. Then we replace $(x_1, y_1)$ with $(x_1', y_1')$ and keep the remaining samples the same. The newly obtained samples are named as $\mathcal{D}'$, and its corresponding weights are $\beta' = \beta(x') = \psi(x')^T \omega$. Define $\beta_{ij} = \beta_i \beta_j$, $\beta_{ijq} = \beta_i \beta_j \beta_q$, $\beta_{ijqr} = \beta_i \beta_j \beta_q \beta_r$. The difference between

$$\hat{\eta} := \frac{1}{(n)_2} \sum_{(i,j) \in \mathbf{i}_2^n} \beta_{ij} k_X^{ij} k_Y^{ij} + \frac{1}{(n)_4} \sum_{(i,j,q,r) \in \mathbf{i}_4^n} \beta_{ijqr} k_X^{ij} k_Y^{qr} - \frac{2}{(n)_3} \sum_{(i,j,q) \in \mathbf{i}_3^n} \beta_{ijq} k_X^{ij} k_Y^{iq}, \tag{37}$$

and the new substitution $\hat{\eta}' := \mathrm{HSIC}_u^\beta(\mathcal{D}')$ can be given by (also define $\beta'_{ij} = \beta'_i\beta'_j$, $\beta'_{ijq} = \beta'_i\beta'_j\beta'_q$, and $\beta'_{ijqr} = \beta'_i\beta'_j\beta'_q\beta'_r$)

$$\left|\hat{\eta} - \hat{\eta}'\right| \le \frac{1}{(n)_2} \sum_{\substack{(i,j)\in\mathbf{i}_2^n \\ 1\in\{i,j\}}} \left|\beta_{ij}k_X^{ij}k_Y^{ij} - \beta'_{ij}k_X^{'ij}k_Y^{'ij}\right| + \frac{1}{(n)_4} \sum_{\substack{(i,j,q,r)\in\mathbf{i}_4^n \\ 1\in\{i,j,q,r\}}} \left|\beta_{ijqr}k_X^{ij}k_Y^{qr} - \beta'_{ijqr}k_X^{'ij}k_Y^{'qr}\right| + \frac{2}{(n)_3} \sum_{\substack{(i,j,q)\in\mathbf{i}_3^n \\ 1\in\{i,j,q\}}} \left|\beta_{ijq}k_X^{ij}k_Y^{iq}\right.$$

$$\left. - \beta'_{ijq}k_X^{'ij}k_Y^{'iq}\right| \le 2 \cdot \frac{(n-1)_1}{(n)_2}R^2\nu^2 + 4 \cdot \frac{(n-1)_3}{(n)_4}R^4\nu^2 + 3 \cdot \frac{2(n-1)_2}{(n)_3}R^3\nu^2 = \frac{2(1+3R+2R^2)R^2\nu^2}{n}.$$

Since for all $i, j$, the term $k_X^{ij}, k_Y^{ij}, k_X^{'ij}, k_Y^{'ij}$ are all in the range $[0, \nu]$ by assumption D.5.i and $\beta_i \in [0, R]$ by assumption D.5.ii. Also note that all the terms that none of $i, j, q, r$ are one is zero. Now using McDiarmid's inequality, we finish the first part of the proof, that is, for fixed $\beta$, with probability at least $1 - \delta$,

$$\left|\Delta_\eta^{(u)}(\beta)\right| \le (1+3R+2R^2)R^2\nu^2\sqrt{\frac{2}{n}\log\frac{2}{\delta}}. \tag{38}$$

Next, we consider the case where $\beta$ changes. Take the function space $\mathcal{B}$ as an example. Firstly since the function space $\mathcal{B}$ is compact, the covering number $\mathcal{N}(\mathcal{B}, r)$, defined as the smallest number of closed balls with centers in $\mathcal{B}$ and radii $r$ whose union covers $\mathcal{B}$, is finite. According to Lemma D.7, for many smooth kernels, e.g., Gaussian kernel, we have

$$\mathcal{N}(\mathcal{B}, r, \|\cdot\|_\infty) \le (4R/r)^{8M/r} \Rightarrow \log\mathcal{N}(\mathcal{B}, r, \|\cdot\|_\infty) \le \frac{8M}{r}\log\left(\frac{R}{r}\right). \tag{39}$$

Also, combining with the assumption D.5.ii, we have for any two $\beta, \tilde{\beta} \in \mathcal{B}$,

$$\left|\hat{\eta}_\beta^{(u)} - \hat{\eta}_{\tilde{\beta}}^{(u)}\right| \le \frac{1}{(n)_2} \sum_{(i,j)\in\mathbf{i}_2^n} \left|\left(\beta_{ij} - \tilde{\beta}_{ij}\right)k_X^{ij}k_Y^{ij}\right| + \frac{1}{(n)_4} \sum_{(i,j,q,r)\in\mathbf{i}_4^n} \left|\left(\beta_{ijqr} - \tilde{\beta}_{ijqr}\right)k_X^{ij}k_Y^{qr}\right|$$

$$+ \frac{2}{(n)_3} \sum_{(i,j,q)\in\mathbf{i}_3^n} \left|\left(\beta_{ijq} - \tilde{\beta}_{ijq}\right)k_X^{ij}k_Y^{iq}\right| \le 2(1+3R+2R^2)R\nu^2\left\|\beta(x) - \tilde{\beta}(x)\right\|_\infty.$$

We derive the last inequality term by term. For the first term, since $\|\beta(x)\|_\infty \le R$ and $|k_X^{ij}k_Y^{qr}| \le |k_X^{ij}||k_Y^{ij}| \le \nu^2$, we have

$$\frac{1}{(n)_2} \sum_{(i,j)\in\mathbf{i}_2^n} \left|\left(\beta_{ij} - \tilde{\beta}_{ij}\right)k_X^{ij}k_Y^{ij}\right| = \frac{1}{(n)_2} \sum_{(i,j)\in\mathbf{i}_2^n} \left|\left(\beta_i - \tilde{\beta}_i\right)\beta_j + \tilde{\beta}_i\left(\beta_j - \tilde{\beta}_j\right)\right| \cdot \left|k_X^{ij}k_Y^{ij}\right| \le 2R\nu^2\left\|\beta(x) - \tilde{\beta}(x)\right\|_\infty.$$

Similarly, for the second and third term we have

$$\frac{1}{(n)_4} \sum_{(i,j,q,r)\in\mathbf{i}_4^n} \left|\left(\beta_{ijqr} - \tilde{\beta}_{ijqr}\right)k_X^{ij}k_Y^{qr}\right| = \frac{1}{(n)_2} \sum_{(i,j)\in\mathbf{i}_2^n} \left|\left(\beta_i - \tilde{\beta}_i\right)\beta_{jqr} + \tilde{\beta}_i\left(\beta_j - \tilde{\beta}_j\right)\beta_{qr}\right.$$

$$\left. + \tilde{\beta}_{ij}\left(\beta_q - \tilde{\beta}_q\right)\beta_r + \tilde{\beta}_{ijq}\left(\beta_r - \tilde{\beta}_r\right)\right| \cdot \left|k_X^{ij}k_Y^{qr}\right| \le 4R^3\nu^2\left\|\beta(x) - \tilde{\beta}(x)\right\|_\infty.$$

$$\frac{2}{(n)_3} \sum_{(i,j,q)\in\mathbf{i}_3^n} \left|\left(\beta_{ijq} - \tilde{\beta}_{ijq}\right)k_X^{ij}k_Y^{iq}\right| = \frac{2}{(n)_3} \sum_{(i,j,q)\in\mathbf{i}_3^n} \left|\left(\beta_i - \tilde{\beta}_i\right)\beta_{jq} + \tilde{\beta}_i\left(\beta_j - \tilde{\beta}_j\right)\beta_q\right.$$

$$\left. + \tilde{\beta}_{ij}\left(\beta_q - \tilde{\beta}_q\right)\right| \cdot \left|k_X^{ij}k_Y^{qr}\right| \le 6R^2\nu^2\left\|\beta(x) - \tilde{\beta}(x)\right\|_\infty.$$

and using the property of unbiased estimate, then

$$\left|\eta_\beta - \eta_{\tilde{\beta}}\right| = \left|\mathbb{E}\hat{\eta}_\beta^{(u)} - \mathbb{E}\hat{\eta}_{\tilde{\beta}}^{(u)}\right| \le \mathbb{E}\left[\left|\hat{\eta}_\beta^{(u)} - \hat{\eta}_{\tilde{\beta}}^{(u)}\right|\right] \le 2(1+3R+2R^2)R\nu^2\left\|\beta(x) - \tilde{\beta}(x)\right\|_\infty. \tag{40}$$

As a result, for any function $\beta \in \mathcal{B}$, we can find a function $\beta^k$ in the cover set such that

$$\left\|\beta(x) - \beta^k(x)\right\|_\infty \le r, \tag{41}$$

and define $(1 + 3R + 2R^2)R \triangleq A$ which gives us

$$\left|\Delta_\eta^{(u)}(\beta)\right| \leq \left|\Delta_\eta^{(u)}(\beta^k)\right| + 4(1 + 3R + 2R^2)R\nu^2 r = \left|\Delta_\eta^{(u)}(\beta^k)\right| + 4A\nu^2 r. \tag{42}$$

Combining with the result in the first part, we show that with probability at least $1 - \delta$,

$$\sup_\beta \left|\Delta_\eta^{(u)}(\beta)\right| \leq \max_k \left|\Delta_\eta^{(u)}(\beta^k)\right| + 4A\nu^2 r$$
$$\leq A^2 \nu^2 \sqrt{\frac{2}{n} \log \frac{2\mathcal{N}(\mathcal{B}, r, \|\cdot\|_\infty)}{\delta}} + 4A\nu^2 r. \tag{43}$$

We finish the proof of this part by combining inequality (39) and setting the radis $r = Rn^{-\frac{1}{3}}$. $\qquad\square$

Therefore, omit the constants and we have the following uniform bound:

**Theorem 3.7** (Uniform Bound). *Suppose $\mathcal{X}$ is a closed and bounded space and the values of the kernels $k_X$ and $k_Y$ are also bounded. Assume the reweighting functions $\beta \in \mathcal{B}$ are continuous and Lipschitz. Then with probability at least $1 - \delta$,*

$$\sup_{\beta \in \mathcal{B}} \left|\mathrm{HSIC}_b^\beta(\mathcal{D}) - \mathrm{HSIC}^\beta(X, Y)\right| \sim \mathcal{O}\left(\sqrt{\frac{1}{n} \log \frac{1}{\delta}} + \frac{\log n}{n^{\frac{2}{3}}} + \frac{1}{n^{\frac{1}{3}}}\right). \tag{44}$$

### D.6. Relation with (Xu & Zheng, 2024).

(Xu & Zheng, 2024) focuses on representation learning and proposes a feature learning framework maximizing dependence with the target by using an extension of Rényi maximal correlation (RMC) in $L^2$ space, with functions parameterized via neural networks. While the goals are different - our paper focuses on hypothesis testing, both works leverage dependence-maximization criteria to learn meaningful functions. Theoretically, RMC provides a general measure of nonlinear dependence via optimal function pairs in $L^2$. Modal decomposition captures the full spectrum of the cross-covariance operator, with the leading mode corresponding to RMC. Restricting to RKHS and aggregating all squared singular values yields HSIC. While RMC is more general, its estimation and statistical inference in $L^2$ neural networks is challenging. In contrast, our methods inherit the kernel-based formulation of HSIC, allowing efficient estimation by kernel trick and asymptotic distribution analysis with statistical guarantees.

## E. Details about the Conditional Independence Statistic

### E.1. Characterization of Conditional Independence

We first define characteristic kernel:

**Lemma E.1** (Characteristic Kernel (Fukumizu et al., 2007)). *A kernel $\mathcal{K}_\mathcal{X}$ is characteristic, if $\forall f \in \mathcal{F}_X$ the condition $\mathbb{E}_{X \sim \mathbb{P}_X}[f(X)] = \mathbb{E}_{X \sim \mathbb{Q}_X}[f(X)]$ implies $\mathbb{P}_X = \mathbb{Q}_X$, where $\mathbb{P}_X$ and $\mathbb{Q}_X$ are two probability distributions of $X$.*

Characteristic kernels have several interesting and useful properties. Most common kernels we used are characteristic, like the Gaussian kernel and Laplacian kernel. As shown in Lemma 3.10, if we use characteristic kernel and define $\ddot{X} \triangleq (X, Z)$, the characterization of CI could be related to the partial cross-covariance operator as $\Sigma_{\ddot{X}Y|Z} = 0 \Longleftrightarrow X \perp Y|Z$, where $\Sigma_{\ddot{X}Y|Z} = \Sigma_{\ddot{X}Y} - \Sigma_{\ddot{X}Z}\Sigma_{ZZ}^{-1}\Sigma_{ZY}$.

We then explain more about the partial cross-covariance operator. For $(X, Y) \in \mathcal{X} \times \mathcal{Y}$, the cross-covariance operator $\Sigma_{XY} : \mathcal{F}_Y \to \mathcal{F}_X$ is defined by (Fukumizu et al., 2004)

$$\langle f, \Sigma_{XY} g \rangle_{\mathcal{F}_X} = \mathbb{E}_{XY}[f(X)g(Y)] - \mathbb{E}_X[f(X)]\mathbb{E}_Y[g(Y)], \ \forall f \in \mathcal{F}_X, g \in \mathcal{F}_Y. \tag{45}$$

The partial cross-covariance operator is defined as

$$\Sigma_{XY|Z} = \Sigma_{XY} - \Sigma_{XZ}\Sigma_{ZZ}^{-1}\Sigma_{ZY}.^3 \tag{46}$$

---

[3]If $\Sigma_{ZZ}$ is not invertible, use the right inverse instead of the inverse. See (Fukumizu et al., 2007).

which can be understood by analogy to the partial cross-covariance between $X$ and $Y$ given $Z$ when they are jointly Gaussian variables. Generally speaking, it can be seen as a partial cross-covariance between functions: $\{f(X), \forall f \in \mathcal{F}_X\}$ and $\{g(Y), \forall g \in \mathcal{F}_Y\}$ given $\{q(Z), \forall q \in \mathcal{F}_{\mathcal{Z}}\}$ (Li et al., 2024a). From this point of view, another characterization of CI would be more intuitive, which enforces the uncorrelatedness of functions in suitable spaces. We introduce it as follows.

**Lemma E.2** (Characterization of CI based on Partial Association (Daudin, 1980))**.** *Let the probability distribution of $X$ be denoted by $\mathbb{P}_X$, and similarly for the joint probability distribution of $\ddot{X}$ as $\mathbb{P}_{XZ}$ or $\mathbb{P}_{\ddot{X}}$. Define the spaces of square-integrable functions of $X$ and $\ddot{X}$ as $L^2_X$ and $L^2_{\ddot{X}}$, respectively, which satisfy $L^2_X = \left\{ f(X) | \mathbb{E}[f^2] < \infty \right\}$ and $L^2_{\ddot{X}} = \left\{ g(\ddot{X}) | \mathbb{E}[g^2] < \infty \right\}$. Notations are similar for $Y$. Then the conditional independence relation between $X$ and $Y$ given $Z$, i.e., $X \perp\!\!\!\perp Y | Z$, is*

$$\Longleftrightarrow \mathbb{E}(fg) = 0, \quad \forall f \in \mathcal{S}_{\ddot{X}} \text{ and } \forall g \in \mathcal{S}_{\ddot{Y}}, \tag{47}$$

$$\Longleftrightarrow \mathbb{E}(f\tilde{g}) = 0, \quad \forall f \in \mathcal{S}_{\ddot{X}} \text{ and } \forall \tilde{g} \in L^2_{\ddot{Y}}, \tag{48}$$

$$\Longleftrightarrow \mathbb{E}(fg') = 0, \quad \forall f \in \mathcal{S}_{\ddot{X}} \text{ and } \forall g' \in \mathcal{S}'_{Y|Z}, \tag{49}$$

$$\Longleftrightarrow \mathbb{E}(f\tilde{g}') = 0, \quad \forall f \in \mathcal{S}_{\ddot{X}} \text{ and } \forall \tilde{g}' \in L^2_Y. \tag{50}$$

*Where the subspaces of $L^2$ are defined as follows:*

$$\mathcal{S}_{\ddot{X}} \triangleq \left\{ f \in L^2_{\ddot{X}} \mid \mathbb{E}(f|Z) = 0 \right\},$$
$$\mathcal{S}_{\ddot{Y}} \triangleq \left\{ g \in L^2_{\ddot{Y}} \mid \mathbb{E}(g|Z) = 0 \right\},$$
$$\mathcal{S}'_{Y|Z} \triangleq \left\{ g' \mid g' = g(Y) - \mathbb{E}(g|Z), g \in L^2_Y \right\}.$$

Note the subspaces $\mathcal{S}_{\ddot{X}}, \mathcal{S}_{\ddot{Y}}$, and $\mathcal{S}'_{Y|Z}$ can be constructed from the corresponding $L^2$ spaces via nonlinear regression. For instance, $\forall f \in L^2_{\ddot{X}}$, its projection in subspace $\tilde{f} \in \mathcal{S}_{\ddot{X}}$ is given by

$$\tilde{f}(\ddot{X}) = f(\ddot{X}) - \mathbb{E}(f|Z) = f(\ddot{X}) - h^*_f(Z) \tag{51}$$

where $h^*_f(Z) \in L^2_Z$ is the regression function of $f(\ddot{X})$ on $Z$.

We can clearly see the connection between the lemma E.2 and CI characterization for Gaussian variables using uncorrelatedness between variables. Particularly, we can regard Eq. (47) as the uncorrelatedness between any residual function of $\ddot{X}$ given $Z$ and $\ddot{Y}$ given $Z$. If we consider spaces $\mathcal{F}_{\ddot{X}}$ and $\mathcal{F}_Y$ instead of $L^2_{XY}$ and $L^2_Y$, lemma E.2 is then reduced to lemma 3.10. (Zhang et al., 2012) provided a way to derive the KCIT statistic according to Lemma E.2, i.e. $J^{CI} = \|\mathbb{C}ov(\psi_{\ddot{X}} - \mathbb{E}(\psi_{\ddot{X}}|Z), \phi_Y - \mathbb{E}(\phi_Y|Z))\|^2_{\mathcal{HS}}$.

### E.2. Derivation of the Importance Reweighted Statistics for CI Test

By the meaning of importance reweighting, with a fixed function $\beta(\cdot)$, at the population level, we can define the statistic of the reweighted KCIT (RKCIT) as:

$$J^{CI}_\beta \triangleq \left\| \Sigma^\beta_{\ddot{X}Y|Z} \right\|^2_{\mathcal{HS}} = \left\| \mathbb{E}_{\tilde{\mathbb{P}}} \left[ (\psi^\beta_{\ddot{X}|Z} - \mathbb{E}_{\tilde{\mathbb{P}}}[\psi^\beta_{\ddot{X}|Z}]) \otimes (\phi^\beta_{Y|Z} - \mathbb{E}_{\tilde{\mathbb{P}}}[\phi^\beta_{Y|Z}]) \right] \right\|^2_{\mathcal{HS}},$$

where $\psi^\beta_{\ddot{X}|Z} \triangleq \psi_{\ddot{X}} - \mathbb{E}_{\tilde{\mathbb{P}}}[\psi_{\ddot{X}}|Z]$, $\phi^\beta_{Y|Z} \triangleq \phi_Y - \mathbb{E}_{\tilde{\mathbb{P}}}[\phi_Y|Z]$ are the residuals of feature maps regressing on $Z$. Here $\tilde{\mathbb{P}} = \beta(X, Z)\mathbb{P}(X, Y, Z)$. Suppose that $x \triangleq (x_1, \ldots, x_t, \ldots, x_n)$ are the i.i.d. samples for $X$, and $y, z$ are the i.i.d. samples for $Y$, $Z$. Denote the centralized kernel matrices for $\ddot{X}, Y, Z$ as $\widetilde{\boldsymbol{K}}_{\ddot{X}}, \widetilde{\boldsymbol{K}}_Y$ and $\widetilde{\boldsymbol{K}}_Z$, respectively. Given importance weights $\beta$ and the feature map $\psi(\cdot)$, the unbiased estimator $\tilde{\psi}^\beta_u(x)$ of the centralization $\tilde{\psi}^\beta(x) = \psi^\beta(x) - \mathbb{E}[\psi^\beta(x)]$ now becomes $\tilde{\psi}^\beta_u(x) = \psi(x) - \frac{1}{n}\sum_{i=1}^n \beta_i \psi(x_i) = \psi(x)(\boldsymbol{I}_n - \frac{1}{n}\boldsymbol{D}_\beta \mathbf{1}\mathbf{1}^T)$. Therefore we have the reweighted centralized kernel matrix $\widetilde{\boldsymbol{K}}^\beta_Z = \tilde{\psi}^\beta_u(x)^T \tilde{\psi}^\beta_u(x) = (\boldsymbol{I}_n - \frac{1}{n}\mathbf{1}\mathbf{1}^T \boldsymbol{D}_\beta)\boldsymbol{K}_Z(\boldsymbol{I}_n - \frac{1}{n}\boldsymbol{D}_\beta \mathbf{1}\mathbf{1}^T)$. $\widetilde{\boldsymbol{K}}^\beta_{\ddot{X}}$ and $\widetilde{\boldsymbol{K}}^\beta_Y$ are defined similarly.

We use kernel ridge regression to estimate conditional expectations. The estimator of the covariance operator is

$$\widehat{\Sigma}^\beta_{XY} = \frac{1}{n}\sum_{k=1}^n \left( \beta_k \tilde{\psi}^\beta_u(x_k)\tilde{\phi}^\beta_u(y_k)^T \right) = \frac{1}{n}\tilde{\psi}^\beta_u(x)\, \boldsymbol{D}_\beta\, \tilde{\phi}^\beta_u(y)^T, \tag{52}$$

where $\boldsymbol{D}_\beta \triangleq \mathrm{diag}(\beta_1, \beta_2, \ldots, \beta_n)$. The estimators of other covariance operators, i.e., $\widehat{\Sigma}^\beta_{XZ}$, $\widehat{\Sigma}^\beta_{YZ}$, and $\widehat{\Sigma}^\beta_{ZZ}$, are similar. Now we calculate the estimators of the residuals and get the residual matrix $\boldsymbol{R}^\beta_Z$ under reweighted distributions. Notice that $\widehat{\mathbb{E}}_{\mathbb{P}_{\breve{X}}}[\tilde{\psi}^\beta_{\breve{X}}|Z] = \tilde{\varphi}^\beta_u(z)^T \left[\widehat{\Sigma}^\beta_{ZZ} + \lambda \boldsymbol{I}_{\dim(\varphi_Z)}\right]^{-1} \widehat{\Sigma}^\beta_{Z\breve{X}}$ using kernel ridge regression. We used the identity $PB^T \left(BPB^T + R\right)^{-1} = \left(P^{-1} + B^T R^{-1}B\right)^{-1} B^T R^{-1}$ in the derivation.

$$\psi^{\beta\,T}_{\breve{X}|Z} = \tilde{\psi}^\beta_u(\ddot{x})^T - \widehat{\mathbb{E}}_{\mathbb{P}_{\breve{X}}}[\tilde{\psi}^\beta_{\breve{X}}|Z] = \tilde{\psi}^\beta_u(\ddot{x})^T - \tilde{\varphi}^\beta_u(z)^T \left[\widehat{\Sigma}^\beta_{ZZ} + \lambda \boldsymbol{I}_{\dim(\varphi_Z)}\right]^{-1} \widehat{\Sigma}^\beta_{Z\breve{X}} \tag{53}$$

$$= \tilde{\psi}^\beta_u(\ddot{x})^T - \tilde{\varphi}^\beta_u(z)^T \left[\frac{1}{n}\tilde{\varphi}^\beta_u(z)\boldsymbol{D}_\beta\tilde{\varphi}^\beta_u(z)^T + \lambda \boldsymbol{I}_{\dim(\varphi_Z)}\right]^{-1} \frac{1}{n}\tilde{\varphi}^\beta_u(z)\boldsymbol{D}_\beta\tilde{\psi}^\beta_u(\ddot{x})^T \tag{54}$$

$$= \tilde{\psi}^\beta_u(\ddot{x})^T - \tilde{\varphi}^\beta_u(z)^T \left[\tilde{\varphi}^\beta_u(z)\boldsymbol{D}_\beta\tilde{\varphi}^\beta_u(z)^T + \epsilon \boldsymbol{I}_{\dim(\varphi_Z)}\right]^{-1} \tilde{\varphi}^\beta_u(z)\boldsymbol{D}_\beta\tilde{\psi}^\beta_u(\ddot{x})^T \tag{55}$$

$$= \tilde{\psi}^\beta_u(\ddot{x})^T - \tilde{\varphi}^\beta_u(z)^T \frac{1}{\epsilon}\boldsymbol{I}_Z\tilde{\varphi}^\beta_u(z) \left[\tilde{\varphi}^\beta_u(z)^T \frac{1}{\epsilon}\boldsymbol{I}_n\tilde{\varphi}^\beta_u(z) + \boldsymbol{D}_{\frac{1}{\beta}}\right]^{-1} \tilde{\psi}^\beta_u(\ddot{x})^T \tag{56}$$

$$= \tilde{\psi}^\beta_u(\ddot{x})^T - \tilde{\varphi}^\beta_u(z)^T \tilde{\varphi}^\beta_u(z) \left[\tilde{\varphi}^\beta_u(z)^T \tilde{\varphi}^\beta_u(z) + \epsilon \boldsymbol{D}_{\frac{1}{\beta}}\right]^{-1} \tilde{\psi}^\beta_u(\ddot{x})^T \tag{57}$$

$$= \epsilon \boldsymbol{D}_{\frac{1}{\beta}} \left[\widetilde{\boldsymbol{K}}^\beta_z + \epsilon \boldsymbol{D}_{\frac{1}{\beta}}\right]^{-1} \tilde{\psi}^\beta_u(\ddot{x})^T \tag{58}$$

$$= \epsilon \left[\widetilde{\boldsymbol{K}}^\beta_z \boldsymbol{D}_\beta + \epsilon \boldsymbol{I}_n\right]^{-1} \tilde{\psi}^\beta_u(\ddot{x})^T = \boldsymbol{R}^\beta_Z \cdot \tilde{\psi}^\beta_u(\ddot{x})^T. \tag{59}$$

Therefore the residual matrix $\boldsymbol{R}^\beta_Z = \epsilon \left[\widetilde{\boldsymbol{K}}^\beta_Z \boldsymbol{D}_\beta + \epsilon \boldsymbol{I}\right]^{-1}$. The estimated residuals become $\boldsymbol{R}^\beta_Z \cdot \tilde{\psi}^\beta_u(\ddot{x})^T$ and $\boldsymbol{R}^\beta_Z \cdot \tilde{\phi}^\beta_u(y)^T$.

Since the covariance operator of the residuals is $\mathrm{Cov}(\psi^\beta_{\breve{X}|Z}, \phi^\beta_{Y|Z}) = \mathbb{E}[\psi^\beta_{\breve{X}|Z} \otimes \phi^\beta_{Y|Z}]$, then the statistic of RKCIT is

$$\hat{J}^{CI}_\beta = \left\|\frac{1}{n}\tilde{\psi}^\beta_{\breve{X}|Z}\boldsymbol{D}_\beta\tilde{\phi}^{\beta\,T}_{Y|Z}\right\|^2_{HS} = \frac{1}{n^2}\mathrm{Tr}\left[\tilde{\psi}^\beta_{\breve{X}|Z}\boldsymbol{D}_\beta\tilde{\phi}^{\beta\,T}_{Y|Z} \cdot \tilde{\phi}^\beta_{Y|Z}\boldsymbol{D}_\beta\tilde{\psi}^{\beta\,T}_{\breve{X}|Z}\right]$$

$$= \frac{1}{n^2}\mathrm{Tr}\left[\boldsymbol{R}^\beta_Z \cdot \tilde{\psi}^\beta_u(\ddot{x})^T\tilde{\psi}^\beta_u(\ddot{x}) \cdot \boldsymbol{R}^{\beta\,T}_Z\boldsymbol{D}_\beta\boldsymbol{R}^\beta_Z \cdot \tilde{\phi}^\beta_u(y)^T\tilde{\phi}^\beta_u(y) \cdot \boldsymbol{R}^{\beta\,T}_Z\boldsymbol{D}_\beta\right]$$

$$= \frac{1}{n^2}\mathrm{Tr}\left[\boldsymbol{R}^\beta_Z\widetilde{\boldsymbol{K}}^\beta_{\ddot{X}}\boldsymbol{R}^{\beta\,T}_Z\boldsymbol{D}_\beta\boldsymbol{R}^\beta_Z\widetilde{\boldsymbol{K}}^\beta_Y\boldsymbol{R}^{\beta\,T}_Z\boldsymbol{D}_\beta\right]$$

$$= \frac{1}{n^2}\mathrm{Tr}\left[\widetilde{\boldsymbol{K}}^\beta_{\ddot{X}|Z}\widetilde{\boldsymbol{K}}^\beta_{Y|Z}\right],$$

where

$$\widetilde{\boldsymbol{K}}^\beta_{\ddot{X}|Z} = \boldsymbol{R}^\beta_Z\widetilde{\boldsymbol{K}}^\beta_{\ddot{X}}\boldsymbol{R}^{\beta\,T}_Z\boldsymbol{D}_\beta, \quad \widetilde{\boldsymbol{K}}^\beta_{Y|Z} = \boldsymbol{R}^\beta_Z\widetilde{\boldsymbol{K}}^\beta_Y\boldsymbol{R}^{\beta\,T}_Z\boldsymbol{D}_\beta.$$

Therefore, for the CI test, the final objective function is similar to the optimization problem (9) with $\hat{J}^{UI}_\beta$ replaced by $\hat{J}^{CI}_\beta$. As in the unconditional case, in the experiment we use a normalized version of this conditional dependence measure with the denominators $\hat{J}^{CI}_{2\beta} = \mathrm{Tr}[\widetilde{\boldsymbol{K}}^\beta_{\ddot{X}|Z}]$ and $\hat{J}^{CI}_{3\beta} = \mathrm{Tr}[\widetilde{\boldsymbol{K}}^\beta_{Y|Z}]$, which can be obtained in a similar manner:

$$\hat{J}^{CI}_{2\beta} = \left\|\frac{1}{n}\tilde{\psi}^\beta_{\ddot{X}|Z}\boldsymbol{D}_\beta\tilde{\psi}^{\beta\,T}_{\ddot{X}|Z}\right\|_{HS} = \frac{1}{n}\mathrm{Tr}\left[\widetilde{\boldsymbol{K}}^\beta_{\ddot{X}|Z}\right],$$

$$\hat{J}^{CI}_{3\beta} = \left\|\frac{1}{n}\tilde{\phi}^\beta_{Y|Z}\boldsymbol{D}_\beta\tilde{\phi}^{\beta\,T}_{Y|Z}\right\|_{HS} = \frac{1}{n}\mathrm{Tr}\left[\widetilde{\boldsymbol{K}}^\beta_{Y|Z}\right].$$

### E.3. Discussion about the Asymptotic distribution of Reweighted KCI

(Zhang et al., 2012) gave the asymptotic distribution of $\hat{J}^{CI}$ under $\mathcal{H}_0$, which follows the distribution of a weighted sum of chi-square variables that is slightly more complex than that of HSIC. However, the author did not provide the distribution under $\mathcal{H}_1$. For the purpose of conditional independence testing, we compute the distribution under $\mathcal{H}_0$ of $\hat{J}^{CI}_\beta$ using the permutation test. We argue that the ground-truth asymptotic distribution has a similar form to the original $\hat{J}^{CI}$ since we analyze the asymptotic characteristics of a reweighted distribution.

### E.4. Discussion of Uniform bound for Reweighted CI test

Actually, after giving the convergence rate of the estimated kernel ridge regression, the derivation of the uniform bound for $\hat{J}_\beta^{CI}$ is similar to that of $\text{HSIC}_b^\beta$ with the minor difference that $\hat{J}_\beta^{CI}$ is calculated for residual variables. We leave this as the future extension of this work.

## F. Details about Causal Discovery

### F.1. Proof of Rare Dependence PC.

In this section, to ensure the asymptotic correctness of the PC algorithm, all theoretical results are based on the assumptions of causal Markov, faithfulness, and causal sufficiency. These explicit definitions can refer to standard literature (Spirtes et al., 2000). If there is no further statement, we assume that these assumptions hold.

**Assumption 4.1.** $\forall X, Y \in \mathbf{V}$, $Z \subseteq \mathbf{V} \setminus \{X, Y\}$, if $\text{KCIT}(X, Y|Z)$ rejects the null hypothesis, then $X \not\perp\!\!\!\perp Y|Z$. Besides, if both $\text{KCIT}(X, Y|Z)$ and $\text{RKCIT}^{\beta(C)}(X, Y|Z)$ fails to reject the null hypothesis, then $X \perp\!\!\!\perp Y|Z$.

**Rule 1.** $\forall X, Y \in \mathbf{V}$, if $\exists Z \subseteq \mathbf{V} \setminus \{X, Y\}$ s.t. both $\text{KCIT}(X, Y|Z)$ and $\text{RKCIT}^{\beta(C)}(X, Y|Z)$ fail to reject the null hypothesis, then $X$ and $Y$ are not adjacent in $G$.

*Proof.* We prove this by contradiction. Suppose $X$ and $Y$ are adjacent in $G$. Without loss of generality, let $X \to Y$ in $G$. There are two cases we need to consider: (i) there does not exist rare dependence in $\mathbb{P}(X, Y)$, and (ii) there exists rare dependence in $\mathbb{P}(X, Y)$.

**Case I:** If there does not exist rare dependence in $\mathbb{P}(X, Y)$ and $X \to Y$, then $\nexists \mathbf{Z} \subset \mathbf{V} \setminus \{X, Y\}$ such that $X \perp\!\!\!\perp Y \mid Z$ holds, according to the faithfulness assumption. This violates the condition that $\exists Z \subseteq \mathbf{V} \setminus \{X, Y\}$ s.t. $\text{KCIT}(X, Y|Z)$ accept the null hypothesis. Furthermore, by Proposition C.2, $\text{RKCIT}^{\beta(C)}(X, Y \mid Z)$ also fails to reject the null hypothesis. Thus, $X$ and $Y$ are not adjacent in $G$.

**Case II:** According to Assumption 4.1, if both $\text{KCIT}(X, Y|Z)$ and $\text{RKCIT}^{\beta(C)}(X, Y|Z)$ fail to reject the null hypothesis, then $X \perp\!\!\!\perp Y|\mathbf{Z}$. This violates the condition that $X \to Y$ in $G$ (If there exists rare dependence in $\mathbb{P}(X, Y)$, $\text{RKCIT}^{\beta(C)}(X, Y|Z)$ reject the null hypothesis under the large sample).

In summary, $X$ and $Y$ are not adjacent in $G$.

$\square$

**Proposition 4.2.** For a pair of variables $X, Y \in V$, if (a) $\forall Z \subseteq V \setminus \{X, Y\}$ s.t. $\text{KCIT}(X, Y|Z)$ fails to reject the null hypothesis, and (b) for all these $Z$, we have that $\text{RKCIT}^{\beta(C)}(X, Y|Z)$ rejects the null hypothesis. Then, under Assumption 4.1, i) $X$ and $Y$ are adjacent with a rare dependence, or ii) $X$ and $Y$ are not adjacent in $G$ and $C$ must be the direct common effect of $X$ and $Y$.

*Proof.* If $\forall Z \subseteq V \setminus \{X, Y\}$, $\text{KCIT}(X, Y \mid Z)$ fails to reject the null hypothesis, and if there is no rare dependence between $\mathbb{P}(X, Y)$, then $X$ and $Y$ are not adjacent in $G$ according to the faithfulness assumption. Furthermore, if we consider $C$ as a conditional set, i.e., $C \subseteq \mathbf{Z}$, and $\text{KCIT}(X, Y \mid Z)$ rejects the null hypothesis, then $C$ must be the common effect of $X$ and $Y$, as the V-structure is activated. Additionally, if $\text{RKCIT}^{\beta(C)}(X, Y \mid Z)$ rejects the null hypothesis for all $Z$, this implies that there exists rare dependence in $\mathbb{P}(X, Y)$, and thus $X$ and $Y$ must be adjacent in $G$ (otherwise, $\exists Z$ such that $\text{RKCIT}^{\beta(C)}(X, Y \mid Z)$ fails to reject the null hypothesis by Proposition C.2).

$\square$

*Remark* F.1. Proposition 4.2 tells us that if $X \to C \leftarrow Y$ forms a $V$-structure, Rule 1 can not correctly remove the edge between $X$ and $Y$ when executing the PC algorithm to recover the causal skeleton. Consequently, the inferred graph tends to be a superset of the true one.

To further distinguish the rare dependence pattern or when $X$ and $Y$ are not adjacent in $G$ in Proposition 4.2, we present the following Rule 2.

**Rule 2.** *For two variables $X, Y \in \mathbf{V}$ that satisfy the condition in Proposition 4.2, if there exists $Z \subseteq \mathbf{V} \backslash \{X, Y\}$, such that* $\text{RKCIT}^{\beta(C^{perm})}(X, Y|Z)$ *fail to reject the null hypothesis, then $X$ and $Y$ are not adjacent in G. Here $C^{perm}$ denotes the shuffled $C$ in dataset $\mathcal{D}$.*

*Proof.* If two variables $X, Y \in \mathbf{V}$ satisfy the condition in Proposition 4.2, it means that either i) they are adjacent with a rare dependence, or ii) they are the direct causes of $C$ (i.e. $C$ is the direct common effect of $X$ and $Y$).

In case i), suppose $C^{perm}$ is the randomly shuffled version of $C$, and $c_{\pi(i)}$ is the $i$-th sample of $C^{perm}$. Firstly, suppose $C$ is continuous, then $C_i \neq C_j, i \neq j$ a.s. Let $\beta$ be the original reweighting function and $\tilde{\beta}$ be the reweighting function learned for $C^{perm}$. If $\tilde{\beta}(c_{\pi(i)}) = \beta(c_i), \forall i$, then $\text{RKCIT}^{\tilde{\beta}(C^{perm})}(X, Y|Z)$ can still enlarge the dependence measure and reject the null hypothesis, which is the same as $\text{RKCIT}^{\beta(C)}(X, Y|Z)$. Since $\tilde{\beta}$ is a simple composition of $\beta$ and the permutation $\pi$, $\tilde{\beta} \in \mathcal{B}$ and thus learnable. When $C$ is discrete, we can add some small noise, e.g. Gaussian noise, to $C$ for each sample and treat it as a continuous variable.

In case ii), we note that $C^{perm}$, the randomly shuffled version of $C$, satisfies $C^{perm} \perp\!\!\!\perp X, Y, Z$ because the data points in the dataset $\mathcal{D}$ are i.i.d., which means $(x_i, y_i, z_i, c_i) \perp\!\!\!\perp c_{\pi(i)}, \forall i \neq \pi(i)$. Due to the randomness, as $n \to \infty$, $\mathbb{P}(\pi(i) = i) \to 0$. This eliminates the spurious dependence between $X$ and $Y$ caused by selection bias. Therefore, $\text{RKCIT}^{\beta(C^{perm})}(X, Y|Z)$ fails to reject the null hypothesis, since $X$ and $Y$ are indeed independent when conditioned on $Z$ in this case. $\qquad\square$

Based on **Rule 1** and **2**, we summarize the procedure of RD-PC in Algorithm 2. Theorem 4.3 proves its soundness. **Theorem 4.3.** With Assumption 4.1, suppose the causal Markov condition, faithfulness assumption, and causal sufficiency assumption hold. Then, Algorithm 2 correctly recovers the underlying causal graph structure up to its Markov equivalence class.

*Proof.* Under Assumption 4.1, the conditional independence relations can be correctly identified using Rule 1 (Line 5 in Algorithm 2). In Line 6, Rule 1 is applied to remove the edges between $X$ and $Y$ for $X, Y \in \mathbf{V}$. As shown in the proof of Proposition 4.2, there are two cases that need to be distinguished, since the equivalence class between the rare dependence pattern and $C$ may correspond to a collider structure. In Line 7, Rule 2 is used to differentiate these structures and ensure that extraneous edges are removed. By the causal Markov assumption, faithfulness assumption, and causal sufficiency assumption, the causal skeleton can be correctly identified (Spirtes et al., 2000). Moreover, in Line 8, Meek's rule is used to orient the directions, the correctness of which is guaranteed by (Meek, 1995). In summary, in line with the PC algorithm (Spirtes et al., 2000), the causal structure can be identified up to a Markov equivalence class.

$\qquad\square$

### F.2. Relation with ReScore (Zhang et al., 2023).

ReScore (Zhang et al., 2023) aims to optimize the structure recovery of score-based causal discovery methods with fewer spurious edges and more robustness to heterogeneous data. They propose an effective model-agnostic framework of reweighting samples to boost differentiable score-based causal discovery methods. The core idea of ReScore is to identify and upweight less-fitted samples, as these samples provide additional insight into uncovering the true causal edges—compared to those easily fitted through spurious associations. Although RD-PC, which employs our proposed RHSIC and RKCIT with correction rules, is also a sample reweighting-based method for causal discovery, its objective is fundamentally different from that of ReScore. ReScore is designed to mitigate the influence of spurious edges by focusing on less-fitted samples, whereas RD-PC aims to recover true causal edges that are easily overlooked or erroneously removed due to rare dependence. Exploring the connection between rare dependence and less-fitted samples may offer an interesting direction for future research.

## G. Supplementary Experimental Details and Results

**Implementation Details.** Here we provide the implementation details of the methods. In all experiments, we use Gaussian kernels in all kernel-based methods. The significance level is set to $0.05$. The results are obtained after averaging the values in the $100$ tests.

In our method, we use the constrained optimization library provided by scipy to minimize our objective function, and set the

maximum number of iterations to 50. We set the number of permutations to 2000 to approximate the null distribution. The hyperparameters in our objective functions (9) are set to $\lambda_1 = \lambda_2 =$ 1e-3 for RHSIC and $\lambda_1 =$ 1e-6, $\lambda_2 =$ 1e-1 for RKCIT. And the $\epsilon$ for kernel ridge regression is set to 1e-3. For the conditional independence test, we use calibration for our RKCIT when we change the dimension of $Z$ (the empirical threshold is set to 0.0375). We observe that some baseline methods naturally maintain a Type I error rate close to 0.05. Other baseline methods have higher Type I errors and lower power, which means that, however calibrated, their performance is worse than ours. Therefore, for these baseline methods, we report their original results.

### G.1. Synthetic Data

More details about the synthetic datasets are explained in this section, including the implementation details, more experiments under different settings and more baseline models, and the results analysis of F1 and SHD for causal discovery.

#### G.1.1. SYNTHETIC EXPERIMENTS FOR UI

**Data Generation I**: we slightly modify Example 1.1 with different variance $\sigma^2$ for $\epsilon \sim \mathcal{N}(0, \sigma^2)$ to evaluate the power. To test the Type I error, we set $Y = \epsilon$ with fixed $\sigma = 0.5$.

**Data Generation II**: we use a variant of the synthetic experiment in (Strobl et al., 2019). To evaluate the Type I error, we generate the data that follows $X = f_1(\varepsilon_x), Y = f_2(\varepsilon_y)$, where $\varepsilon_x, \varepsilon_y$ are drawn independently from $\mathcal{N}(0, 1)$ and $f_1$ and $f_2$ are smooth functions chosen uniformly from a set $\{x^3, \cos(x), 1 + \sin(3x)^2, 20\sin(4\pi x^2), \text{Gsign}(x)\}$, where $\text{Gsign}(x) = |\varepsilon|\Pi_{i=1}^d \text{sgn}(X_i)$, where $d$ is the dimension of variable $X$ and $\varepsilon \sim \mathcal{N}(0, 1)$. To compare the power, we generate $X = f_1(\varepsilon_x) + \varepsilon_b$, and then we generate $Y = f_2(\varepsilon_y) + \varepsilon_b$ if $X < \tau$ where $\tau$ is a threshold, and $Y = f_2(\varepsilon_y)$ otherwise.

**Data Generation III**: The data generation III is similar to the procedure of DG II except that we use a third variable $Q \sim U(0, 1)$ to compare with the threshold $\tau$ and decide the generation process accordingly.

**Results**: See Fig. 6 for the experiment results for Data Generation III. Other experiment results are in the main paper.

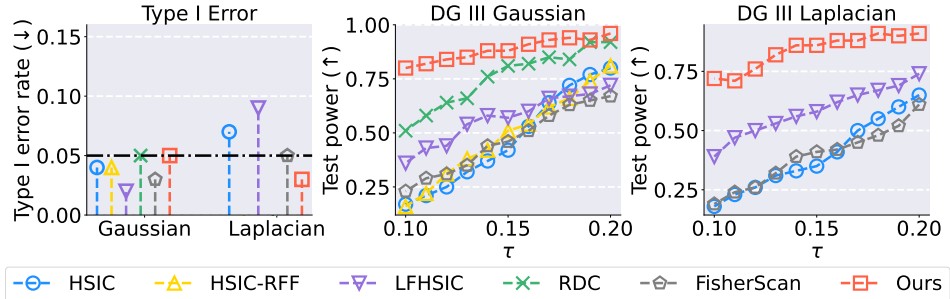

*Figure 6.* Type I error rate and Power of Data Generation III with Gaussian and Laplacian noise.

**Baselines for UI.** All the baselines follow their default settings unless stated otherwise. **HSIC** (Gretton et al., 2007): the original HSIC test using gamma approximation for $p$-value. Code from python library causal-learn (Zheng et al., 2024); **LFHSIC** (Ren et al., 2024): HSIC test with adaptively learned bandwidth. Code from https://github.com/renyixin666/HSIC-LK; **RDC** (Lopez-Paz et al., 2013): use canonical correlation between a finite set of random Fourier features. We permute the samples 500 times to compute the empirical $p$-value. Code from https://github.com/garydoranjr/rdc; **FHSIC** (Zhang et al., 2018): HSIC using finite-dimensional random Fourier feature mappings to approximate kernels. Code from https://github.com/oxcsml/kerpy.

#### G.1.2. SYNTHETIC EXPERIMENTS FOR CI

**Data Generation I (DG I)**: We follow the synthetic experiment proposed in (Scetbon et al., 2022) with a slight variation. To compare the Type I error, we generate simulated data by:

$$X = f_1\left(\bar{Z} + \varepsilon_x\right), Y = f_2\left(\bar{Z} + \varepsilon_y\right) \tag{60}$$

Above, $\bar{Z}$ is the average of $Z = (Z_1, \cdots, Z_{d_z})$, $\varepsilon_x$ and $\varepsilon_y$ are sampled independently from $\mathcal{N}(0, 1)$, and $f_1$ and $f_2$ are smooth functions chosen from the same set as in DG II. The following generating function is for evaluating power:

$$\begin{cases} X = f_1(\bar{Z} + \varepsilon_x) + \varepsilon_b, Y = f_2(\bar{Z} + \varepsilon_y) + \varepsilon_b, & \text{if } Q < \tau, \\ X = f_1(\bar{Z} + \varepsilon_x) + \varepsilon_b, Y = f_2(\bar{Z} + \varepsilon_y), & \text{if } Q \geq \tau. \end{cases}$$

where $Q \sim U(0, 1)$, $\varepsilon_b \sim \mathcal{N}(0, 1)$, $\tau \in [0, 1]$ is a threshold.

**Data Generation II (DG II)**: The data generation II is similar to the procedure of DG I except that we use the $\tau$ percentile of the count of $X$ as the threshold.

**Baselines for CI.** We compare with the following CI methods. **KCIT** (Zhang et al., 2012): the original KCI test based on the partial association framework. Code from python library causal-learn (Zheng et al., 2024); **CCIT** (Sen et al., 2017): a classifier-based test leveraging nearest neighbor sampling to approximate conditional distribution; **RCIT** (Strobl et al., 2019): KCI using random Fourier features to approximate the kernel function. Code from python library causal-learn (Zheng et al., 2024); **GCIT** (Bellot & van der Schaar, 2019): a test employing generative adversarial network to approximate null distribution; **GCM** (Shah & Peters, 2020): a test based on the generalized covariance measure; **NNLSCIT** (Li et al., 2024b): a classifier-based conditional mutual information estimator.

**Results.** See the Fig. 7, Fig. 8 and Fig. 9, we compare our methods with the CI testing.

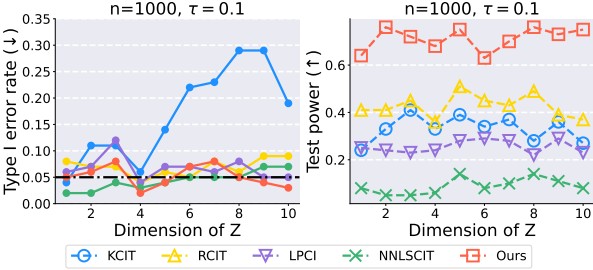

*Figure 7.* Type I error rates and Test powers for CI tests with Data Generation I while Changing the dimension of $Z$. Here the number of samples is fixed $n = 1000$ and the level of rare dependence is $\tau = 0.1$.

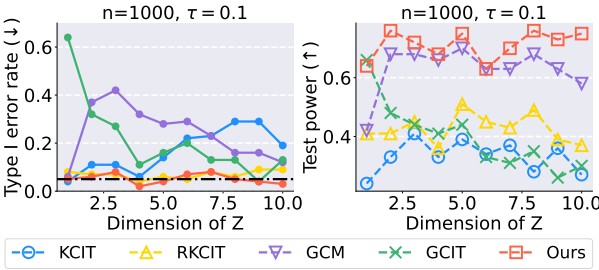

*Figure 8.* Type I error rates and Test powers with Data Generation I for more CI baselines. Again here we change the $Z$ dimensions.

### G.2. Causal Discovery

**Data Generation**: We first randomly generate DAGs with 6 nodes and 10 directed edges from the graphical model Erdös-Rényi (ER) (Erdos & Renyi, 1959). We generate 30 graphs, and for each DAG, we randomly select one node from the graph as the reference variable $C$. We construct the rare dependence between $C$ and its children by adding dependence only when $C$ is in its lower 3% percentile. The synthetic data is generated according to the following nonlinear structural equation:

$$X_i = f_i\left(pa\left(X_i\right) + \epsilon_i\right), i = 1, \ldots, n,$$

where the causal mechanism $f_i$ is randomly selected from functions such as linear, $\sin, \cos, \tanh$, sigmoid, and their combinations and $pa(X_i)$ is the parents set of variable $X_i$. The noise term $\epsilon_i$ is sampled from Gaussian distribution.

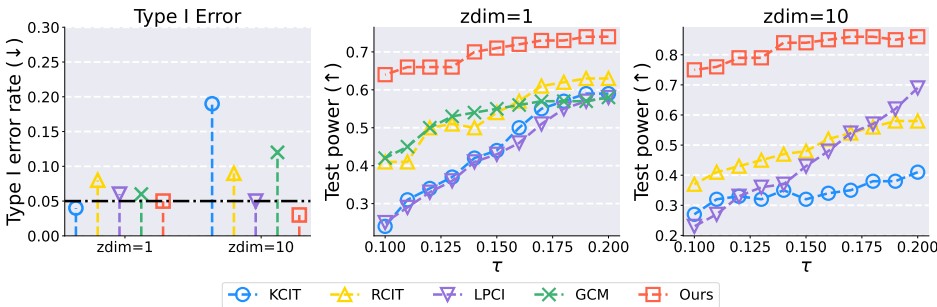

*Figure 9.* Type I error rates and Powers of Data Generation I (DG I) with different sizes of the conditional variable..

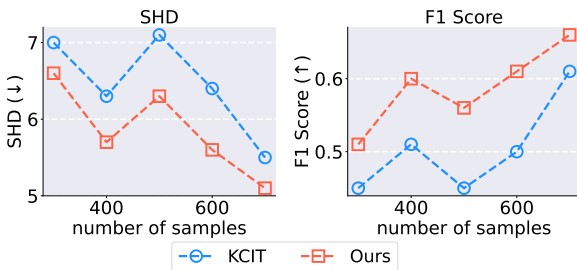

*Figure 10.* SHD and F1 Score of the causal discover experiment. We follow the data generation and change the number of samples $n \in [300, 400, 500, 600, 700]$.

**Result.** We use PC algorithm with KCIT as our baseline. The result can be found in Fig. 10. Our RD-PC consistently performs well here.

### G.3. Real-world Data

The the flow-cytometry data published by (Sachs et al., 2005) is a popular real-world data set for causal discovery methods, which gives expression levels of 11 proteins under various experimental conditions. We take the popular learned causal structures as the ground-truth causal graph for this dataset, as shown in Fig. 11.

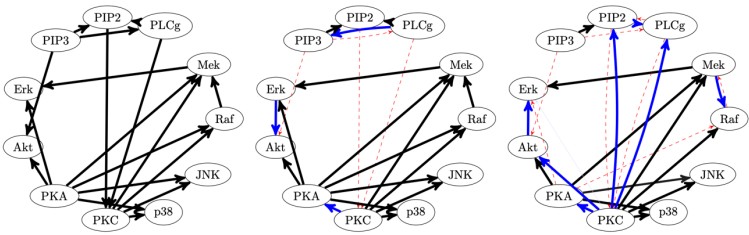

*Figure 11.* Figure 5 in (Mooij & Heskes, 2013). **Left:** Consensus network, according to (Sachs et al., 2005); **Middle:** Reconstruction of the signaling network by (Sachs et al., 2005), in comparison with the consensus network; **Right:** The best acyclic reconstruction found by (Mooij & Heskes, 2013). Black edges: expected. Blue edges: unexpected, novel findings. Red dashed edges: missing.

## H. Discussions

### H.1. Why not Use Mutual Information?

In principle, we can also detect rare dependence by maximizing the mutual information between $X$ and $Y$ on the reweighted sample. However, maximizing the mutual information with the importance reweighted data involves estimation of the data densities, which is a difficult problem. Instead, we maximize a specific kernel-based dependence measure on the data w.r.t.

the importance reweighting ratio $\beta(C)$.

### H.2. Kernel Choice

The performance of RHSIC/RKCIT depends on the choice of the kernel. Our theoretical claims require a characteristic kernel. In our experiments (including Examples 1.1 and 3.1), we use the Gaussian kernel with median heuristic bandwidth.

We conducted additional experiments comparing kernels and found Gaussian consistently performs best. The poor performance of the polynomial kernel is expected, as it is not characteristic. Laplace is characteristic but has heavier tails than Gaussian; its performance drop suggests lighter-tailed kernels may be preferable.

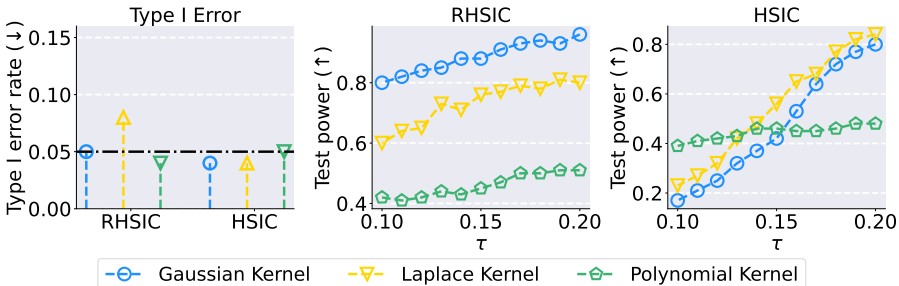

*Figure 12.* Type I error and test power comparisons of RHSIC and HSIC with different choices of kernels. Used the DG III in Appendix with $\tau = 0.1$.

### H.3. Selection of Reference Variable $C$

When do UI/CI tests where only $X$ and $Y$ are observed, we consider $C = X$ or $C = Y$. Once more information is available, e.g. more observed variables as in causal discovery, $C$ can be a third variable to leverage such information.

Take the DG III $\tau = 0.1$ as an example, in the causal graph $X \leftarrow \epsilon_b \rightarrow Y \leftarrow Q$, we observe $X, Y, Q$ and want to test $X \perp Y$. Proposition C.2 shows that $C = X, Y$, or $Q$ are all valid here (i.e., do not introduce spurious dependence). RHSIC with $C = Q$ performs the best since $Q$ directly controls the dependence between $X$ and $Y$, while $C = Y$, a child of $Q$, brings some power loss though still acceptable. RHSIC with $C = X$ is ineffective since $X \perp Q$.

| Method | Type I Error ↓ | Test Power ↑ |
|---|---|---|
| RHSIC($C = Q$) | 0.05 | 0.8 |
| RHSIC($C = Y$) | 0.1 | 0.57 |
| RHSIC($C = X$) | 0.01 | 0.06 |
| HSIC | 0.04 | 0.17 |

*Table 2.* Comparison of Type I error and Power across different choices of the reference variable $C$ for RHSIC.

In practice, if only $X, Y$ are available, we recommend testing with both $C = X$ and $C = Y$ and selecting the lower $p$-value.

### H.4. Train-Test Splitting Ratio

Different train-test splitting may lead to different performance. However, RHSIC significantly outperforms baselines when rare dependence exists, suggesting the splitting loss is limited and outweighed by the gain in detecting rare dependencies. To examine the effect of splitting on performance, we list different split ratios on DG III $\tau = 0.1$. A 0.5 ratio generally performs well, consistent with prior work (Jitkrittum et al., 2016; 2017).

*Table 3.* On performance under different ratios of test-train splitting

| Train : Test | DG III ($\tau = 0.1$) | | DG I ($\sigma = 0.4$) | |
|---|---|---|---|---|
| | Type I Error ↓ | Test Power ↑ | Type I Error ↓ | Test Power ↑ |
| 90% : 10% | 0.05 | 0.41 | 0.05 | 0.28 |
| 80% : 20% | 0.07 | 0.61 | 0.08 | 0.47 |
| 70% : 30% | 0.07 | 0.68 | 0.04 | 0.59 |
| 60% : 40% | 0.05 | 0.76 | 0.05 | 0.63 |
| 50% : 50% | 0.05 | 0.80 | 0.05 | 0.69 |
| 40% : 60% | 0.04 | 0.78 | 0.05 | 0.69 |
| 30% : 70% | 0.01 | 0.74 | 0.06 | 0.71 |
| 20% : 80% | 0.01 | 0.70 | 0.03 | 0.57 |
| 10% : 90% | 0.04 | 0.68 | 0.05 | 0.47 |

