# OpenReview forum: "Extracting Rare Dependence Patterns via Adaptive Sample Reweighting"
_ICML.cc/2025/Conference — ICML 2025 poster_

### Official Review · Reviewer_6TwN · 2025-03-13

**Overall Recommendation:** 3

**Summary:**

This paper tackles independence testing, when there is rare dependence. Rare dependence is defined as the case when most of the data points exhibit independent behaviour between two variables, but a subset exhibits dependence. The authors propose to solve this problem by augmenting the dataset with weights that are a function of some reference variable, that reweight the samples by maximising the HSIC test statistic for dependence (maximising a measure of dependence). The authors derive the null distribution, show that it is too complex and calculate the required quantile using a permutation test. A conditional independence test version is also introduced. Experiments show that the tests perform better than baselines.

## update after rebuttal
On reading the response and the other reviews, I will keep my score.

**Claims And Evidence:**

The only claim that is not thoroughly tested is the fact that rare dependence is actually an issue in reality. The authors also test their causal discovery method in the appendix, however, I would also like to see application of their method on a consensus benchmark (one that is not constructed to exhibit rare dependence), to see what is lost in this case.

**Essential References Not Discussed:**

Up to my knowledge the required references are discussed in Appendix B.

**Experimental Designs Or Analyses:**

The experiments seem sound (see above for some issues).

**Methods And Evaluation Criteria:**

The authors test only on examples where rare dependence is synthetically created. Some real world example of this would be beneficial. Furthermore, it would be interesting to see what is lost compared to baselines if the data does not actually exhibit rare dependence.

**Other Comments Or Suggestions:**

It is not quite clear to me how to select the reference variable C. From the motivating examples, it seemed to me that it has to be either X or Y, but the experiments seem to suggest that it can be a third variable as well. How is this variable chosen in practice? What are the effects of choosing one variable over another given a certain causal graph?

**Other Strengths And Weaknesses:**

Strengths:
- The paper is tacking an issue that has not been tackled before.
- The solution is sound with experiments showing improvement over the baselines.

Weaknesses:
- The paper does not convincingly argue that rare dependence is a problem in practice.
- The splitting into test and train may lead to a loss of performance compared to other baselines. This has not been explored properly.

**Questions For Authors:**

- Figure 3 is not clear at all, n1 and n2 are not defined, the text around the figure is also confusing. I can see what point is being made, but unsure how the figure is helping here.
- Eq 1, why do you want the expected $\beta(C)$ to be equal to 1? This should be explained in the text.
- L169 LHS, it might make sense to say that the test statistic is being maximised (instead of optimised).
- KCIT and RKCIT are not formally defined in Section 4.
- I'm unsure why your rules are defined in terms of KCIT and RKCIT both? The relation and the compromises between KCIT and RKCIT should be explained properly here.

**Relation To Broader Scientific Literature:**

I have not seen the issue of rare dependence tackled in this way.

**Theoretical Claims:**

I can't seem to find the proof for proposition 3.6. The proofs in the Appendix are also numbered differently than the main paper, this makes finding the exact proof for a claim a little difficult. Furthermore, it might make sense to write (at the start of the section or where the proof is written), exactly where this is proven.

The rest of the proofs seem correct.

---

> ### Author Rebuttal · Authors · 2025-03-31
>
> We thank the reviewer for the detailed and constructive feedback. Below we address all raised points grouped by topic. All figures and tables are available at https://tinyurl.com/mwafx6kh.
>
> -  **Rare dependence in reality** (Claims & Methods & W1)
> We clarify that we have evaluated our method on a real-world dataset (Sachs et al., 2005) in Sec. 5.2. We agree that real-world support is important and include additional real-world experiments (see first & second responses to Reviewer MSVk).
>
> - **Consensus datasets** (Claims & Methods)
> We tested our method on three standard dependence benchmarks: Data Generation III (DG III) in Appendix with $\tau=1$, i.e. totally dependent; Sinusoid in (Ren et al., 2024) and ISA in (Gretton et al., 2007). As shown in the table, RHSIC slightly underperforms full HSIC due to data splitting to learn reweighting function (split ratio=0.5), but outperforms HSIC using half of the data.
>   In practice, when we are unsure whether rare dependence exists, it is preferable to use HSIC first, and then apply RHSIC if HSIC fails.
>
>   |Method|RHSIC||HSIC||HSIC(n/2)||
>   |-|-|-|-|-|-|-|
>   |Data Generation|Type I ↓|Power ↑|Type I ↓|Power ↑|Type I ↓|Power ↑|
>   |DG III|0.03|0.99|0.04|1|0.04|0.67|
>   |Sinusoid|0.04|1|0.05|1|0.02|1|
>   |ISA|0.09|0.16|0.1|0.26|0.1|0.15|
>
> - **Test-train splitting** (W2)
> We agree that splitting for reweighting may cause a performance drop compared to full-data baselines, as shown above. However, RHSIC significantly outperforms them when rare dependence exists, suggesting the splitting loss is limited and outweighed by the gain in detecting rare dependencies.
>   To examine the effect, we list different split ratios on DG III $\tau=0.1$. A 0.5/0.5 ratio generally performs well, consistent with prior work [1,2,3].
> |Tr:Te|Type I ↓|Power ↑|
> |-|-|-|
> |7:3|0.07|0.68|
> |5:5|0.05|0.8|
> |3:7|0.01|0.74|
> |HSIC|0.04|0.17|
> - **Reference variable C selection** (Comments)
> When do UI/CI tests where only X and Y are observed, we consider C=X or Y. Once more information is available, e.g. more observed variables as in causal discovery, C can be a third variable to leverage such information.
>   Take the DG III $\tau=0.1$ as an example, in the causal graph X<--$\epsilon_b$-->Y<--Q, we observe X, Y, Q and want to test X $\perp$ Y. Proposition C.2 shows that C=X, Y, or Q are all valid here (i.e., do not introduce spurious dependence). RHSIC with C=Q performs the best since Q directly controls the dependence between X and Y, while C=Y, a child of Q, brings some power loss though still acceptable. C=X is ineffective since X $\perp$ Q.
> |Method|Type I ↓|Power ↑|
> |-|-|-|
> |RHSIC (C=Q)|0.05|0.8|
> |RHSIC (C=Y)|0.1|0.57|
> |RHSIC (C=X)|0.01|0.06|
> |HSIC|0.04|0.17|
>
>   In practice, if only X, Y are available, we recommend testing with both C=X and C=Y and selecting the lower p-value.
>
> - **Proof for Prop. 3.6** (Theoretical)
>  We are sorry for omitting the proof for Proposition 3.6 in the submission, and we include it below. The proof uses the independence preservation under measurable transformations.
>
>   *Proof.*  Since the data are i.i.d., $D_{tr} \perp D_{te}$.  As $\hat{\beta} = f(D_{tr})$ is a measurable function of $D_{tr}$, it follows that $\hat{\beta} \perp D_{te}$  (Thm 4.3.5 in [4]). $\square$
>
>    We also agree that the appendix numbering could be improved, and we will revise it to clearly indicate where each proposition is proven.
>
> - (Q1) We have updated Fig. 3 (see the link above) and will revise the surrounding text for clarity. We use it to visualize that the original HSIC may fail to detect dependence in rare dependence settings, even if we have infinite samples.
>
> - (Q2) We should make sure the reweighted p.d.f., $\tilde{\mathbb{P}}(X,Y)=\beta(C)\mathbb{P}(X,Y)$, is still a well-defined one, i.e. $\int_{\mathcal{X\times Y}}\tilde{\mathbb{P}}(X,Y)dXdY=1$, which gives us $\mathbb{E}[\beta(C)]=1$. We will clarify this in our manuscript.
>
> - (Q3&4) Thank you for your careful reading. We will correct the terminology in L169 and formally define KCIT and RKCIT in Section 4.
>
> - (Q5) Thanks for your question. In our rules, we employ KCIT to detect non-rare dependencies and RKCIT to detect rare ones. While RKCIT is capable of identifying non-rare dependencies as well, its reliance on data splitting may introduce unnecessary statistical inefficiency. Therefore, we prefer KCIT in such cases. Assumption 4.1 guarantees the reliability of KCIT when it rejects the null hypothesis. However, if KCIT fails to reject, this could be due to the presence of a rare dependence, in which case RKCIT serves as a complementary test.
>
> We sincerely appreciate your thoughtful feedback and recognition of our work. Thank you for your time.
>
> [1] Interpretable distribution features with maximum testing power. NeurIPS 2016.
>
> [2] An adaptive test of independence with analytic kernel embeddings. ICML 2017.
>
> [3] Learning adaptive kernels for statistical independence tests. AISTATS 2024.
>
> [4] Casella & Berger, Statistical Inference.

---

### Official Review · Reviewer_L7CG · 2025-03-14

**Overall Recommendation:** 3

**Summary:**

This paper considers the discovery of the dependence pattern in a specific small region, coined "rare dependence". The authors proposed a reweighting (importance sampling) -based approach and presented several statistical properties. They also demonstrated several applications, e.g., causal discovery, on synthetic and real-world datasets.

## update after rebuttal
I appreciate the author addressing some of my concerns, and I will keep my score.

**Claims And Evidence:**

The theoretical claims are clear in general. However, it is unclear how the kernel choice might affect these claims. In particular, the value of HSIC is subject to the kernel choice, which should significantly impact the results, e.g., the independence test.

**Essential References Not Discussed:**

Recent works on dependence and conditional dependence learning have extended Rényi's maximal correlation to general analyses of dependence and conditional dependence, e.g.,

[XZ2024] Xu, Xiangxiang, and Lizhong Zheng. "Neural feature learning in function space." Journal of Machine Learning Research 25.142 (2024): 1-76.

For example, the covariance ($\Sigma$) -based criterion also appeared in analyzing maximal correlation [XZ2024]. The construction of $\ddot Y$ (cf. Lemma 3.10 of the manuscript) also appeared in [XZ2024] without the kernel assumptions. It could be interesting to compare these analyses and approaches.

**Experimental Designs Or Analyses:**

The experiment designs are reasonable.

**Methods And Evaluation Criteria:**

The proposed methods and evaluations are reasonable.

**Other Comments Or Suggestions:**

The presentation can be improved. In general, the notations can be so heavy that key ideas are deeply buried in the equations.

Examples:
1. The characteristic kernel was not defined in its first appearance (end of Sec. 2);
2. The notation $\mathcal{H}$ was used to indicate both hypothesis and the RKHS.

**Other Strengths And Weaknesses:**

Strength: The theoretical claims are sound.

Weaknesses: see questions.

**Questions For Authors:**

Q1: How do kernel choices affect the results? In practice, how should we select the kernel (e.g., to compute the HSIC claimed by Examples 1.1 and 3.1)?

Q2: Why use two different examples (1.1 and 3.1)?

Q3: Section 3.2, first line: "two disjoint sets of variables": is X a set of random variables or a single random variable? Similarly, the definition of C as a "subset of X or Y" is problematic. Note that this definition of sets does not apply to examples 1.1. and 3.1, and from later contexts, X and Y seem to be random variables instead of sets.

Q4: Can the designed reweighting be extended to a more general case? Note that in the manuscript, it was assumed that the rare dependence happens in a rectangular region. (equivalently, in Eq. (1), C can only be a "subset" of X or Y, not arbitrary shapes.) This might not necessarily be the case when considering nonlinear dependence.

**Relation To Broader Scientific Literature:**

The proposed approach could potentially help improve the performance of causal discovery, which has broader applications in scientific domains.

**Theoretical Claims:**

The results are mostly intuitive despite the heavy mathematical notations. The proofs seem reasonable, though I did not check them line by line.

---

> ### Author Rebuttal · Authors · 2025-04-01
>
> We appreciate the reviewer's constructive comments and helpful feedback. Please see below for our response. All figures and tables are available at https://tinyurl.com/mwafx6kh.
>
> - **On kernel choice** (Comments & Q1)
>   Indeed, RHSIC/RKCIT performance depends on kernel choice. Our theoretical claims require a characteristic kernel. In practice (e.g., Examples 1.1 and 3.1), we use the Gaussian kernel with median heuristic width, as in standard HSIC.
>
>     We conducted additional experiments comparing kernels and found that Gaussian consistently performs best. The poor performance of the polynomial kernel is expected, as it is not characteristic. Laplace is characteristic but has heavier tails than Gaussian; its performance drop suggests that lighter-tailed kernels may be preferable.
>   |Method (Kernel)|Type I ↓|Power ↑|
>   |-|-|-|
>   |RHSIC (Gaussian)|0.05|0.8|
>   |RHSIC (Laplace)|0.08|0.6|
>   |RHSIC (Polynomial)|0.04|0.42|
>
> - **On related work** (References)
>   We thank the reviewer for pointing out this related work. While our goals differ — ours focus on hypothesis testing, theirs on representation learning — both leverage dependence-maximization criteria to learn meaningful functions. We will cite and discuss [XZ2024] in the manuscript.
>
>   [XZ2024] proposes a feature learning framework maximizing dependence with the target using an extension of Rényi maximal correlation (RMC) in $L^2$ space, with functions parameterized via neural networks. In contrast, we focus on detecting rare dependence using HSIC/KCI, widely adopted in independence testing, and learn a reweighting function to assign sample-wise importance. Theoretically, RMC provides a general measure of nonlinear dependence via optimal function pairs in $L^2$. Modal decomposition captures the full spectrum of the cross-covariance operator, with the leading mode corresponding to RMC. Restricting to RKHS and aggregating all squared singular values yields HSIC. While RMC is more general, its estimation and statistical inference in $L^2$ with neural networks might be more challenging. In contrast, our methods inherit the kernel-based formulation of HSIC, allowing efficient estimation by kernel trick and asymptotic distribution analysis with statistical guarantees.
>
> - (Q2) Examples 1.1 and 3.1 illustrate two types of rare dependence and show a shared solution. Example 3.1 is *local*, with $X \perp Y$ holding in a subregion. Example 1.1 shows *global but weak* dependence, where the signal exists throughout but is largely buried by noise. Both of them motivate our method, which adaptively reweights samples to better reveal underlying dependence patterns.
>
> - (Q3) We thank the reviewer for pointing this out. We acknowledge the inconsistency in notation: X, Y are introduced as random variables in Sec. 2 but later referred to as variable sets. We will unify the notation to "random variables or sets of variables". Most examples in our paper treat X and Y as variables for simplicity and illustration purpose. The phrase “C is a subset of X or Y” assumes that X and Y are sets of variables; when they are individual random variables, C is either X or Y. We agree this was ambiguous and will rephrase accordingly.
>
> - (Q4)  We thank the reviewer for this question. Our reweighting framework is general and does not assume rare dependence lies in a rectangular region. In nonlinear Example 1.1, axis-aligned boxes are purely for visualization; they are not a limitation of the method. The nonlinearity is captured by the reweighting function $\beta(C)$, which is flexible and nonlinear. It assigns higher weights to samples more likely to exhibit dependence, based on their values of C. This allows the method to adapt to complex dependence structures - as long as some signal is present along the reference variable C. Hence, our method naturally handles nonlinear rare dependence.
>
>   We say that C can be either X or Y, but not both. As mentioned in Proposition 3.3, C being either X or Y preserves the independence between them and avoids introducing spurious dependence. However, using $C=(X,Y)$ in Eq. (15) makes the reweighted distribution $\tilde{\mathbb{P}}_{XY}=\beta(X,Y)\mathbb{P}_X\mathbb{P}_Y$ no longer factorizable in X and Y. This means the reweighed distribution changes the original independence relation. We will revise the manuscript to clarify these points.
>
> - **On notation and presentation** (Comments)
>
>   We appreciate the reviewer's comments. We will define "characteristic kernel" upon its first mention (end of Sec. 2), and revise ambiguous notation — in particular, replacing $\mathcal{H}$ with $\mathcal{F}$, denoting RKHSs of X, Y, Z as $\mathcal{F}_X, \mathcal{F}_Y, \mathcal{F}_Z$. We will also provide intuitive explanations and examples for each theorem or proposition when it appears, and streamline the notation for clarity.
>
> We sincerely thank the reviewer for your recognition and your valuable feedback that helps improve the presentation of our submission.

---

### Official Review · Reviewer_MSVk · 2025-03-15

**Overall Recommendation:** 4

**Summary:**

Existing conditional independence testing methods suffer to detect dependencies that occur in a small regions of the data which is referred to as rare dependence. This work aims to resolve this issue by proposing  a kernel-based independence testing with an importance reweighting, which assigns higher weight to data point that involves such rare dependencies. Theoretical analysis and experimental results demonstrate the validity and effectiveness of the proposed method.

**Claims And Evidence:**

The paper proposes a (conditional) independence tests for detecting rare dependencies and provides a theoretical guarantee regarding bound and asymptotic property. The effectiveness of the method is evaluated with two synthetic data generating process and one real-world dataset.

**Essential References Not Discussed:**

I think the motivation should be better presented since in it’s current form, it might be unclear whether such rare dependencies are prevalent in real-world applications which is related to the significance of the problem. Accordingly, I suggest the authors to include the literature on local independences [1-5] to discuss and better motivate the relevance of “rare dependencies” in practical applications.

**Experimental Designs Or Analyses:**

- Experimental setup looks reasonable. That being said, evaluation on only a single real-world dataset seems a bit narrow.

**Methods And Evaluation Criteria:**

- The method and evaluation criteria are appropriate for the task.

**Other Comments Or Suggestions:**

See above and below.

**Other Strengths And Weaknesses:**

**[Strengths]**

- The paper is well-written and easy to follow.
- The paper tackles an important problem and theoretical analysis looks solid. Another strength is the application to causal discovery and corresponding evaluation.

**[Weaknesses]**

- The motivation is somewhat weak in the current manuscript, since some readers might question the importance and relevance of “rare dependencies” in real-world datasets and scenarios.
- The authors mention that discovering local (in-)dependencies (e.g., context-specific independence [1, 2] or local independence [3]) are “opposite to our objective”. Can you elaborate on what this means? Intuitively, such local (in-)dependencies can be regarded as a special case of “rare dependence”, i.e., soft vs. hard, and I don’t see any reason why it is opposite to the scope of this work. It would be interesting to see how the proposed method performs in such settings.
- The evaluation considers only a single real-world dataset and the proposed method is limited to continuous variables.

**Questions For Authors:**

- Could the proposed method be evaluated on discovering local (in-)dependencies?
- One desirable property is to find a particular region of the data where the “rare dependence” exhibits (e.g., [-2, 2] in Fig. 1 and [0,0.25] in Fig. 2) as it provides an interpretability and could be further leveraged, e.g., for efficient [4], robust inference [5], and causal effect identification [6]. I assume the proposed method could be further extended to capture such specific subgroup of the data, e.g., by collecting datapoints with large importance weights; I would like to hear the thoughts from the authors.
- Could the proposed method be extended to discrete variables?

***References***

[1] Context-specific independence in Bayesian networks

[2] The role of local partial independence in learning of Bayesian networks

[3] On discovery of local independence over continuous variables via neural contextual decomposition

[4] Exploiting contextual independence in probabilistic inference

[5] Fine-grained causal dynamics learning with quantization for improving robustness in reinforcement learning

[6] Identifying causal effects via context-specific independence relations

**Relation To Broader Scientific Literature:**

The dependency pattern within the data is heterogenous and imbalanced in many practical scenarios (e.g., economics, biology, social science). I believe the paper has potential impact to broader scientific literature.

**Theoretical Claims:**

- The paper provides an asymptotic analysis on the importance reweighting statistics of the test. Theoretical claims seem to be convincing, though I did not check the proof in detail.

---

> ### Author Rebuttal · Authors · 2025-03-31
>
> We thank the reviewer for the insightful and constructive feedback. Please see below for our response. All figures and tables are available at https://tinyurl.com/mwafx6kh.
>
> - **On Motivation** (Reference & W1)
>   We thank the reviewer for this valuable suggestion. We agree that the motivation can be strengthened, especially regarding real-world relevance. We reviewed the literature on local and context-specific independence [1–5], which supports the practical significance of rare dependence. For instance, [3] applies CSI to physical dynamics such as friction; [5] demonstrates local dependence in reinforcement learning, e.g., autonomous driving; [6] discusses CSI in biomedical inference, e.g., dose-response effects in antibiotic treatment. These works are insightful and we will cite and integrate these as examples.
>
>   Rare dependence is common across domains. In economics, income and consumption may appear weakly related in low-income groups but become strongly coupled at higher income levels. In psychology[7], the impact of social media on adolescent mental health becomes significant only with excessive usage. Similar rare dependencies occur in medicine[8], physics[9], and sociology[10], highlighting the practical relevance of detecting rare dependencies.
>
>   We will revise the motivation accordingly and include references [1–5] and these real-world examples to better contextualize the relevance of rare dependence in practice.
>
> - **On real-world datasets** (Experimental & W3)
>
>   We agree that additional real-world evaluation would further strengthen the work. In response, we have added an experiment **(see link above)** using monthly JPY/USD exchange rates (E) and U.S. federal funds rates (F) from 1990 to 2010, sourced from FRED. While the original HSIC fails to reject independence (p = 0.2174), RHSIC detects dependence (p = 0.0005) using F as the reference variable. The learned weights assign higher importance to the samples in 2001 and 2008. These correspond to the Dot-com recession and the global financial crisis, respectively — showing that our method not only detects rare dependence but also provides interpretable insights.
>
> - **On local (in-)dependencies** (W2 & Q1)
>
>   We thank the reviewer for this thoughtful question. We apologize for the misleading wording "opposite" and will revise it accordingly. We intended to highlight a difference in focus — which we now realize is not necessarily exclusive: prior works aim to model precise conditional independence structures, while our goal is to detect rare dependence overlooked by existing tests.
>
>   We agree that our methods can deal with context-specific or local (in-)dependence problems. For example, Example 3.1 and Data Generation II & III in our experiments involve local dependence, and our method performs well in these cases. Moreover, the learned sample weights can help identify independent regions — low-weight samples often correspond to locally independent areas. We will clarify this point in the revision.
>
> - (Q2) Indeed, one advantage of our method is that the learned importance weights provide a natural way to identify subgroups of data that contribute most to the rare dependence signal. As shown in our real-world dataset analysis and local-independence discussion above, these weights help uncover meaningful patterns and guide downstream interpretation, enhancing interpretability.
>
>   We agree that explicitly extracting high-weight subgroups could further extend the utility of our method. In particular, the learned weights can support fine-grained causal structure discovery by highlighting context-dependent relationships. Moreover, the structures recovered by our approach (e.g., the algorithm in Sec. 4) can provide valuable inputs for downstream tasks. Both aspects are relevant to applications discussed in [4,5,6], as suggested by the reviewer. We view this as a promising direction for future work.
>
> - (Q3) Yes. Since kernel matrices can be constructed for discrete variables using Kronecker kernels [11], our method — which operates on these matrices — directly applies to the discrete case.
>
> We sincerely thank you for your constructive feedback, the time you have dedicated, and your recognition of our work. We especially appreciate the valuable suggestions regarding future directions, which we find highly inspiring and will consider in our subsequent research. Thank you.
>
> [7] A Systematic Review: The Influence of Social Media on Depression, Anxiety and Psychological Distress in Adolescents.
>
> [8] Opioid-induced Hyperalgesia: A Qualitative Systematic Review.
>
> [9] Frequency-Dependent Local Interactions and Low-Energy Effective Models from Electronic Structure Calculations.
>
> [10] Time Spent on Social Network Sites and Psychological Well-Being.

---

> > ### Comment · Reviewer_MSVk · 2025-04-01
> >
> > I appreciate the authors for the rebuttal. I have read the author response and my concerns are now well-addressed.
> >
> > Specifically:
> > - As acknowledged by the authors, the paper will benefit from better motivation and how the rare dependencies arise in real-world scenarios. I noticed that the reviewer 6TwN shares the same concern. Please include the relevant discussions on the motivation and local independences and references which would further strengthen the paper.
> > - Thanks for the experiments on the additional real-world dataset.
> > - Thanks for acknowledging the implications and the importance of identifying subgroups of the data where the rare dependencies arise. It's nice to see that the proposed method does naturally provide this interpretability through the learned weights. Please include the discussions in the revised version.
> >
> > Accordingly, I increased my score from 3 to 4.

---

> > > ### Author Response · Authors · 2025-04-02
> > >
> > > We sincerely thank the reviewer for taking the time to read our rebuttal and for the recognition of our work. We are glad that our responses addressed your concerns. We will incorporate the suggested revisions into the revised version of the paper. Thank you again for the constructive suggestions, and we truly appreciate your feedback.

---

### Official Review · Reviewer_YWTp · 2025-03-17

**Overall Recommendation:** 3

**Summary:**

This paper proposes the use of adaptive sample weighting to detect rate dependencies in data. The key idea is to incorporate weights for data points by formulating an objective problem that maximizes the reweighted HSIC along with regularization terms. Asymptotic hypothesis test guarantees for the resulting reweighted HSIC test are provided through direct applications of the classical theory of V-estimators and U-estimators.

The estimation of sample weights is analyzed as a statistical learning problem (empirical risk minimization) using simple arguments from empirical process theory, leading to non-asymptotic uniform convergence guarantees.

Finally, this idea is extended to conditional HSIC, and, building on this, a new version of the PC algorithm is designed that utilizes the reweighted conditional HSIC to infer conditional independence. In the experiments, the strong performance of the new PC algorithm is demonstrated on both synthetic data and a real-world dataset from flow cytometry (Sachs et al., 2005).

**Claims And Evidence:**

Yes, the theoretical claims are indeed classical, and the proofs are provided. The experiments are also discussed in detail.

**Essential References Not Discussed:**

Yes, indeed, the idea of adaptive sample reweighting for causality is not novel and has been proposed before (and is not cited in this manuscript), particularly in:

Zhang, A., Liu, F., Ma, W., Cai, Z., Wang, X., & Chua, T. S. Boosting Causal Discovery via Adaptive Sample Reweighting. In The Eleventh International Conference on Learning Representations.

In that work, adaptive sample reweighting was formulated as an optimization problem in combination with score-based causal learning algorithms. In contrast, this paper considers a constraint-based approach (PC). However, in general, this prior work significantly undermines the novelty of this submission.

**Experimental Designs Or Analyses:**

Yes, they seem to be valid.

**Methods And Evaluation Criteria:**

Yes, test power and Type I error are reported for the experiments, which is common in the causality literature.

**Other Comments Or Suggestions:**

Besides the missing reference, I don't have any other suggestions.

**Other Strengths And Weaknesses:**

Strengths:
- Paper is well-written and all aspects of theory are analyzed (despite following classical results and not necessarily very novel) in depth, especially on asymptotics of hypothesis testing, causal discovery of the resulting PC algorithm up to markov blanket and uniform convergence results for estimation of the sample weights.
- A comprehensive set of synthetic and real-world experiments are provided.

Weakness:
- The idea is not novel and similar idea have been proposed before (Zhang, An, et al. 2023).

Because the paper is complete (in terms of theory and experiments), I am leaning toward acceptance.

**Questions For Authors:**

The paper is well-written, and I don’t have any questions.

**Relation To Broader Scientific Literature:**

The authors design a new independence test, leading to a new class of constraint-based causal discovery algorithms, which may have broader impacts in applications such as biology or the human sciences.

**Theoretical Claims:**

Yes, I skimmed briefly, and they seem to be correct.

However, one important note should be mentioned in the paper: the uniform bound in Theorem 3.7 is not designed for the optimization problem (9) but rather for the version of the empirical risk that does not include any regularization terms. With regularization, the term $B$ in line 1080 is not negative, and as a result, the entire conversion of the problem into uniform convergence bounds breaks down.

---

> ### Author Rebuttal · Authors · 2025-04-01
>
> We appreciate the reviewer's constructive comments and helpful feedback. Please see below for our response. All figures and tables are available at https://tinyurl.com/mwafx6kh.
>
> - **(Weakness & Reference)**
> Thank you for your comments and for pointing out the work by Zhang et al. (2023), which we will cite in our revised version. **Our work and Zhang et al. (2023) originate from different motivations and aim to address distinct problems.** Our research is motivated by the specific challenge of detecting rare dependence, a topic that remains unexplored to the best of our knowledge. Rare dependence is widespread across many real-world domains and confirmed by lots of literature, such as psychology[1], medicine[2], physics[3], sociology[4], etc. Our focus is on **detecting rare dependencies** that are otherwise missed by existing independent tests. The proposed RHSIC and RKCIT are statistical tools for independence and conditional independence testing designed to recover rare dependencies. We emphasize that discovering causal relations in the presence of rare dependence is one important application of our methods, not the ultimate goal. Our method is not restricted to causal discovery.
>
>   On the other hand, ReScore (Zhang et al., 2023) aims to optimize the structure recovery of score-based causal discovery methods with fewer spurious edges and more robustness to heterogeneous data. They propose an effective model-agnostic framework of reweighting samples to boost differentiable score-based causal discovery methods. The core idea of ReScore is to identify and upweight less-fitted samples, as these samples provide additional insight into uncovering the true causal edges—compared to those easily fitted through spurious associations. Although RD-PC, which employs our proposed RHSIC and RKCIT with correction rules, is also a sample reweighting-based method for causal discovery, its objective is fundamentally different from that of ReScore. ReScore is designed to mitigate the influence of spurious edges by focusing on less-fitted samples, whereas RD-PC aims to recover true causal edges that are easily overlooked or erroneously removed due to rare dependence. Exploring the connection between rare dependence and less-fitted samples may offer an interesting direction for future research.
>
>   We appreciate the opportunity to clarify this and will update the manuscript accordingly.
>
> - **(Theoretical claims)**
> Thank you for your insightful comments. We agree that the uniform bound in Theorem 3.7 applies to the empirical risk without regularization terms, rather than to the optimization problem (9). We acknowledge that this discrepancy exists and appreciate the opportunity to address it. We should have discussed this in detail in the paper. A gap remains between the theoretical aspect and the current algorithmic implementation. We have included a discussion in our updated manuscript. In practice, however, when analyzing data, we have to introduce a trade-off by employing the normalized RHSIC with two penalties in (9): the first term to control the smoothness of the $\hat{\beta}$ function, and the second to constrain the deviation of $\hat{\beta}$ from one to select as many samples as possible. These two penalties allow us to balance theoretical rigor with practical performance in practice.
>
>   We appreciate the reviewer for raising this issue. To address this concern more thoroughly, we will revise the manuscript by adding an ablation study in the Experiment Section to quantify the regularization’s effect and discuss this theoretical-practical tradeoff explicitly.
>
> We are encouraged that the reviewer recognizes our work as "well-written" and our theoretical analysis as "in depth." We also appreciate the reviewer’s acknowledgment of our comprehensive synthetic and real-world experiments.
>
> Besides, we would like to add that our contribution goes beyond the classical results that the reviewer mentioned; we identify a new problem setting, rare dependence in statistical testing, and establish a general framework to solve this problem. In real-world scenarios, such rare dependencies often arise due to noise or imbalanced data distribution, making them difficult to detect using standard tests. To this end, we hope this work provides a foundation for various potential practical extensions, such as interpretable subgroup discovery, robust causal inference, and scientific analysis in domains like medicine, economics, and social science.
>
> We sincerely appreciate your thoughtful feedback and recognition of our work. Thank you for your time and efforts.
>
> [1] A Systematic Review: The Influence of Social Media on Depression, Anxiety and Psychological Distress in Adolescents.
>
> [2] Opioid-induced Hyperalgesia: A Qualitative Systematic Review.
>
> [3] Frequency-Dependent Local Interactions and Low-Energy Effective Models from Electronic Structure Calculations.
>
> [4] Time Spent on Social Network Sites and Psychological Well-Being.

---

> > ### Comment · Reviewer_YWTp · 2025-04-03
> >
> > Thank you very much for the detailed response and clarification, especially regarding Zhang et al. (2023). As I mentioned earlier, this is a complete and well-executed paper, though the extent of its contribution is not game-changing. For this reason, I am leaning toward acceptance, and I believe my score accurately reflects my evaluation.

---

> > > ### Author Response · Authors · 2025-04-06
> > >
> > > Thank you very much for your time, thoughtful comments, and engagement throughout the review process. We truly appreciate your recognition of the completeness and clarity of the paper, and we're glad our responses were helpful.

---

### Decision · Program_Chairs · 2025-05-01

**Decision:**

Accept (poster)

**Comment:**

The paper proposes a novel kernel-based conditional independence test method. The method captures rare dependence patterns through adaptive sample importance reweighting. The authors incorporate this conditional independence test into the PC algorithm and demonstrate its effectiveness using both synthetic and real-world datasets. The paper addresses an important problem in causal discovery. The method comes with theoretical guarantees, including bounds and asymptotic properties.